# CRACKING THE HESSIAN: CLOSED-FORM HESSIAN SPECTRA FOR FUNDAMENTAL NEURAL NETWORKS

## ABSTRACT

The Hessian and its spectrum hold significant theoretical and practical relevance for building optimizers, measuring generalization, compressing models, and more. Prior works have characterized the Hessian through its spectral density, rank, and the outlier–bulk structure of its spectrum, often relying on approximations. However, the precise behavior of Hessian eigenvalues and eigenvectors remains unclear, owing both to the absence of closed-form results for non-trivial neural networks and the computational expense of empirical estimation. In this work, we derive closed-form expressions for *all Hessian eigenvalues and eigenvectors* in two-layer linear and ReLU networks with scalar input, *arbitrary hidden width*, and where the loss is aggregated *over any number of samples*. We further provide closed-form eigenvalues for the core component of Transformer architectures — a single self-attention layer with arbitrary sequence length. Our results reveal a previously undiscovered 'paired' structure of outlier eigenvalues, a cell-wise decomposition of the Hessian spectrum with ReLU, and the sensitivity of the Hessian condition number to the query and key matrix norms, as well as the presence of attention sinks. We complement these findings with experiments beyond the assumed model setting, showing strong correlation between the largest eigenvalue and the spectral norm of weight matrices, and empirical evidence that the paired eigenvalue structure persists more generally. Overall, by establishing these closed forms for the first time, and introducing the corresponding proof technique, we advance our understanding of the Hessian and open new avenues for its use.

## 1 INTRODUCTION

**Motivation.** Modern deep learning architectures induce loss landscapes that are high dimensional and structurally complex. The Hessian of the loss function and in particular its spectrum, meaning the set of all its eigenvalues, are highly informative quantities describing this landscape. As a result, they play a central role in understanding deep learning phenomena, designing memory-efficient models, and developing more effective training algorithms. (Ghorbani et al., 2019; Zhang et al., 2024a; LeCun et al., 1989; Foret et al., 2020; Martens & Grosse, 2015a).

Despite its pivotal role, access to the Hessian remains limited to numerical approximations that trade accuracy for computational efficiency. We bridge this fundamental gap by opening up an alternate line of inquiry that focuses on deriving the closed-form expression of the exact Hessian spectrum, where possible, and then utilizing it to draw key insights that apply broadly. Furthermore, we do this without resorting to any approximations of the Hessian and obtain precise results that are valid at any point during training.

**Challenges in Deriving the Spectrum.** Deriving the spectrum of the Hessian ultimately requires solving its characteristic polynomial, which in turn is obtained from its determinant. Constructing this polynomial is formally straightforward but practically prohibitive, since determinants of large matrices are computationally expensive. Even if the polynomial is available, the Abel–Ruffini theorem rules out closed-form solutions for characteristic equations of degree $n \geq 5$ (Stewart, 2022; Stillwell, 1989). A further challenge lies in the complexity of writing the Hessian itself in closed form: for many neural architectures, the expressions can span pages of derivations even in relatively simple cases (Singh et al., 2021; 2023; Ormaniec et al., 2024). These obstacles explain the reliance on numerical or approximate methods in prior work, and motivate our focus on closed-form derivations in settings where analytic results remain within reach.

**Contributions.** We introduce a proof technique that yields the *complete* spectrum of Hessian eigenvalues for a set of fundamental neural networks architectures. Using it, we:

**(1).** Derive and interpret closed-form expressions for the eigenvalues and eigenvectors of the loss Hessian in specific cases — including a uni-dimensional two-layer linear MLP and a two-layer ReLU MLP — and extend these results to models with arbitrary input or output size under structural assumptions on the weight matrices (Sections 3.1 and 3.2).

**(2).** Uncover a paired nature of the outlier eigenvalues, namely that there exist two eigenvalues that sum up to an input-size dependent fraction of trace, a property that approximately persists to models beyond our assumptions (Sections 3.1 and 4).

**(3).** Show empirically that, for models outside our theoretical assumptions, the spectral norm of certain weight matrices correlates strongly with the largest loss Hessian eigenvalue, a claim that holds across different input and output sizes, widths, depths, and model parametrizations (Section 4).

**(4).** Derive closed-form eigenvalues for the Hessian of a network consisting of a single self-attention layer — a key component of the modern Transformer architecture. This sheds insights on the approximation quality of the Hessian eigenvalues by GGN eigenvalues, the negative impact of larger query and key weight matrix norms on the Hessian condition number, and how attention sinks promote larger outlier eigenvalues. We validate several of the insights on a GPT2 (124M) model in the language modeling setting (Sections 3.3 and 4).

## 2 RELATED WORK

**Relevance of the Hessian.** *(i) Optimization:* As the Hessian characterizes the curvature of the loss landscape, it is crucial for optimization and has thus inspired many prominent optimizers such as Adam (Kingma & Ba, 2015), Kronecker-Factored Approximate Curvature (Martens & Grosse, 2015b), and Shampoo (Gupta et al., 2018). Besides, the Hessian structure and spectra are commonly used to reason about the effectiveness of (batch) normalization techniques (Ghorbani et al., 2019; Yao et al., 2020), understand why Transformers train better with adaptive optimizers than with SGD (Zhang et al., 2024a; Ormaniec et al., 2024), and more. *(ii) Generalization:* investigation by Keskar et al. (2016) on the influence of the batch size on generalization led to the resurgence of the flat-minima idea: the hypothesis that flatter minima generalize better (Hochreiter & Schmidhuber, 1997; Jastrzębski et al., 2017; Chaudhari et al., 2019), with flatness often interpreted via small magnitudes of the top Hessian eigenvalues. While this hypothesis has to be taken with care (Dinh et al., 2017; Andriushchenko et al., 2023), it has inspired a wide range of optimizers that improve generalization (Foret et al., 2021; Kwon et al., 2021) by promoting convergence to flatter solutions instead of sharper ones. *(iii) Applications:* Finally, as the Hessian and its eigenvalues help capture the local geometry of learning, they have also been highly relevant for the purposes of model pruning (LeCun et al., 1989; Hassibi et al., 1993), quantization (Frantar et al., 2022), continual learning (Mirzadeh et al., 2020; 2021), and model merging (Matena & Raffel, 2022).

**Prior Work and Limitations.** The Hessian spectrum has been studied extensively for its implications in optimization and generalization. Early work linked large eigenvalues to optimization difficulties (LeCun et al., 2002), and later studies (Sagun et al., 2016; 2017; Papyan, 2018) identified a characteristic structure: a few large outliers explained by the generalized Gauss–Newton (GGN) component, and a bulk of near-zero eigenvalues. Random matrix theory provided complementary perspectives, including spin-glass models of critical points (Choromanska et al., 2015), approximations for two-layer networks (Pennington & Bahri, 2017), and analyses of the stochastic Hessian where the top eigenvalue controls batch-size–dependent learning rate scaling (Granziol et al., 2022). Much of this literature has focused on the largest eigenvalue, owing to its tractability and tight link to the learning rate. Empirically, it has been noted to grow during training before stabilizing near the edge-of-stability threshold $2/\eta$ (Jastrzębski et al., 2018; Cohen et al., 2020), with larger learning rates producing temporary divergence known as the catapult effect (Lewkowycz et al., 2020).

Despite these advances, most results remain either empirical (Sagun et al., 2016; 2017; Papyan, 2018; Zhang et al., 2024a), which are memory- and compute-intensive, or theoretical but restricted to bounding the largest eigenvalues, often via the generalized Gauss-Newton matrix (GGN) or, in the mean-squared error case, the neural tangent kernel (Wang et al., 2022; Agarwala et al., 2023; Zhao et al., 2024; Noci et al., 2024). *To date, essentially no work has obtained closed-form spectra*

*for non-trivial networks, and none has addressed eigenvectors.* The only rare exception is Zhu et al. (2023), who derived the spectrum of a four-parameter linear network — a result that was incidental to their study of progressive sharpening, and far from models with nonlinearities or arbitrary width.

# 3 CLOSED FORM OF THE HESSIAN SPECTRUM

Although closed-form Hessian eigenvalues are generally intractable as alluded to before, we show that exact formulae can be derived for simplified yet non-trivial networks without any approximations. In this section, we present the closed-form formulae for two-layer linear and ReLU network and a model consisting of a single self-attention layer, as well as discuss the insights they provide. Later, in section 4, we explore how the resulting insights extend to more realistic models.

**Setup.** In the first part of the paper, we consider two-layer neural networks with input dimension $D$ and output dimension $K$ both equal to 1. We denote the models by $f_\theta : \mathbb{R} \mapsto \mathbb{R}$, where $\theta$ is the set of learnable network parameters. We refer the reader to the appendix for extensions to larger input/output dimensions under additional assumptions. We consider mean-squared error (MSE) loss $\ell(\theta) = \frac{1}{2n} \sum_{i=1}^{n} (f_\theta(x_i) - y_i)^2$ computed over a set of $n$ (input, target) data points $\{(x_i, y_i)\}_{i=1}^{n}$, and consider the Hessian $\mathbf{H}_\mathrm{L}$ of this loss with respect to the parameters $\theta$.

## 3.1 LINEAR NETWORK

Consider a one-hidden layer linear network, so $\theta = \{\mathbf{w}, \mathbf{v}\}$ and $f_{\mathbf{w},\mathbf{v}}(x) = \langle \mathbf{w}, \mathbf{v} \rangle x$, with the parameters $\mathbf{w}, \mathbf{v} \in \mathbb{R}^m$. Although this is a simplified setting, it has been put to significant use by past works such as the understanding of the occurrence of the catapults in loss with large learning rates (Lewkowycz et al., 2020) as well as understanding the edge-of-stability behaviour (Zhu et al., 2023). Further, let us define the shorthands for the (uncentered) input standard deviation as $\sigma_x = \sqrt{\frac{1}{n} \sum_{i=1}^{n} x_i^2}$, the (uncentered) input-output covariance as $\overline{yx} = \frac{1}{n} \sum_{i=1}^{n} y_i x_i$, and the (uncentered) residual-input covariance as $\overline{\delta x} = \frac{1}{n} \sum_{i=1}^{n} x_i \delta_i$, where the residual $\delta_i = \langle \mathbf{w}, \mathbf{v} \rangle x_i - y_i$ denotes how far off the network is on fitting the datapoint $(x_i, y_i)$. Then we have the following key result:

> **Theorem 3.1.** For the two-layer linear network $f_{\mathbf{w},\mathbf{v}}(x) = \langle \mathbf{w}, \mathbf{v} \rangle x$ with scalar inputs and $2m$ parameters as detailed above, the Hessian $\mathbf{H}_\mathrm{L}$ spectrum consists of $m-1$ repeated eigenvalues $\lambda_\mathrm{bulk} = \pm \overline{\delta x}$ and two paired outlying [a] eigenvalues defined by the following expression:
>
> $$\lambda_{\mathrm{outlier}_{1,2}} = \Lambda_{1,2} \left( \|\mathbf{w}\|^2, \|\mathbf{v}\|^2, \langle \mathbf{w}, \mathbf{v} \rangle, \sigma_x^2, \overline{\delta x} \right) = \frac{1}{2} (\sigma_{\mathbf{x}}^2 \|\mathbf{w}\|^2 + \sigma_{\mathbf{x}}^2 \|\mathbf{v}\|^2)$$
> $$\pm \frac{1}{2} \sqrt{ (\sigma_{\mathbf{x}}^2 \|\mathbf{w}\|^2 + \sigma_{\mathbf{x}}^2 \|\mathbf{v}\|^2)^2 + 4(\overline{\delta x}^2 + 2\sigma_{\mathbf{x}}^2 \overline{\delta x} \langle \mathbf{w}, \mathbf{v} \rangle) }. \tag{1}$$
>
> ---
> [a] We term these *outliers* due to their distinct formulation rather than magnitude; indeed, the small outlier can overlap with the bulk spectrum.

The proof, in appendix B.1, relies on computing the Hessian and reducing its characteristic polynomial to a determinant of a rank-one perturbation of a scaled identity, from which its zeros follow.

**An Interpretation of the Top Hessian Eigenvalue, or Sharpness.** Beyond the standard interpretation of the largest magnitude Hessian eigenvalue as the sharpness of the curvature, we show that it can be further understood in terms of the specific underlying contributions. In particular, we reformulate our above expression to the one where the part inside the square-root is a sum of squares. Besides confirming the soundness of this square-root part, this allows for a new interpretation:

$$\lambda_{\mathrm{outlier}_{1,2}} = \frac{1}{2} \left( \sigma_x^2 \|\mathbf{w}\|^2 + \sigma_x^2 \|\mathbf{v}\|^2 \right)$$

$$\pm \frac{1}{2} \sqrt{ \left( \sigma_x^2 \|\mathbf{w}\|^2 - \sigma_x^2 \|\mathbf{v}\|^2 \right)^2 + 4\sigma_{\mathbf{x}}^4 \left( \|\mathbf{w}\|^2 \|\mathbf{v}\|^2 - \langle \mathbf{w}, \mathbf{v} \rangle^2 \right) + 4 \left( 2\langle \mathbf{w}, \mathbf{v} \rangle \sigma_x^2 - \overline{yx} \right)^2 } \tag{2}$$

Hence, this suggests that inherently the *semantics of sharpness lie in a net quantification of:* (1) imbalance between the layer parameter norms, (2) non-collinearity of the parameters, (3) deviation from the target, and alongside (4) overall parameter norm.

Another key observation is that eigenvalues scale in proportion to the input variance $\sigma_{\mathbf{x}}^2$, which is visualized in fig. 7. Additionally, the input variance can be absorbed in with the parameter-related terms and thus underscores how feature scale directly affects the eigenvalues. Hence, this backs the common empirical practice of normalizing data/features[1].

**The Structure of Eigenvectors.** Moving ahead, let us shift our focus on deriving a closed form and understanding the structure of the eigenvectors of the linear network with input and output dimension 1, ordered as per $\boldsymbol{\theta} = (\mathbf{v}^\top, \mathbf{w}^\top)^\top$.

**Theorem 3.2.** For the above setting, the Hessian eigenvectors corresponding to the outlying eigenvalues, determined up to scaling and sign, take the form, for $i \in \{1, 2\}$, given below:

$$\mathbf{z}_{\text{outlier}_i} = \begin{pmatrix} \lambda_{\text{outlier}_i} \mathbf{w} + \overline{\delta x} \, \mathbf{v} \\ \overline{\delta x} \, \mathbf{w} + \lambda_{\text{outlier}_i} \mathbf{v} \end{pmatrix} = \lambda_{\text{outlier}_i} \begin{pmatrix} \mathbf{w} \\ \mathbf{v} \end{pmatrix} + \overline{\delta x} \begin{pmatrix} \mathbf{v} \\ \mathbf{w} \end{pmatrix}$$

Thus, the outlier eigenvectors live in a two-dimensional space spanned by vectors: $(\mathbf{v}^\top, \mathbf{w}^\top)^\top$ and $(\mathbf{w}^\top, \mathbf{v}^\top)^\top$. In fact, they form its orthonormal basis. When the gradient of the loss is non-zero, we can further express these eigenvectors as, $\mathbf{z}_{\text{outlier}_i} = \lambda_{\text{outlier}_i} \overline{\delta x}^{-1} \cdot \nabla_{\boldsymbol{\theta}} \ell + \overline{\delta x} \cdot \boldsymbol{\theta}$. In contrast, the eigenvectors corresponding to the bulk eigenvalues live in a $2m - 2$ dimensional subspace, which is essentially determined by the orthogonal complements of the vectors $\mathbf{w} + \mathbf{v}$, $\mathbf{w} - \mathbf{v}$. More explicitly:

**Theorem 3.3.** For the above setting, the Hessian eigenvectors corresponding to the bulk eigenvalues have the form, determined up to scaling and sign, $\mathbf{z}_{\text{bulk}} = \begin{pmatrix} \hat{\mathbf{z}}_{\text{bulk}}^\top & \text{sgn}(\lambda_{\text{bulk}}) \hat{\mathbf{z}}_{\text{bulk}}^\top \end{pmatrix}^\top$

with $\hat{\mathbf{z}}_{\text{bulk}} = \left( \mathbf{I} - \dfrac{(\mathbf{w} + \text{sgn}(\lambda_{\text{bulk}}) \mathbf{v})(\mathbf{w} + \text{sgn}(\lambda_{\text{bulk}}) \mathbf{v})^\top}{\|\mathbf{w} + \text{sgn}(\lambda_{\text{bulk}}) \mathbf{v}\|^2} \right) \mathbf{c}$, for some vector $\mathbf{c}$.

Note that, the expression of the eigenvectors above also suggests that they would change smoothly over the course of training, unless there are rapid changes in the subspace spanned by the parameters.

**The 2-Dimensional Subspace of Gradient Descent.** As a matter of fact, we can prove that the space spanned by the bulk eigenvectors does not evolve when training with gradient descent and that all the gradient descent steps as well as the initialization, and hence all the iterates, are orthogonal to that bulk eigenspace as stated in corollary 3.1.

**Corollary 3.1.** Assume that we initialize the parameters as $\boldsymbol{\theta}_0 = (\mathbf{v}_0^\top, \mathbf{w}_0^\top)^\top$, and run gradient descent with step $\Delta \boldsymbol{\theta}_t = -\eta_t \overline{\delta x}_t (\mathbf{w}_t^\top, \mathbf{v}_t^\top)^\top$ to go from $\boldsymbol{\theta}_t$ to $\boldsymbol{\theta}_{t+1}$. Then the bulk eigenspace does not change over training $\hat{\mathbf{z}}_{\text{bulk}} = \hat{\mathbf{z}}_{\text{bulk}}^{(t)} = \hat{\mathbf{z}}_{\text{bulk}}^{(0)}$, and $\langle \boldsymbol{\theta}_t, \hat{\mathbf{z}}_{\text{bulk}} \rangle = 0$, $\forall t \in [0, T]$.

Hence, gradient descent occurs entirely within a 2-dimensional subspace, spanned by the outlier eigenvectors. Previously, prior work (Gur-Ari et al., 2018) had *empirically* suggested the gradient descent tends to happen in a tiny subspace; and here we show a clear and specific instance where this holds theoretically. Importantly, our results do not contradict the recent work of Song et al. (2025), who show that the 'bulk subspace' is crucial for training, since their definition of 'bulk subspace' includes not only the span of bulk Hessian eigenvectors (as in corollary 3.1) but also the eigenvector of the smaller outlier $\lambda_{\text{outlier}_2}$, which may be hidden within the bulk.

## 3.2 ReLU Network

Assume the network is now, $f_{\mathbf{w}, \mathbf{v}}(x) = \mathbf{w}^\top (\mathbf{v} \cdot x)_+$, where, $(a)_+ = \mathbf{1}\{a > 0\}$ is the ReLU non-linearity and is applied elementwise. For this particular network, a hidden neuron $j$ 'fires' if

---

[1]A standard element of the ML pipeline is to normalize the data. Besides, Ghorbani et al. (2019) note batch normalization (BN), which acts as a normalization within the network, can suppress outlier eigenvalues.

$v_j x > 0$, i.e., when either $v_j > 0$, $x > 0$ and $v_j < 0$, $x < 0$. Hence, this parameter space partitions in tandem with the partitions of the input space, the latter will be referred to as cells, which are namely, the $+$ cell where $x > 0$ and the $-$ cell where $x \leq 0$.

Notice, we can write $\mathbf{v} = \mathbf{v} \odot \mathbb{1}\{\mathbf{v} > 0\} + \mathbf{v} \odot \mathbb{1}\{\mathbf{v} \leq 0\} =: \mathbf{v}_+ + \mathbf{v}_-$, where $\odot$ denotes the Hadamard product and $\mathbb{1}\{\mathbf{a} > 0\}_j = \mathbb{1}\{a_j > 0\}$. Likewise, we can associate the $\mathbf{w}$ parameters to these two cells as, $\mathbf{w} = \mathbf{w}_+ + \mathbf{w}_-$, with $\mathbf{w}_+ = \mathbf{w} \odot \mathbb{1}\{\mathbf{v} > 0\}$ and $\mathbf{w}_- = \mathbf{w} \odot \mathbb{1}\{\mathbf{v} \leq 0\}$. Thus, we can express the network function in the following manner,

$$f_{\mathbf{w},\mathbf{v}}(x) = \langle \mathbf{w}_+, \mathbf{v}_+ \rangle x \, \mathbb{1}\{x > 0\} + \langle \mathbf{w}_-, \mathbf{v}_- \rangle x \, \mathbb{1}\{x \leq 0\}. \tag{3}$$

We specialize the previously defined shorthands for various data-dependent quantities to each of the two cells. In particular, let us define the (uncentered) standard deviation of the positive and negative datapoints as, $\sigma_{x+} = \sqrt{\frac{1}{n_+} \sum_{i=1}^{n} x_i^2 \mathbb{1}\{x_i > 0\}}$ and $\sigma_{x-} = \sqrt{\frac{1}{n_-} \sum_{i=1}^{n} x_i^2 \mathbb{1}\{x_i < 0\}}$ respectively, and besides, the total number of points can split as $n = n_+ + n_-$. Also, let us denote the (uncentered) input-output covariance for the positive and negative cells respectively as follows, $\overline{yx}_+ = \frac{1}{n_+} \sum_{i=1}^{n} y_i x_i \mathbb{1}\{x_i > 0\}$, and $\overline{yx}_- = \frac{1}{n_-} \sum_{i=1}^{n} y_i x_i \mathbb{1}\{x_i \leq 0\}$. In this setting, the parameter space can be nicely partitioned in such a way that the Hessian can be decoupled between the positive and the negative cells. Let us additionally assume that $q$ coordinates of the parameter vector $\mathbf{v}$ are positive. Then we have that the spectrum in this ReLU case follows from the discussed Hessian decoupling and the eigenvalues in the linear case.

**Theorem 3.4.** For the above ReLU setting, the bulk Hessian spectrum consists of $q - 1$ and $m - q - 1$ repeated eigenvalues in signed pairs, $\lambda_{\text{bulk}}^+ = \pm \frac{n_+}{n} \overline{x\delta}_+$, $\lambda_{\text{bulk}}^- = \pm \frac{n_-}{n} \overline{x\delta}_-$ and with the outlying eigenvalues being $\lambda_{\text{outlier}_{1,2}}^+ = \frac{n_+}{n} \Lambda_{1,2}\left(\|\mathbf{w}_+\|^2, \|\mathbf{v}_+\|^2, \langle \mathbf{w}_+, \mathbf{v}_+ \rangle, \sigma_{x+}^2, \overline{\delta x}_+\right)$ and $\lambda_{\text{outlier}_{1,2}}^- = \frac{n_-}{n} \Lambda_{1,2}\left(\|\mathbf{w}_-\|^2, \|\mathbf{v}_-\|^2, \langle \mathbf{w}_-, \mathbf{v}_- \rangle, \sigma_{x-}^2, \overline{\delta x}_-\right)$.

We see that just as in the linear case, we obtain a set of paired outlier eigenvalues having a similar functional form through $\Lambda_{1,2}$, but with a dependence on respective cell-wise quantities. Similarly, the eigenvectors from the linear case also carry over to the ReLU case here. Figure 8 in the appendix highlights the paired nature of the outlying eigenvalues, for both linear and ReLU, throug training.

**Semantics of Sharpness/Top Eigenvalue in the Non-Linear Case.** Similarly, we can reformulate the above expressions into more semantic terms as done before in eq. (2), but as defined at the level of cells. However, an additional aspect to note is that in the ReLU case, the maximum eigenvalue $\lambda_{\max} = \max(\lambda_{\text{outlier}_1}^+, \lambda_{\text{outlier}_1}^-)$. *Thus, the cell of the largest curvature dictates the maximum eigenvalue.*

**Impact of ReLU on the Hessian Spectrum.** Firstly, we notice that the eigenvalues scale in proportion to the density ($n_+/n$ or $n_-/n$) of the corresponding cell, i.e., cells which are active for a small number of samples will be less prominent in the spectrum, and vice-versa. This also suggests a natural principle for the occurrence of numerous spuriously tiny eigenvalues with ReLU, as opposed to the linear case where there is a much clearer demarcation between

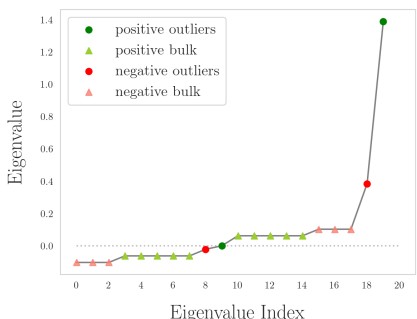

Figure 1: Hessian spectrum for a ReLU network with width $m = 10$. The eigenvalues are coloured based on their source cell, i.e., positive or negative. The dotted gray line demarcates the negative & positive eigenvalues.

zero and non-zero eigenvalues (Singh et al., 2021). Lastly, while the above setup involves only two mutually exclusive partitions of the input space, in the general case we can expect overlaps and cross-terms. It would form an interesting question for future work to see how well the independent cell-wise Hessian serves as an approximation. Overall, this cell-wise Hessian decomposition hints at how the spectrum with a non-linearity like ReLU might be structured beyond that of a linear network.

### 3.3 SELF-ATTENTION

Thanks to the tremendous success of large language models, Transformers are nowadays the most widely used architecture (Vaswani et al., 2017; Radford et al., 2019; Dosovitskiy et al., 2021). A key

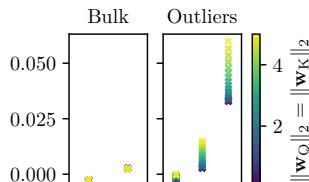 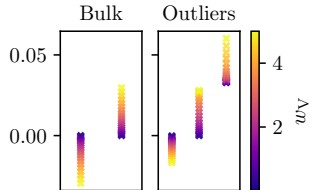 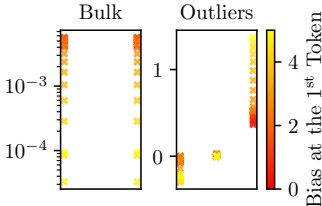

(a) Larger norm of query and key parameters implies larger magnitude of the outlier eigenvalues.

(b) Larger magnitude of the value parameter implies larger magnitude of the whole spectrum.

(c) Larger bias at the first token (an attention sink) shrinks the bulk and grows the outliers.

Figure 2: Influence of model parameter norms and presence of the attention sinks on the spectrum in the simple self-attention model (see appendix D.1.1 for details).

component distinguishing the Transformer architecture from the other models is the self-attention layer. Although the self-attention is more non-linear than an MLP and the loss Hessian of a model involving self-attention becomes highly involved (Ormaniec et al., 2024), our proof technique can still be applied to obtain the spectrum.

Specifically, let us consider a model consisting of a single-head self-attention layer (Bahdanau, 2014; Vaswani et al., 2017) with embedding dimension 1 and sequence length $n$ as in eq. (4)[2].

$$\boldsymbol{f}_{w_\text{V},\mathbf{w}_\text{Q},\mathbf{w}_\text{K}}(\mathbf{x}) := w_\text{V}\mathbf{A}(\mathbf{x})\mathbf{x} \quad \text{where} \quad \mathbf{A}(\mathbf{x}) = \text{softmax}\left(\frac{1}{\sqrt{d_K}}\mathbf{x}\mathbf{w}_\text{Q}^\top\mathbf{w}_\text{K}\mathbf{x}^\top\right), \qquad (4)$$

where $\mathbf{x} \in \mathbb{R}^n$ is a single sequence, $\mathbf{w}_\text{Q}, \mathbf{w}_\text{K} \in \mathbb{R}^{d_K}$ are query and key weight vectors, and $w_\text{V} \in \mathbb{R}$ is a value parameter. Moreover, using the same definition of self-attention moment vectors as in Ormaniec et al. (2024) (definition B.1) let us define some helper terms, specifically $\alpha := \|\mathbf{m}_1\|^2/n$, $\beta := \frac{1}{n\sqrt{d_K}}((w_\text{V}\mathbf{m}_1 + \delta_{\mathbf{x},\mathbf{y}})^\top(\mathbf{x} \odot \mathbf{m}_2))$, $\gamma := \frac{1}{n\sqrt{d_K}}(w_\text{V}\delta_{\mathbf{x},\mathbf{y}}^\top(\mathbf{x} \odot \mathbf{m}_2))$, $\zeta := \frac{1}{nd_K}(w_\text{V}^2\|\mathbf{x} \odot \mathbf{m}_2\|^2 + w_\text{V}\delta_{\mathbf{x},\mathbf{y}}^\top(\mathbf{x} \odot \mathbf{x} \odot \mathbf{m}_3))$, where $\odot$ stands for Hadamard product.

**Theorem 3.5.** For the above setting, the Hessian has $d_K - 1$ pairs of bulk eigenvalues $\pm\gamma$. Moreover, there are three outlier eigenvalues $\lambda_{\text{outlier}_{1,2,3}}$, that satisfy

$$\frac{1}{3}(\alpha + \zeta\|\mathbf{w}_\text{K}\|^2 + \zeta\|\mathbf{w}_\text{Q}\|^2) - \xi \leq \lambda_{\text{outlier}_{1,2,3}} \leq \frac{1}{3}(\alpha + \zeta\|\mathbf{w}_\text{K}\|^2 + \zeta\|\mathbf{w}_\text{Q}\|^2) + \xi, \text{ where}$$

$$\xi = \frac{2}{3}\sqrt{3\gamma^2 + 6\zeta\gamma\langle\mathbf{w}_\text{K},\mathbf{w}_\text{Q}\rangle + 3(\beta^2 - \alpha\zeta)(\|\mathbf{w}_\text{K}\|^2 + \|\mathbf{w}_\text{Q}\|^2) + (\zeta\|\mathbf{w}_\text{K}\| + \zeta\|\mathbf{w}_\text{Q}\| + \alpha)^2}.$$

Proof of the above theorem can be found in appendix B.5. We note that the same proof technique can be used to derive the Hessian spectrum for a self-attention layer with any twice-differentiable activation instead of softmax, as long as the activation is applied separately per each row.

**Large Magnitude of the Query and Key Weights Implies a Badly-Conditioned Hessian.** Note that on one hand the bulk eigenvalues $\pm\gamma$ depend on $\mathbf{w}_\text{K}$ and $\mathbf{w}_\text{Q}$ only through the attention scores in $\mathbf{m}_2$. These in turn at initialization converge almost surely with growing $d_K$ to a uniform distribution over tokens as shown by Noci et al. (2022). We demonstrate this in fig. 2a where the change in the norm of the query and key weight vector does not affect the bulk eigenvalues. Since the positive bulk upper bounds the absolute smallest eigenvalue $|\lambda_{\min}|$, the above reasoning implies that $|\lambda_{\min}|$ is upper bounded by a value approximately constant in the magnitude of the entries of $\mathbf{w}_\text{K}$ and $\mathbf{w}_\text{Q}$. On the other hand, as predicted by theorem 3.5 the outliers, so also the largest magnitude eigenvalue $\lambda_{\max}$ explicitly depends on the magnitude of the query and key weights (again see fig. 2a). In fact, we can precisely state a lower bound on the Hessian condition number as stated in remark 3.1.

**Remark 3.1.** The Hessian condition number $\kappa(\mathbf{H}_\text{L}) = \frac{|\lambda_{\max}|}{|\lambda_{\min}|} \geq \frac{2(\alpha + \zeta(\|\mathbf{w}_\text{K}\|^2 + \|\mathbf{w}_\text{Q}\|^2)) + 3\xi}{6|\gamma|}$.

The lower bound implies that the condition number is monotonic in magnitude of the key and query weights. This claim is specific to queries and keys, and we do not expect such a relationship for the magnitude of $w_\text{V}$ as both the bulk and outlier eigenvalues depend on $w_\text{V}$ (see fig. 2b).

---

[2]Instead of a batch of scalars as in the previous sections, here we consider a sequence of scalars. Since self-attention does not work separately on every element of the sequence, we group $x_i$ into a vector $\mathbf{x}$ .

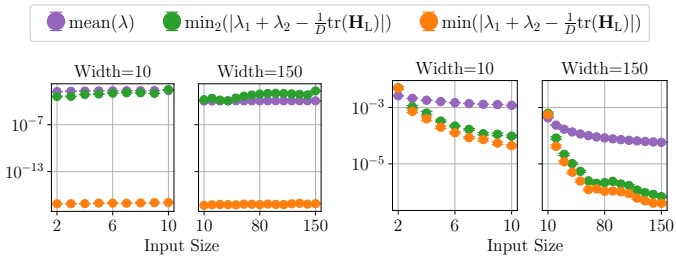

(a) Orthogonal parametrization and decorrelated data.

(b) Standard parametrization and standardized data.

Figure 3: Smallest differences between the paired eigenvalues and $\frac{1}{D} \operatorname{tr}(\mathbf{H}_\mathrm{L})$ compared to the mean eigenvalue across varying input sizes. For orthogonal parametrization and decorrelated data, there are two clearly paired outliers. Beyond these assumptions, the hypothesis holds approximately.

**Massive Activations and Attention Sinks Increase the Sharpness and Shrink the Bulk.** Sun et al. (2024); Gu et al. (2025) observed that in many large language models the activations for certain tokens, frequently the first one, are significantly larger than for others (massive activations) and that these tokens later on have significantly higher attention scores associated with them (attention sinks). Our eigenvalue expressions allow for reasoning, how emergence of this phenomenon influences the loss landscape. Note that as an attention sink grows, an increasing probability mass (as defined by the attention scores) is allocated to a token with a massive activation, which in turn raises the magnitude of the first attention moment $\mathbf{m}_1$. In contrast, as the distribution concentrates on a single token, the central moments like $\mathbf{m}_2$ and $\mathbf{m}_3$ shrink towards zero. These imply that the bulk shrinks towards zero and, assuming that the massive activation $x_i$ is large enough, the largest eigenvalue grows, resulting in a sharper, badly conditioned landscape. Figures 2c and 9, where we simulate attention sinks by adding a bias term on the first token inside the self-attention, illustrate this phenomenon.

**Comparison With the Linear Network Spectrum.** Due to the softmax nonlinearity, the formulae needed to define the spectrum are more complicated compared to the ones of the linear and ReLU networks. Note that all terms dependent on the residuals $\delta_i = f_i - y_i$ in the formulae come from the functional Hessian[3]. Through this we infer that most of the terms in the largest Hessian eigenvalue bounds from remark B.4 are dictated by the functional Hessian, which is not true for linear and ReLU networks. Therefore, we hypothesize that approximating the largest eigenvalue of a Hessian by the largest eigenvalue of the GGN may be less accurate in the case of Transformers than in the case of MLP-based architectures. In section 4.3 we empirically study this hypothesis in a full Transformer model.

## 4 WIDENING THE APPLICABILITY OF KEY RESULTS

In this section, we empirically verify how much the closed-form Hessian expressions for the eigenvalues and ensuing insights extend beyond our theoretical assumptions.

First, it turns out that, under certain assumptions on the weights and data, our proof technique can also be applied to derive the Hessian spectrum in closed form of a linear network with a bias in the first layer, so $f_{\mathbf{w},\mathbf{v},\mathbf{b}}(x) = \mathbf{w}^\top(\mathbf{v}x + \mathbf{b})$ (see appendix B.3). In this case we again obtain a set of symmetric bulk eigenvalues and five outlier eigenvalues. Two of these outliers are centered at $1/4 \cdot \operatorname{tr}(\mathbf{H}_\mathrm{L})$, while another three around $1/6 \cdot \operatorname{tr}(\mathbf{H}_\mathrm{L})$. Then, again under some assumptions on the weight matrix and data structure, we can also directly extend our linear network results to larger input or output sizes, so models of a form $f_{\mathbf{w},\mathbf{V}}(\mathbf{x}) = \mathbf{w}\mathbf{V}\mathbf{x}$ or $f_{\mathbf{W},\mathbf{v}}(\mathbf{x}) = \mathbf{W}\mathbf{v}\mathbf{x}$, with block diagonal $\mathbf{W}$ and $\mathbf{V}$ (see appendix B.4).

### 4.1 PAIRED OUTLIER EIGENVALUES

In section 3.1, we learned that for input size $D = 1$, two outlier eigenvalues sum up to the Hessian trace. Together with the extension to bias case, which is a specific instance of a network with input size 2, as well as general $D$ but with a specific structure in the weight matrix, we hypothesize that for a two-layer linear network with input size $D$ and output size 1, there exist two outlier eigenvalues that sum to $1/D \operatorname{tr}(\mathbf{H})$. To verify this hypothesis, for a fixed width, we vary the input size $D$, and for every model, we numerically compute all the Hessian eigenvalues without approximation. Then

---

[3]We adopt the functional Hessian terminology from (Singh et al., 2021; 2023; Ormaniec et al., 2024). The functional Hessian is simply a difference between the Hessian and GGN.

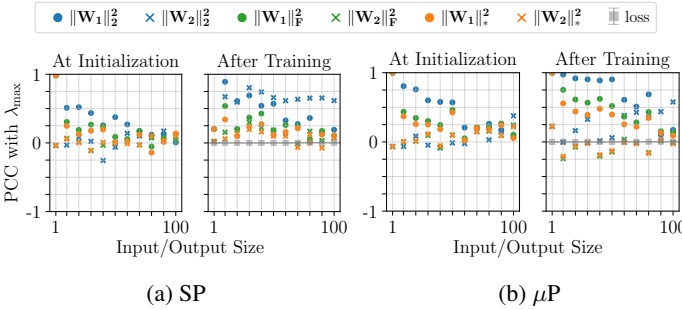

Figure 4: Pearson correlation between the sharpness and the squared norm (spectral, Frobenius, nuclear) of the weight matrices for a two-layer network of width 100 across different input/output sizes. **Sharpness correlates most with (squared) spectral norm of weight matrix.**

we look for a pair of eigenvalues that sum up to a value closest to $1/D \operatorname{tr}(\mathbf{H}_L)$. Specifically, we compare the quantity $\min(|\lambda_1 + \lambda_2 - 1/D \operatorname{tr}(\mathbf{H}_L)|)$ against the next best fit of two eigenvalues $\min_2(|\lambda_1 + \lambda_2 - 1/D \operatorname{tr}(\mathbf{H}_L)|)$, and against the mean eigenvalue for reference.

Figure 3a shows these quantities under the same assumptions that were used to derive the closed-form eigenvalue expressions for the two-layer neural network with bias, such as orthogonal initialization of the first weight matrix. We find that, at any input size, the sum of the two eigenvalues is close to the considered fraction of the trace. Their absolute difference is orders of magnitude smaller than the second closest sum of eigenvalue pair, as well as the reference mean eigenvalue. This suggests that, indeed, there exists an outlier eigenvalue pair which sum up very close to $1/d \operatorname{tr}(\mathbf{H})$. Figures 3b, 13b and 13c show the case beyond the orthonormality assumption. Despite that, the sum of the two eigenvalues is very close to the trace fraction, but there is no clear best pair, as the pair that induces second-best fit to the considered fraction of trace is also close to the smallest one.

## 4.2 SHARPNESS CORRELATES WITH SPECTRAL NORM IN LINEAR NETWORKS

In this section, we examine how the largest Hessian eigenvalue, or sharpness, as obtained in formula eq. (1), extends beyond the two-layer, unidimensional setting from before. Recall that, according to eq. (1), sharpness should depend on the norms and alignment of weighs. We now focus on deep linear networks $\boldsymbol{f}_{\{\mathbf{W}_b\}} = \mathbf{W}_B \cdots \mathbf{W}_1 \mathbf{x}$, with $\mathbf{W}_b$ shaped to allow arbitrary input and output sizes. We consider weight-matrix-dependent quantities that reduce to vector norms and inner-products (alignments) in the two-layer case with scalar input and output, and study their correlation with the largest Hessian eigenvalue (see appendix D.3.3 for details).

**Sharpness Correlates With the Spectral Norm of Weight Matrices in Two-Layer Networks.** We first focus on a two-layer, arbitrary input/output size network, and consider Frobenius, spectral, and nuclear norms, as in the vector case they all amount to the $\ell_2$ norm we see in eq. (1). In fig. 4, under standard and maximum-update parametrization (SP and $\mu$P respectively), we demonstrate that it is the spectral norm of the weight matrix that correlates the most with the largest Hessian eigenvalue. This effect is most prominent in trained networks, where the Hessian is well-approximated by the GGN. While Zhao et al. (2024) establish that the GGN condition number in linear MLPs is controlled by the singular values of the weight matrices, our results connect these singular values directly to the Hessian eigenvalues themselves. In appendix D.3.3 we provide a preliminary formal analysis of the relationship between the largest Hessian eigenvalue and the squared spectral norm of weight matrices in two layer linear networks with arbitrary input/output dimension. Finally, we note that the specific weight matrix whose spectral norm drives the correlation depends on initialization (see appendix D.3.3 for results for neural tangent parametrisation and further discussion).

**Sharpness Correlates With the Spectral Norm of Weight Matrix Products in Deep Networks.** Having honed in on the (squared) spectral norm, for depth-$B$ linear networks, we examine correlations with sharpness considering both individual weight matrices $\mathbf{W}_i$ and of "knocked-out" products $\widehat{\mathbf{W}}_i = \prod_{j \neq i} \mathbf{W}_j$. Note that in the case of a two-layer network, both of these formulations happen to be mathematically equivalent. The knocked-out form is inspired by the gradient structure, since the derivative with respect to $\mathbf{W}_i$ naturally involves the product of all other matrices. Among these quantities, the squared spectral norm of $\widehat{\mathbf{W}}_B$ shows the strongest correlation with sharpness, especially near convergence and in low-dimensional settings, with the effect partially persisting for larger input/output size under $\mu$P (figs. 17 and 18 in the Appendix). This is a promising sign, and we aim to study other potential quantities in future work.

In appendix D.3.3, we conduct similar experiments for ReLU networks. We find that although the squared spectral norm shows non-negligible correlation with the largest Hessian eigenvalue (between 0.25 and 0.75 for the first weight matrix), the squared Frobenius norm correlates most strongly.

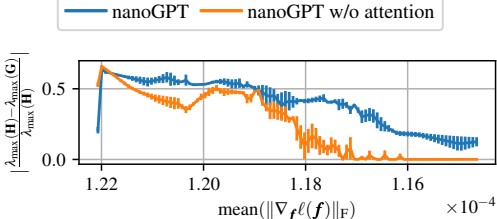

Figure 5: Relative approx. error of sharpness by the largest GGN eigenvalue against mean residual norm, while pretraining GPT2 and similarly-sized model without attention. For almost all values of the mean residual norm, the presence of attention makes for a less accurate approximation.

### 4.3 INSIGHTS
### ON THE TRANSFORMER MODELS

Moving on from MLPs, here we investigate how our observations about the single-head self-attention model extend to a full GPT2-like (Karpathy, 2022) Transformer model on a language modeling task, when trained from scratch on OpenWebText and with 124M parameters. Specifically, our interpretation of the Hessian eigenvalue expressions in section 3.3 suggests that the GGN provides a worse approximation in Transformers than in MLPs, and implies a connection between attention sinks and loss Hessian eigenvalues. We present empirical evidence below that supports these hypotheses, confirming distinct spectral behaviors in deep Transformers compared to simple MLPs.

**GGN Top Eigenvalue Approximates Sharpness Worse in Transformers Than in MLPs.** In section 3.3, based on the closed-form outlier Hessian eigenvalue for a model consisting of a single self-attention layer, we hypothesized that the largest eigenvalue of the GGN might be a worse approximation of the largest Hessian eigenvalue in models with self-attention than in models without it. To verify this hypothesis, we compute the relative approximation error across the first 10000 pretraining steps and compare it with a reference 'attention-less' GPT2 model, where we substitute MLP blocks in place of the attention layers while having a similar number of parameters. In fig. 5, we plot[4] this relative approximation error against the loss residual (i.e., the gradient of the loss with respect to the model output and which technically drives the difference between Hessian and GGN to zero during training). We see that for almost every fixed value of the loss and residual (except at the initialization) the relative approximation error for the model without attention is smaller than for a model with attention, confirming our hypothesis. The intuitive explanation is as follows: Since the GGN is the loss Hessian of a linearized model (Grosse, 2022; Dauphin et al., 2024), we expect the approximation error to be positively correlated with the curvature of the network's activation functions. Specifically, activations with non-trivial second derivatives, such as softmax, are expected to induce a larger approximation gap than the piecewise linear activations common in MLPs.

**Attention Sinks Amplify Outlier Eigenvalues.** Finally, we also test our hypothesis that presence of the attention sink promotes larger outlier eigenvalues. To do that, we enforce an attention sink at the first token of the first head at every Transformer block in the GPT2 model, and compute the whole Hessian spectrum. In fig. 6 we demonstrate the spectra together with the average attention scores of the first token in the edited attention heads. We see that more prominent the attention sinks in the model, the larger the outliers in the spectrum (while still preserving many eigenvalues close to zero).

## 5 CONCLUSION AND DISCUSSION

**Summary.** In this work, we derived the first closed-form formulae for the Hessian spectrum of a set of non-trivial yet fundamental neural networks. Our results concern the full Hessian—not an approximation—and hold at arbitrary points in training. We interpreted the resulting eigenvalue and eigenvector expressions, verified their validity empirically, and showed that several observations (such as the paired nature of outlying eigenvalues, dependence on weight matrix norms, weaker approximation strengths of the GGN in Transformers than in MLPs, and the influence of attention sinks) extend beyond our assumptions, including to large-scale settings such as GPT-2.

---

[4]We present a similar plot against the loss in fig. 24 of the appendix.

**Limitations and Future Work.** While our theoretical results are, to our knowledge, the first successful derivation of closed-form Hessian spectra for non-trivial neural networks, the models we analyze in closed form remain comparatively simple. This choice reflects our focus on overcoming a major structural hurdle. Our proof technique relies on the specific Hessian structure of these models. By leveraging tools from linear algebra (such as Schur Complements), we reduce the characteristic matrix

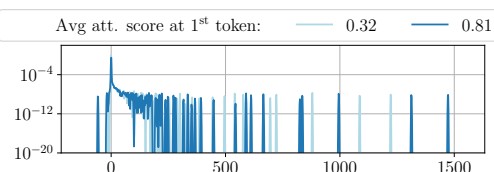

Figure 6: Hessian spectra in the GPT2 model for different magnitudes of attention sinks. Larger attention sinks stretch the spectrum.

of the Hessian to a matrix with a more tractable determinant and then solve the resulting system of equations. We note that, without introducing additional assumptions, for arbitrarily deep neural networks even though such a procedure does simplify the determinant calculations, the resulting equation does not necessarily have closed-form solutions (but only implicit solutions). This, however, indicates directions for further work. On the theoretical side, one could explore whether with stronger or alternative assumptions, further families of networks admit exact closed-form spectra. On the empirical side, it would be valuable to identify additional analogues that correlate with sharpness in more complex architectures, thereby further testing the scope of our theoretical insights. Such developments would extend the reach of the present framework without diminishing its rigor. Besides, this focus on deriving exact eigenvalue and eigenvector formulae necessarily left less room for analyzing training dynamics. Nevertheless, the expressions of the spectra already suggest a natural categorization of the phases in learning — early, late-stage, and divergent —which we outline in appendix C. A full-fledged and detailed study of these dynamical aspects is an important direction for future work.

### Usefulness of Simple Models

The simple models presented in this manuscript can be viewed as a solvable baseline: when a researcher observes a Hessian spectral phenomenon in a deep network that also exists in our simple models, the derived expressions provide mechanistic, precise insight into the phenomenon's fundamental drivers. To present an example of this, we have verified that progressive sharpening and edge of stability (Cohen et al., 2020) are present in the self-attention layer we study in section 3.3 (see appendix D.2). We believe our results could be particularly useful for studying the effects of initialization, width, and data characteristics on the neural network landscape, as well as the role of softmax/activations in Transformers. These concepts are explicitly captured by the models considered in this manuscript.

To conclude, we hope that the closed-form Hessian eigenvalues and eigenvectors derived here will unlock the possibility to study eigenvalue-related phenomena rigorously in closed form and can inform design of new numerical algorithms for approximating the Hessian.

## REPRODUCIBILITY STATEMENT

We attach proofs of all theorems presented in this manuscript in appendix B. We also provide all experimental details in appendix D.1.

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

CONTENTS

## A  EXTENDED RELATED WORK

**Significance of Neural Network Loss Hessian and Its Spectrum.** The Hessian of neural network loss and its eigenvalues have for a long time been of interest for the machine learning community. The reason for this is their relevance for loss landscape analysis, optimization, model generalization, and utilization of the Hessian in areas such as pruning and quantization.

*Generalization and Flat Minima Discussion.*  There is a significant body of work in deep learning connecting loss Hessian properties at local minimum and the generatlization properties of the model. Observations about small batch training resulting in both better generalizing models and flatter minima, meaning with smaller largest eigenvalue (sharpness) of the loss Hessian, resulted in a hypothesis that models of smaller sharpness should generalize better (Hochreiter & Schmidhuber, 1997; Keskar et al., 2016; Jastrzębski et al., 2017; Chaudhari et al., 2019). Since then this hypothesis has been disputed (Dinh et al., 2017; Andriushchenko et al., 2023), but it gave rise to a family of sharpness-aware optimization techniques (Foret et al., 2021; Kwon et al., 2021; Andriushchenko & Flammarion, 2022).

*Optimization and Architecture Analysis.*  Loss Hessian and its eigenvalues have also been used to reason about architecture design and optimization. For both convex problems and deep learning models, the learning rate $\eta$ used by gradient-based methods and the largest loss Hessian eigenvalue are intrinsically related. While for convex problems the choice of the most optimal learning rate is clear from theory, deep learning optimization displays more entangled phenomena. The largest eigenvalue of the Hessian during optimization seems to first increase and then stabilize around the $2/\eta$ threshold (Jastrzębski et al., 2018). The first phase of this phenomenon has been coined *progressive sharpening*, and the stabilization phase *the edge of stability* (Cohen et al., 2020). Another interesting observation is that while training with a learning rate larger than the stability threshold a catapult mechanism occurs in the loss, meaning that the loss explodes but then goes back to converging (Lewkowycz et al., 2020). Recently, the largest Hessian eigenvalue has also been used to understand the hyperparameter transfer phenomenon occuring under $\mu P$ parameterization

(Noci et al., 2024). Gur-Ari et al. (2018) empirically showed that stochastic gradient descent often moves along the Hessian eigenvectors corresponding to the largest eigenvalues, a phenomenon that is preserved through training. The influence of the batch normalization and residual connections on optimization has been studied through an empirical eigenvalue analysis Ghorbani et al. (2019); Yao et al. (2020), resulting in a claim that they make the optimization easier by smoothing the loss landscape. Finally, recently the block Hessian spectra have been used to link Transformer Hessian heterogeneity to the superior performance of adaptive optimizers over SGD in optimization of Transformer models (Zhang et al., 2024a) and inspired new-optimizer design that saves memory and accelerates training compared to widespread AdamW (Zhang et al., 2024b; Wang et al., 2025).

*Applications.* As evident from the above discussion, the loss Hessian contains useful signals for optimization. Various of its approximations have been used as pre-conditioners (Martens, 2010; Martens & Grosse, 2015a; Grosse & Martens, 2016; Gupta et al., 2018). Moreover, pruning and quantization techniques can be framed as finding the subset of parameters or numerical approximation of the model that best fits a Taylor approximation of the pruned/quantized model. At local minimum, this approximately boils down to finding the model that matches the largest Hessian eigenvalues. LeCun et al. (1989) introduced Optimal Brain Damage, a pruning framework that uses a diagonal approximation of the Hessian through its eigenvalues, to estimate each weight's contribution to the loss, allowing low-saliency weights to be removed with minimal performance degradation. Hassibi et al. (1993) extended this idea with Optimal Brain Surgeon, leveraging the full Hessian to account for weight interactions and more accurately guide pruning decisions. Frantar et al. (2022) demonstrated how the optimal brain damage framework can efficently be applied to post-training network quantization. Hessian and its eigenvalues are also relevant for continual learning. Mirzadeh et al. (2020) studied the effect of dropout, learning rate decay, and batch size, on preventing catastrophic forgetting through forming training regimes that widen the tasks' local minima, as measured with Hessian large eigenvalues. Mirzadeh et al. (2021) used loss Hessian eigenvalues to study the linear connectivity between continual learning and multitask learning solutions.

**Characterising Hessian Spectra.** Given the relevant insights and applications of the Hessian and its spectra described in the paragraph above, it is not surprising that studying and characterising the spectra of the Hessian became a new branch of research in itself. Already LeCun et al. (2002) measured the loss Hessian spectra to draw insights about optimization, that is, they claimed that the largest eigenvalues can cause problems for optimization. Sagun et al. (2016; 2017) empirically characterized the spectrum of small neural networks, by splitting it to so-called bulk eigenvalues that are grouped around zero and characterize how overparametrized the system is and so-called outlier eigenvalues which are bigger and depend on the input data. Importantly, they also found that even after training, when the loss does not improve, a significant number of negative bulk eigenvalues still remains. Papyan (2018) verified that the bulk and outlier eigenvalue structure is preserved in larger networks and attributed the outlier eigenvalues to the positive-definite summand of the Gauss-Newton decomposition of the Hessian (GGN) and the bulk eigenvalues to the remaining part. In the classification setting, Papyan (2019; 2020) analyzed the Hessian spectrum and revealed additional structures in both the bulk and the outliers, arising from class splits in the data and from interactions among gradients of same-class versus different-class samples. Singh et al. (2021; 2023) provided tight bounds on the rank of the MLP and CNN Hessian, indirectly characterizing the number of its zero eigenvalues.

There is also a line of work that studies the Hessian and its spectrum using tools from random matrix theory. Choromanska et al. (2015) model ReLU neural networks as random polynomials in a spin-glass framework, and study the distribution of critical points and their indices (i.e., the number of negative Hessian eigenvalues). d'Ascoli et al. (2022) uses the approximate spectral density of the loss Hessian in the limit of infinite input size to characterize local curvature, confirming that in non-convex settings, gradient descent dynamics with learning rate decay never completely escape saddle points. Similarly, Sarao Mannelli et al. (2020) find the formulas for two smallest eigenvalues of an approximate Hessian in the phase-retrieval task with a single-layer linear network, assuming an infinite input dimension. Pennington & Bahri (2017) derive an approximation of the Hessian spectrum of a two-layer linear network in the infinite-width and large-sample limit, showing that the shape of the spectrum depends strongly on the value of the loss and the ratio of parameters to data points. Granziol et al. (2022) examine the spectrum of the stochastic (mini-batch) Hessian, particularly its largest eigenvalue, and use this to derive analytical expressions for how the maximum

learning rate should scale with batch size. In contrast to the formulas presented in this manuscript, the Hessian eigenvalue formulas and densities discussed above require approximating the Hessian with a random matrix, are probabilistic in nature, and hold exactly only in the asymptotic limit.

**Analysis of Simplified Models.** While empirical analysis of the Hessian spectra is possible to some extent thanks to Hessian-vector products (Pearlmutter, 1994) and tools from randomized linear algebra (Papyan, 2020; Dangel et al., 2025), theoretical analyses are restricted to simple models, which nevertheless can capture the empirical phenomena and provide insights for their explanation. Lewkowycz et al. (2020) theoretically characterize three learning rate phases (lazy, catapult, and divergent) by analizing the training loss and the largest eigenvalue of the NTK of a 2 layer linear network with MSE loss. They start with even a simpler setting (input dimension 1 and equal to 1) which captures the most important aspects of the full solution. Ahn et al. (2022) conducted an empirical study of gradient descent's unstable convergence, characterizing the behavior of loss and largest Hessian eigenvalue, and supported their results by theoretical analysis across quadratic objectives and simple scalar neural networks to explain the observed dynamics. Zhu et al. (2023) derived the loss Hessian eigenvlaues of a 4-scalar linear network and studied the edge of stability phenomenon in this setting. Wang et al. (2022) theoretically characterized the progressive sharpening and edge of stability phenomena by analyzing the largest NTK eigenvalue in two-layer linear networks. Similarly, Noci et al. (2024) studied the largest NTK eigenvalue under $\mu$P, to show that the largest NTK eigenvalue is not dependent on the network width. Agarwala et al. (2023) proved that the second-order regression model (a model that is a quadratic form of parameters with a quadratic loss) exhibits progressive sharpening of the NTK eigenvalue towards a value that differs slightly from the edge of stability.

# B  PROOFS

## B.1  LINEAR NETWORK

**Theorem 3.1.** For the two-layer linear network $f_{\mathbf{w},\mathbf{v}}(x) = \langle \mathbf{w}, \mathbf{v} \rangle x$ with scalar inputs and $2m$ parameters as detailed above, the Hessian $\mathbf{H}_{\mathrm{L}}$ spectrum consists of $m-1$ repeated eigenvalues $\lambda_{\mathrm{bulk}} = \pm \overline{\delta x}$ and two paired outlying [a] eigenvalues defined by the following expression:

$$\lambda_{\mathrm{outlier}_{1,2}} = \Lambda_{1,2}\left(\|\mathbf{w}\|^2, \|\mathbf{v}\|^2, \langle \mathbf{w}, \mathbf{v} \rangle, \sigma_x^2, \overline{\delta x}\right) = \frac{1}{2}(\sigma_{\mathbf{x}}^2\|\mathbf{w}\|^2 + \sigma_{\mathbf{x}}^2\|\mathbf{v}\|^2)$$
$$\pm \frac{1}{2}\sqrt{(\sigma_{\mathbf{x}}^2\|\mathbf{w}\|^2 + \sigma_{\mathbf{x}}^2\|\mathbf{v}\|^2)^2 + 4(\overline{\delta x}^2 + 2\sigma_{\mathbf{x}}^2\overline{\delta x}\langle \mathbf{w}, \mathbf{v} \rangle)}. \tag{1}$$

---

[a] We term these *outliers* due to their distinct formulation rather than magnitude; indeed, the small outlier can overlap with the bulk spectrum.

*Proof.*
**Stating the Hessian.** The loss function can be written as $\ell(\mathbf{w}, \mathbf{v}) = \frac{1}{2n}\sum_{i=1}^{n}(\mathbf{w}^\top\mathbf{v}\cdot x_i - y_i)^2$, where $n$ is the number of data points under consideration. Hence the gradient with respect to the parameters comes out to be:

$$\nabla_{\mathbf{w}}\ell = \left(\frac{1}{n}\sum_{i=1}^{n}x_i\delta_i\right)\mathbf{v} =: \overline{\delta x}\,\mathbf{v} \tag{5}$$

$$\nabla_{\mathbf{v}}\ell = \left(\frac{1}{n}\sum_{i=1}^{n}x_i\delta_i\right)\mathbf{w} =: \overline{\delta x}\,\mathbf{w} \tag{6}$$

where, $\delta_i = \langle \mathbf{w}, \mathbf{v} \rangle x_i - y_i$ and we use the shorthand $\overline{x\delta}$ to designate the (uncentered) residual-input covariance. Also, let us denote the input mean as $\mu = \frac{1}{n}\sum_{i=1}^{n}x_i$ and the (uncentered) input standard deviation as $\sigma = \sqrt{\frac{1}{n}\sum_{i=1}^{n}x_i^2}$. Besides, let us denote the (uncentered) input-output covariance as $\overline{yx} = \frac{1}{n}\sum_{i=1}^{n}y_ix_i$. The second-order partial derivatives which will constitute the Hessian matrix turn out to be:

$$\nabla_{\mathbf{w},\mathbf{w}}^2\ell = \sigma_x^2\,\mathbf{v}\mathbf{v}^\top \tag{7}$$

$$\nabla_{\mathbf{v},\mathbf{v}}^2\ell = \sigma_x^2\,\mathbf{w}\mathbf{w}^\top \tag{8}$$

$$\nabla_{\mathbf{w},\mathbf{v}}^2\ell = \frac{\partial^2\ell}{\partial\mathbf{w}\partial\mathbf{v}} = \overline{\delta x}\,\mathbf{I}_m + \sigma_x^2\,\mathbf{v}\mathbf{w}^\top = (\nabla_{\mathbf{v},\mathbf{w}}^2\ell)^\top \tag{9}$$

We can also express the above in the matrix form as follows:

$$\mathbf{H}_{\mathrm{L}} = \begin{matrix} & \frac{\partial}{\partial\mathbf{v}^\top} & \frac{\partial}{\partial\mathbf{w}^\top} \\ \frac{\partial}{\partial\mathbf{v}} & \\ \frac{\partial}{\partial\mathbf{w}} & \end{matrix}\begin{pmatrix} \sigma_x^2\,\mathbf{w}\mathbf{w}^\top & \sigma_x^2\,\mathbf{w}\mathbf{v}^\top + \overline{\delta x}\mathbf{I}_m \\ \sigma_x^2\,\mathbf{v}\mathbf{w}^\top + \overline{\delta x}\mathbf{I}_m & \sigma_x^2\,\mathbf{v}\mathbf{v}^\top \end{pmatrix} \tag{10}$$

$$\mathbf{H}_{\mathrm{L}}\cdot\nabla_{\boldsymbol{\theta}}L(\boldsymbol{\theta}) = \begin{matrix} & \frac{\partial}{\partial\mathbf{v}^\top} & \frac{\partial}{\partial\mathbf{w}^\top} \\ \frac{\partial}{\partial\mathbf{v}} & \\ \frac{\partial}{\partial\mathbf{w}} & \end{matrix}\begin{pmatrix} \sigma_x^2\,\mathbf{w}\mathbf{w}^\top & \sigma_x^2\,\mathbf{w}\mathbf{v}^\top + \overline{\delta x}\mathbf{I}_m \\ \sigma_x^2\,\mathbf{v}\mathbf{w}^\top + \overline{\delta x}\mathbf{I}_m & \sigma_x^2\,\mathbf{v}\mathbf{v}^\top \end{pmatrix}\begin{pmatrix} \overline{\delta x}\mathbf{w} \\ \overline{\delta x}\mathbf{v} \end{pmatrix} \tag{11}$$

$$= \begin{pmatrix} \overline{\delta x}\sigma^2(\|\mathbf{w}\|^2 + \|\mathbf{v}\|^2)\mathbf{w} + \overline{\delta x}^2\mathbf{v} \\ \overline{\delta x}\sigma^2(\|\mathbf{w}\|^2 + \|\mathbf{v}\|^2)\mathbf{v} + \overline{\delta x}^2\mathbf{w} \end{pmatrix} \tag{12}$$

**Solving the eigenvalues.** To solve for the eigenvalues, we solve its characteristic equation, namely $|\mathbf{H}_{\mathrm{L}} - \lambda\mathbf{I}_p| = 0$, where $p = 2m$ is the number of parameters. Alternatively we have,

$$\left| \begin{pmatrix} \sigma_x^2 \mathbf{w}\mathbf{w}^\top - \lambda \mathbf{I}_m & \sigma_x^2 \mathbf{w}\mathbf{v}^\top + \overline{\delta x}\mathbf{I}_m \\ \sigma_x^2 \mathbf{v}\mathbf{w}^\top + \overline{\delta x}\mathbf{I}_m & \sigma_x^2 \mathbf{v}\mathbf{v}^\top - \lambda \mathbf{I}_m \end{pmatrix} \right| = 0 \tag{13}$$

Via the Schur complement, we have $\left| \begin{pmatrix} \mathbf{A} & \mathbf{B} \\ \mathbf{C} & \mathbf{D} \end{pmatrix} \right| = |\mathbf{D}||\mathbf{A} - \mathbf{B}\mathbf{D}^{-1}\mathbf{C}|$. We can apply this to above equation, where $\mathbf{D} = \sigma_x^2 \mathbf{v}\mathbf{v}^\top - \lambda \mathbf{I}_m$, which is invertible as long as $\lambda$ is non-zero and $\lambda \neq \|\mathbf{v}\|^2 \sigma^2$. Hence determinant of $\mathbf{D}$ won't be zero, and the roots of the equation above (which will yield us the eigenvalues) will come from the other term, $|\mathbf{A} - \mathbf{B}\mathbf{D}^{-1}\mathbf{C}| = 0$. Let us then calculate it:

$$\left| \sigma_x^2 \mathbf{w}\mathbf{w}^\top - \lambda \mathbf{I}_m - (\sigma_x^2 \mathbf{w}\mathbf{v}^\top + \overline{\delta x}\,\mathbf{I}_m)(\sigma_x^2 \mathbf{v}\mathbf{v}^\top - \lambda \mathbf{I}_m)^{-1}(\sigma_x^2 \mathbf{v}\mathbf{w}^\top + \overline{\delta x}\,\mathbf{I}_m) \right| = 0 \tag{14}$$

Next, we use the Woodbury matrix identity, i.e., $(\mathbf{A} + \mathbf{U}\mathbf{C}\mathbf{V})^{-1} = \mathbf{A}^{-1} - \mathbf{A}^{-1}\mathbf{U}(\mathbf{C}^{-1} + \mathbf{V}\mathbf{A}^{-1}\mathbf{U})^{-1}\mathbf{V}\mathbf{A}^{-1}$, which gives us that:

$$(\sigma^2 \mathbf{v}\mathbf{v}^\top - \lambda \mathbf{I}_m)^{-1} = \frac{-1}{\lambda}\mathbf{I}_m - \frac{\sigma_x^2}{\lambda(\lambda - \|\mathbf{v}\|^2 \sigma_x^2)}\mathbf{v}\mathbf{v}^\top \tag{15}$$

Putting this back in the above equation and expanding the product gives us,

$$\left| \sigma_x^2 \mathbf{w}\mathbf{w}^\top - \lambda \mathbf{I}_m + (\sigma_x^2 \mathbf{w}\mathbf{v}^\top + \overline{\delta x}\,\mathbf{I}_m)(\frac{1}{\lambda}\mathbf{I}_m + \frac{\sigma_x^2}{\lambda(\lambda - \|\mathbf{v}\|^2 \sigma_x^2)}\mathbf{v}\mathbf{v}^\top)(\sigma_x^2 \mathbf{v}\mathbf{w}^\top + \overline{\delta x}\,\mathbf{I}_m) \right| = 0 \tag{16}$$

$$\Big| \sigma_x^2 \mathbf{w}\mathbf{w}^\top - \lambda \mathbf{I}_m \tag{17}$$

$$+ \frac{\sigma^4 \|\mathbf{v}\|^2}{\lambda}\mathbf{w}\mathbf{w}^\top + \frac{\sigma_x^2 \overline{\delta x}}{\lambda}\mathbf{w}\mathbf{v}^\top + \frac{\sigma^6 \|\mathbf{v}\|^4}{\lambda(\lambda - \sigma_x^2 \|\mathbf{v}\|^2)}\mathbf{w}\mathbf{w}^\top + \frac{\sigma^4 \overline{\delta x}\|\mathbf{v}\|^2}{\lambda(\lambda - \sigma_x^2 \|\mathbf{v}\|^2)}\mathbf{w}\mathbf{v}^\top \tag{18}$$

$$+ \frac{\sigma_x^2 \overline{\delta x}}{\lambda}\mathbf{v}\mathbf{w}^\top + \frac{\overline{\delta x}^2}{\lambda}\mathbf{I}_m + \frac{\sigma^4 \overline{\delta x}\|\mathbf{v}\|^2}{\lambda(\lambda - \sigma_x^2 \|\mathbf{v}\|^2)}\mathbf{v}\mathbf{w}^\top + \frac{\sigma_x^2 \overline{\delta x}^2}{\lambda(\lambda - \sigma_x^2 \|\mathbf{v}\|^2)}\mathbf{v}\mathbf{v}^\top \Big| = 0 \tag{19}$$

Let us analyze the coefficients for each of the matrices one by one, starting with $\mathbf{w}\mathbf{w}^\top$

$$\sigma_x^2 + \frac{\sigma^4 \|\mathbf{v}\|^2}{\lambda} + \frac{\sigma^6 \|\mathbf{v}\|^4}{\lambda(\lambda - \sigma_x^2 \|\mathbf{v}\|^2)} = \frac{(\sigma_x^2 \lambda(\lambda - \sigma_x^2 \|\mathbf{v}\|^2) + \sigma^4 \|\mathbf{v}\|^2(\lambda - \sigma_x^2 \|\mathbf{v}\|^2) + \sigma^6 \|\mathbf{v}\|^4)}{\lambda(\lambda - \sigma_x^2 \|\mathbf{v}\|^2)} \tag{20}$$

$$= \frac{\sigma^2 \lambda}{\lambda - \sigma_x^2 \|\mathbf{v}\|^2} \tag{21}$$

Next up, both $\mathbf{w}\mathbf{v}^\top$ and $\mathbf{v}\mathbf{w}^\top$ have the same coefficient:

$$\frac{\sigma_x^2 \overline{\delta x}}{\lambda} + \frac{\sigma^4 \overline{\delta x}\|\mathbf{v}\|^2}{\lambda(\lambda - \sigma_x^2 \|\mathbf{v}\|^2)} = \frac{(\lambda - \sigma_x^2 \|\mathbf{v}\|^2)\sigma_x^2 \overline{\delta x} + \sigma^4 \overline{\delta x}\|\mathbf{v}\|^2}{\lambda(\lambda - \sigma_x^2 \|\mathbf{v}\|^2)} \tag{22}$$

$$= \frac{\sigma_x^2 \overline{\delta x}}{\lambda - \sigma_x^2 \|\mathbf{v}\|^2} \tag{23}$$

Hence, the above characteristic equation can be rewritten as,

$$\Big| \frac{\sigma_x^2 \lambda}{\lambda - \sigma_x^2 \|\mathbf{v}\|^2}\mathbf{w}\mathbf{w}^\top + \frac{\sigma_x^2 \overline{\delta x}}{\lambda - \sigma_x^2 \|\mathbf{v}\|^2}(\mathbf{w}\mathbf{v}^\top + \mathbf{v}\mathbf{w}^\top) + \frac{\sigma_x^2 \overline{\delta x}^2}{\lambda(\lambda - \sigma_x^2 \|\mathbf{v}\|^2)}\mathbf{v}\mathbf{v}^\top \tag{24}$$

$$- (\lambda - \frac{\overline{\delta x}^2}{\lambda})\mathbf{I}_m \Big| = 0 \tag{25}$$

Assuming $\lambda \neq 0$, multiply the equation by $\lambda$ and $(\lambda - \sigma_x^2 \|\mathbf{v}\|^2)$ (since $|\sigma_x^2 \mathbf{v}\mathbf{v}^\top - \lambda \mathbf{I}_m| \neq 0$) yields,

$$\left| \sigma_x^2 \lambda^2 \mathbf{w}\mathbf{w}^\top + \sigma_x^2 \overline{\delta x}\lambda(\mathbf{w}\mathbf{v}^\top + \mathbf{v}\mathbf{w}^\top) + \sigma_x^2 \overline{\delta x}^2 \mathbf{v}\mathbf{v}^\top - (\lambda^2 - \overline{\delta x}^2)(\lambda - \sigma_x^2 \|\mathbf{v}\|^2)\mathbf{I}_m \right| = 0 \tag{26}$$

Set $\mathbf{z} = \lambda\mathbf{w} + \overline{\delta x}\mathbf{v}$, and $\nu = (\lambda^2 - \overline{\delta x}^2)(\lambda - \sigma_x^2\|\mathbf{v}\|^2)$ we can express the above equation more compactly as

$$\left|\sigma_x^2\mathbf{z}\mathbf{z}^\top - \nu\mathbf{I}_m\right| = 0 \tag{27}$$

The determinant of the above matrix is the product of its eigenvalues, which comes out as:

$$\left|\sigma_x^2\mathbf{z}\mathbf{z}^\top - \nu\mathbf{I}_m\right| = \nu^{m-1}(\sigma_x^2\|\mathbf{z}\|^2 - \nu) = 0 \tag{28}$$

This implies that they are $m - 1$ repeated roots of $\nu = 0$, and once when $\sigma_x^2\|\mathbf{z}\|^2 - \nu = 0$.

Since $\lambda \neq \sigma_x^2\|\mathbf{v}\|^2$, we get $m - 1$ repeated solutions of $\lambda^2 - \overline{\delta x}^2 = 0$ or $m - 1$ times

$$\lambda = \pm\overline{\delta x} = \pm\frac{1}{n}\sum_{i=1}^{n}(\langle\mathbf{w}, \mathbf{v}\rangle x_i - y_i)\,x_i\,. \tag{29}$$

The other solution corresponds to solving the following equation in $\lambda$:

$$\sigma_x^2\lambda^2\|\mathbf{w}\|^2 + \sigma_x^2\overline{\delta x}^2\|\mathbf{v}\|^2 + 2\sigma_x^2\overline{\delta x}\lambda\langle\mathbf{w},\mathbf{v}\rangle - \lambda^3 + \overline{\delta x}^2\|\mathbf{v}\|^2 + \overline{\delta x}^2 - \sigma_x^2\overline{\delta x}^2\|\mathbf{v}\|^2 = 0 \tag{30}$$

$$- \lambda^3 + (\sigma_x^2\|\mathbf{w}\|^2 + \sigma_x^2\|\mathbf{v}\|^2)\lambda^2 + \lambda(\overline{\delta x}^2 + 2\sigma_x^2\overline{\delta x}\langle\mathbf{w},\mathbf{v}\rangle) \tag{31}$$

$$= -\lambda(\lambda^2 - (\sigma_x^2\|\mathbf{w}\|^2 + \sigma_x^2\|\mathbf{v}\|^2)\lambda - (\overline{\delta x}^2 + 2\sigma_x^2\overline{\delta x}\langle\mathbf{w},\mathbf{v}\rangle)) = 0 \tag{32}$$

Again, $\lambda \neq 0$ by assumption above. Then solving the quadratic in $\lambda$ gives us the following two roots:

$$\boxed{\lambda = \frac{1}{2}(\sigma_x^2\|\mathbf{w}\|^2 + \sigma_x^2\|\mathbf{v}\|^2) \pm \frac{1}{2}\sqrt{(\sigma_x^2\|\mathbf{w}\|^2 + \sigma_x^2\|\mathbf{v}\|^2)^2 + 4(\overline{\delta x}^2 + 2\sigma_x^2\overline{\delta x}\langle\mathbf{w},\mathbf{v}\rangle)}} \tag{33}$$

where $\overline{\delta x}^2 + 2\sigma_x^2\overline{\delta x}\langle\mathbf{w},\mathbf{v}\rangle = \overline{\delta x}(\overline{\delta x} + 2\sigma_x^2\langle\mathbf{w},\mathbf{v}\rangle)$, which can be better written as,

$$(\langle\mathbf{w},\mathbf{v}\rangle\sigma_x^2 - \overline{yx})(3\langle\mathbf{w},\mathbf{v}\rangle\sigma_x^2 - \overline{yx})\,.$$

Further, notice that the residual input covariance can be upper-bounded in terms of the overall loss and input variance.

$$\overline{\delta x} = \frac{1}{n}\sum_{i=1}^{n}\delta_i x_i \leq \frac{1}{n}\sqrt{\sum_{i=1}^{n}\delta_i^2}\sqrt{\sum_{i=1}^{n}x_i^2} = \sqrt{\frac{1}{n}\sum_{i=1}^{n}\delta_i^2}\sqrt{\frac{1}{n}\sum_{i=1}^{n}x_i^2} \tag{34}$$

Thus, $\overline{\delta x} \leq \sqrt{2\ell}\,\sigma$. Further since the most negative eigenvalue is given by, $\lambda_{\min} = -\overline{\delta x}$, we have that

$$\lambda_{\min} \geq -\sqrt{2\ell}\,\sigma\,.$$

$\square$

**Theorem 3.2.** For the above setting, the Hessian eigenvectors corresponding to the outlying eigenvalues, determined up to scaling and sign, take the form, for $i \in \{1, 2\}$, given below:

$$\mathbf{z}_{\text{outlier}_i} = \begin{pmatrix} \lambda_{\text{outlier}_i}\,\mathbf{w} + \overline{\delta x}\,\mathbf{v} \\ \overline{\delta x}\,\mathbf{w} + \lambda_{\text{outlier}_i}\,\mathbf{v} \end{pmatrix} = \lambda_{\text{outlier}_i}\begin{pmatrix} \mathbf{w} \\ \mathbf{v} \end{pmatrix} + \overline{\delta x}\begin{pmatrix} \mathbf{v} \\ \mathbf{w} \end{pmatrix}$$

*Proof.*

Let us recall the Hessian considered in Eq. 10 for the multiple datapoints setting in the case of linear networks.

$$
\mathbf{H}_{\mathrm{L}} = \begin{matrix} & \overset{\frac{\partial}{\partial \mathbf{v}^\top}}{} & \overset{\frac{\partial}{\partial \mathbf{w}^\top}}{} \\ \begin{matrix} \frac{\partial}{\partial \mathbf{v}} \\ \frac{\partial}{\partial \mathbf{w}} \end{matrix} & \begin{pmatrix} \sigma_x^2\, \mathbf{w}\mathbf{w}^\top & \sigma_x^2\, \mathbf{w}\mathbf{v}^\top + \overline{\delta x}\mathbf{I}_m \\ \sigma_x^2\, \mathbf{v}\mathbf{w}^\top + \overline{\delta x}\mathbf{I}_m & \sigma_x^2\, \mathbf{v}\mathbf{v}^\top \end{pmatrix} \end{matrix} \tag{35}
$$

While we have solved for the eigenvalues, the form of the eigenvectors is still not apparent. In this section, we aim to work towards a closed form for the eigenvectors. For starters, let us assume an eigenvector of the Hessian matrix above, $\mathbf{z}$, is of the form $\mathbf{z} = \begin{pmatrix} \mathbf{z}_1 \\ \mathbf{z}_2 \end{pmatrix}$. Then in order to obtain the eigenvectors we need to solve the following system of equations:

$$
\begin{pmatrix} \sigma_x^2\, \mathbf{w}\mathbf{w}^\top & \sigma_x^2\, \mathbf{w}\mathbf{v}^\top + \overline{\delta x}\mathbf{I}_m \\ \sigma_x^2\, \mathbf{v}\mathbf{w}^\top + \overline{\delta x}\mathbf{I}_m & \sigma_x^2\, \mathbf{v}\mathbf{v}^\top \end{pmatrix} \begin{pmatrix} \mathbf{z}_1 \\ \mathbf{z}_2 \end{pmatrix} = \lambda \begin{pmatrix} \mathbf{z}_1 \\ \mathbf{z}_2 \end{pmatrix} \tag{36}
$$

which can be expressed as,

$$
\sigma_x^2 \langle \mathbf{w}, \mathbf{z}_1 \rangle\, \mathbf{w} + \sigma_x^2 \langle \mathbf{v}, \mathbf{z}_2 \rangle\, \mathbf{w} + \overline{\delta x}\, \mathbf{z}_2 = \lambda \mathbf{z}_1 \tag{37}
$$

$$
\sigma_x^2 \langle \mathbf{w}, \mathbf{z}_1 \rangle\, \mathbf{v} + \sigma_x^2 \langle \mathbf{v}, \mathbf{z}_2 \rangle\, \mathbf{v} + \overline{\delta x}\, \mathbf{z}_1 = \lambda \mathbf{z}_2 \tag{38}
$$

Taking the inner-product with $\mathbf{v}$ on both sides of first equation and with $\mathbf{w}$ in second equation yields,

$$
\sigma_x^2 \langle \mathbf{w}, \mathbf{z}_1 \rangle \langle \mathbf{w}, \mathbf{v} \rangle + \sigma_x^2 \langle \mathbf{v}, \mathbf{z}_2 \rangle \langle \mathbf{w}, \mathbf{v} \rangle + \overline{\delta x}\, \langle \mathbf{z}_2, \mathbf{v} \rangle = \lambda \langle \mathbf{z}_1, \mathbf{v} \rangle \tag{39}
$$

$$
\sigma_x^2 \langle \mathbf{w}, \mathbf{z}_1 \rangle \langle \mathbf{v}, \mathbf{w} \rangle + \sigma_x^2 \langle \mathbf{v}, \mathbf{z}_2 \rangle \langle \mathbf{v}, \mathbf{w} \rangle + \overline{\delta x}\, \langle \mathbf{z}_1, \mathbf{w} \rangle = \lambda \langle \mathbf{z}_2, \mathbf{w} \rangle \tag{40}
$$

Subtracting the second equation from the first and rearranging gives,

$$
\overline{\delta x}\, \langle \mathbf{z}_2, \mathbf{v} \rangle - \lambda \langle \mathbf{z}_1, \mathbf{v} \rangle = \overline{\delta x}\, \langle \mathbf{z}_1, \mathbf{w} \rangle - \lambda \langle \mathbf{z}_2, \mathbf{w} \rangle \tag{41}
$$

Alternatively,

$$
\langle \overline{\delta x}\, \mathbf{z}_2 - \lambda\, \mathbf{z}_1, \mathbf{v} \rangle = \langle \overline{\delta x}\, \mathbf{z}_1 - \lambda\, \mathbf{z}_2, \mathbf{w} \rangle \tag{42}
$$

Thus we have an equation of the form $\langle \mathbf{a}, \mathbf{b} \rangle = \langle \mathbf{c}, \mathbf{d} \rangle$, whose possible solutions are:

1. $\langle \mathbf{a}, \mathbf{b} \rangle = 0$ and $\langle \mathbf{c}, \mathbf{d} \rangle = 0$. While this is a general condition, there are also some specific instantiations of this when this is possible as listed below.

   (a) $\mathbf{a} = 0$ and $\mathbf{c} = 0$.
   (b) $\mathbf{a} = 0$ and $\mathbf{d} = 0$.
   (c) $\mathbf{b} = 0$ and $\mathbf{c} = 0$.
   (d) $\mathbf{b} = 0$ and $\mathbf{d} = 0$.

   The grayed out possibilities require the parameter vectors to be zero, so we discard them as that need not be the case. As we will see in the next proof in this section, possibility (1a) will give us bulk eigenvectors.

2. $\mathbf{a} = \mathbf{c}$ and $\mathbf{b} = \mathbf{d}$, and $\mathbf{a}, \mathbf{b}, \mathbf{c}, \mathbf{d} \neq 0$ (up to scale, since $\alpha\, \mathbf{a}$ and $\alpha^{-1}\mathbf{b}$, for $\alpha \neq 0$ is also a valid solution).

   Again this requires the parameter vectors to be equal, which may not be the case necessarily.

3. $\mathbf{a} = \mathbf{d}$ and $\mathbf{b} = \mathbf{c}$, and $\mathbf{a}, \mathbf{b}, \mathbf{c}, \mathbf{d} \neq 0$ (up to scale).

Considering that $\mathbf{v} \neq \mathbf{w}$ and $\mathbf{v}, \mathbf{w} \neq 0$. Then using the (3) option above, we get (for some $\alpha \neq 0$):

$$\overline{\delta x}\, \mathbf{z}_2 - \lambda \mathbf{z}_1 = \alpha \mathbf{w} \tag{43}$$

$$\overline{\delta x}\mathbf{z}_1 - \lambda\, \mathbf{z}_2 = \alpha \mathbf{v} \tag{44}$$

Then solving this pair of equations for $\mathbf{z}_1$ and $\mathbf{z}_2$ gives:

$$(\overline{\delta x}^2 - \lambda^2)\, \mathbf{z}_1 = \alpha \lambda\, \mathbf{w} + \alpha \overline{\delta x}\, \mathbf{v} \tag{45}$$

$$(\overline{\delta x}^2 - \lambda^2)\, \mathbf{z}_2 = \alpha \overline{\delta x}\, \mathbf{w} + \alpha \lambda\, \mathbf{v} \tag{46}$$

Hence, at last, the eigenvector is of the following form:

$$\mathbf{z} = \frac{\alpha}{\overline{\delta x}^2 - \lambda^2} \begin{pmatrix} \lambda \mathbf{w} + \overline{\delta x}\mathbf{v} \\ \overline{\delta x}\mathbf{w} + \lambda \mathbf{v} \end{pmatrix} \tag{47}$$

In order to check the validity of the above solution, let us multiply the Hessian with it. Besides, from there we may also see the eigenvalue corresponding to this eigenvector.

$$\begin{pmatrix} \sigma_x^2\, \mathbf{w}\mathbf{w}^\top & \sigma_x^2\, \mathbf{w}\mathbf{v}^\top + \overline{\delta x}\mathbf{I}_m \\ \sigma_x^2\, \mathbf{v}\mathbf{w}^\top + \overline{\delta x}\mathbf{I}_m & \sigma_x^2\, \mathbf{v}\mathbf{v}^\top \end{pmatrix} \begin{pmatrix} \lambda \mathbf{w} + \overline{\delta x}\mathbf{v} \\ \overline{\delta x}\mathbf{w} + \lambda \mathbf{v} \end{pmatrix} \frac{\alpha}{\overline{\delta x}^2 - \lambda^2}$$

$$= \begin{pmatrix} \lambda\sigma_x^2\langle\mathbf{w},\mathbf{w}\rangle\,\mathbf{w} + \overline{\delta x}\sigma_x^2\langle\mathbf{w},\mathbf{v}\rangle\mathbf{w} + \overline{\delta x}\sigma_x^2\langle\mathbf{w},\mathbf{v}\rangle\mathbf{w} + \lambda\sigma_x^2\langle\mathbf{v},\mathbf{v}\rangle\mathbf{w} + \overline{\delta x}^2\mathbf{w} + \lambda\overline{\delta x}\mathbf{v} \\ \lambda\sigma_x^2\langle\mathbf{w},\mathbf{w}\rangle\mathbf{v} + \overline{\delta x}\sigma_x^2\langle\mathbf{w},\mathbf{v}\rangle\mathbf{v} + \lambda\overline{\delta x}\mathbf{w} + \overline{\delta x}^2\mathbf{v} + \overline{\delta x}\sigma_x^2\langle\mathbf{w},\mathbf{v}\rangle\mathbf{v} + \lambda\sigma_x^2\langle\mathbf{v},\mathbf{v}\rangle\mathbf{v} \end{pmatrix} \frac{\alpha}{\overline{\delta x}^2 - \lambda^2}$$

The above can be simplified to:

$$\begin{pmatrix} \left(\lambda\sigma_x^2\|\mathbf{w}\|^2 + \lambda\sigma_x^2\|\mathbf{v}\|^2 + 2\overline{\delta x}\sigma_x^2\langle\mathbf{w},\mathbf{v}\rangle + \overline{\delta x}^2\right)\,\mathbf{w} + \lambda\overline{\delta x}\mathbf{v} \\ \lambda\overline{\delta x}\mathbf{w} + \left(\lambda\sigma_x^2\|\mathbf{w}\|^2 + \lambda\sigma_x^2\|\mathbf{v}\|^2 + 2\overline{\delta x}\sigma_x^2\langle\mathbf{w},\mathbf{v}\rangle + \overline{\delta x}^2\right)\,\mathbf{v} \end{pmatrix} \frac{\alpha}{\overline{\delta x}^2 - \lambda^2}$$

For this to be a valid eigenvector, the term in brackets should be equal to $\lambda^2$, i.e.,

$$\lambda\sigma_x^2\|\mathbf{w}\|^2 + \lambda\sigma_x^2\|\mathbf{v}\|^2 + 2\overline{\delta x}\sigma_x^2\langle\mathbf{w},\mathbf{v}\rangle + \overline{\delta x}^2 = \lambda^2$$

Solving this yields,

$$\lambda = \frac{1}{2}(\sigma_x^2\|\mathbf{w}\|^2 + \sigma_x^2\|\mathbf{v}\|^2) \pm \frac{1}{2}\sqrt{(\sigma_x^2\|\mathbf{w}\|^2 + \sigma_x^2\|\mathbf{v}\|^2)^2 + 4(2\overline{\delta x}\sigma_x^2\langle\mathbf{w},\mathbf{v}\rangle + \overline{\delta x}^2)}$$

*which is precisely the solution for the eigenvalues we had obtained by solving the characteristic equation.*

**Remark B.1.** Given the decoupling of the Hessian spectrum in ReLU cells, the above eigenvector derivation should also generalize for the ReLU case.

$\square$

**Theorem 3.3.** For the above setting, the Hessian eigenvectors corresponding to the bulk eigenvalues have the form, determined up to scaling and sign, $\mathbf{z}_{\text{bulk}} = \begin{pmatrix} \hat{\mathbf{z}}_{\text{bulk}}^\top & \text{sgn}(\lambda_{\text{bulk}})\,\hat{\mathbf{z}}_{\text{bulk}}^\top \end{pmatrix}^\top$

with $\hat{\mathbf{z}}_{\text{bulk}} = \left(\mathbf{I} - \dfrac{(\mathbf{w} + \text{sgn}(\lambda_{\text{bulk}})\,\mathbf{v})(\mathbf{w} + \text{sgn}(\lambda_{\text{bulk}})\,\mathbf{v})^\top}{\|\mathbf{w} + \text{sgn}(\lambda_{\text{bulk}})\,\mathbf{v}\|^2}\right)\mathbf{c}$, for some vector $\mathbf{c}$.

*Proof.*

From the derivation of the eigenvector in the previous proof in the section, we have the possibility (1a) remaining, and we find that, as the bulk eigenvalues are $\lambda = \pm\overline{\delta x}$, this would fit neatly with constraint from Eqn. 42. Moreover we obtain the following:

$$\lambda = \overline{\delta x} : \quad \mathbf{z}_1 = \mathbf{z}_2 \tag{48}$$

$$\lambda = -\overline{\delta x} : \quad \mathbf{z}_1 = -\mathbf{z}_2 \tag{49}$$

Further as the bulk eigenvectors have to be orthogonal to the outlier eigenvectors, we get the following constraint:

$$\lambda_{\text{outlier}}\langle\mathbf{w}, \mathbf{z}_1\rangle + \overline{\delta x}\langle\mathbf{v}, \mathbf{z}_1\rangle + \overline{\delta x}\langle\mathbf{w}, \mathbf{z}_2\rangle + \lambda_{\text{outlier}}\langle\mathbf{v}, \mathbf{z}_2\rangle = 0 \tag{50}$$

$$\langle\mathbf{w}, \lambda_{\text{outlier}}\mathbf{z}_1 + \overline{\delta x}\mathbf{z}_2\rangle + \langle\mathbf{v}, \overline{\delta x}\mathbf{z}_1 + \lambda_{\text{outlier}}\mathbf{z}_2\rangle = 0 \tag{51}$$

*(a) Bulk eigenvalue and residual-input covariance have same signs:* Let us now consider $\mathbf{z}_1 = \mathbf{z}_2$, then we get the following constraint:

$$(\lambda_{\text{outlier}} + \overline{\delta x})\langle\mathbf{w} + \mathbf{v}, \mathbf{z}_1\rangle = 0 \tag{52}$$

Hence, in this case $\mathbf{z}_1$ is of the form $\left(\mathbf{I} - \frac{(\mathbf{w}+\mathbf{v})(\mathbf{w}+\mathbf{v})^\top}{\|\mathbf{w}+\mathbf{v}\|^2}\right)\mathbf{c}$ for some vector $\mathbf{c}$, and they can simply be obtained by computing the eigenvectors of this matrix, corresponding to non-zero eigenvalues.

*(b) Bulk eigenvalue and residual-input covariance have opposite signs:* While plugging in $\mathbf{z}_1 = -\mathbf{z}_2$ for the other half of the bulk, we get the following constraint:

$$(\lambda_{\text{outlier}} - \overline{\delta x})\langle\mathbf{w} - \mathbf{v}, \mathbf{z}_1\rangle = 0 \tag{53}$$

Hence, in this case $\mathbf{z}_1$ is of the form $\left(\mathbf{I} - \frac{(\mathbf{w}-\mathbf{v})(\mathbf{w}-\mathbf{v})^\top}{\|\mathbf{w}-\mathbf{v}\|^2}\right)\mathbf{c}$ for some vector $\mathbf{c}$, and they can simply be obtained by computing the eigenvectors of this matrix, corresponding to non-zero eigenvalues.

Finally, the bulk eigenvectors from the cases (a) and (b) are orthogonal between themselves since,

$$\begin{pmatrix}\mathbf{z}_1 & \mathbf{z}_1\end{pmatrix}^\top \begin{pmatrix}\mathbf{z}_1 \\ -\mathbf{z}_1\end{pmatrix} = 0$$

$\square$

**Corollary 3.1.** Assume that we initialize the parameters as $\boldsymbol{\theta}_0 = (\mathbf{v}_0^\top, \mathbf{w}_0^\top)^\top$, and run gradient descent with step $\Delta\boldsymbol{\theta}_t = -\eta_t\overline{\delta x}_t(\mathbf{w}_t^\top, \mathbf{v}_t^\top)^\top$ to go from $\boldsymbol{\theta}_t$ to $\boldsymbol{\theta}_{t+1}$. Then the bulk eigenspace does not change over training $\hat{\mathbf{z}}_{\text{bulk}} = \hat{\mathbf{z}}_{\text{bulk}}^{(t)} = \hat{\mathbf{z}}_{\text{bulk}}^{(0)}$, and $\langle\boldsymbol{\theta}_t, \hat{\mathbf{z}}_{\text{bulk}}\rangle = 0$, $\forall t \in [0, T]$.

*Proof.*

To see how the bulk subspace changes, we consider how the vector whose orthogonal complement we are interested in changes. Namely, we want to look at $\mathbf{w}_t + \mathbf{v}_t$ and $\mathbf{w}_t - \mathbf{v}_t$.

From the gradient descent equations, we have:

$$\mathbf{w}_{t+1} = \mathbf{w}_t - \eta_t\overline{\delta x}_t\,\mathbf{v}_t$$

$$\mathbf{v}_{t+1} = \mathbf{v}_t - \eta_t\overline{\delta x}_t\,\mathbf{w}_t$$

Hence, we have that:

$$\mathbf{w}_{t+1} + \mathbf{v}_{t+1} = \left(1 - \eta_t \overline{\delta x_t}\right)\left(\mathbf{w}_t + \mathbf{v}_t\right)$$

$$\mathbf{w}_{t+1} - \mathbf{v}_{t+1} = \left(1 + \eta_t \overline{\delta x_t}\right)\left(\mathbf{w}_t - \mathbf{v}_t\right)$$

Which itself can be written as

$$\mathbf{w}_{t+1} + \mathbf{v}_{t+1} = \prod_{i=0}^{t}(1 - \eta_i \overline{\delta x_i}) \cdot (\mathbf{w}_0 + \mathbf{v}_0)$$

$$\mathbf{w}_{t+1} - \mathbf{v}_{t+1} = \prod_{i=0}^{t}(1 + \eta_i \overline{\delta x_i}) \cdot (\mathbf{w}_0 - \mathbf{v}_0)$$

Therefore even if the parameters are being updated, the bulk subspace does not change as $\hat{z}_{\text{bulk}}$ continues to live in the orthogonal complement of $\mathbf{w}_0 \pm \mathbf{v}_0$ throughout optimization. The only possibility is that of a decrease in dimension when $\lim_{t \to \infty} \overline{\delta x_t} = 0$ and $\lambda_{\text{bulk}} = 0$. $\qquad\square$

## B.2 ReLU Network

**Theorem 3.4.** For the above ReLU setting, the bulk Hessian spectrum consists of $q - 1$ and $m - q - 1$ repeated eigenvalues in signed pairs, $\lambda_{\text{bulk}}^{+} = \pm \frac{n_+}{n} \overline{x\delta}_+$ , $\lambda_{\text{bulk}}^{-} = \pm \frac{n_-}{n} \overline{x\delta}_-$ and with the outlying eigenvalues being $\lambda_{\text{outlier}_{1,2}}^{+} = \frac{n_+}{n} \Lambda_{1,2}\left(\|\mathbf{w}_+\|^2, \|\mathbf{v}_+\|^2, \langle \mathbf{w}_+, \mathbf{v}_+ \rangle, \sigma_{x+}^2, \overline{\delta x}_+\right)$ and $\lambda_{\text{outlier}_{1,2}}^{-} = \frac{n_-}{n} \Lambda_{1,2}\left(\|\mathbf{w}_-\|^2, \|\mathbf{v}_-\|^2, \langle \mathbf{w}_-, \mathbf{v}_- \rangle, \sigma_{x-}^2, \overline{\delta x}_-\right)$ .

*Proof.*
Let us now analyze the case with the ReLU non-linearity. So we have as the network function, $f(x) = \mathbf{w}^\top (\mathbf{v}x)_+$, with $(z)_+ = z\mathbb{1}\{z > 0\}$. The loss function is given by, $\ell(\mathbf{w}, \mathbf{v}) = \frac{1}{2}(\mathbf{w}^\top (\mathbf{v}x)_+ - y)^2$.

Without loss of generality, assume that first $q$ coordinates of the vector $\mathbf{v}$ have positive sign as $x$ and collected in the vector $\mathbf{v}_+$, while the rest $m - q$ coordinates being zero. On the other hand, we can collect the negative coordinates of $\mathbf{v}$ in the vector $\mathbf{v}_-$ whose, first $q$ components are zero and the rest $m - q$ contain the negative coordinates. Then we can write $\mathbf{v} = \mathbf{v}_+ + \mathbf{v}_-$. Since in this simple network, each parameter of $\mathbf{w}$ is coupled with a parameter in $\mathbf{v}$, based on the partition of $\mathbf{v}$ we can also split the $\mathbf{w}$ vector into two corresponding parts, $(\mathbf{w}_+)_j = \mathbf{w}_j \mathbb{1}\{\mathbf{v}_j > 0\}$ and $(\mathbf{w}_-)_j = \mathbf{w}_j \mathbb{1}\{\mathbf{v}_j < 0\}$. Alternatively, we can express the effect of non-linearity on $\mathbf{w}$ by the Hadamard product $\mathbf{w} \odot \mathbb{1}\{\mathbf{v} > 0\}$. But, to emphasize, the components of $\mathbf{w}_+$ need not be positive neither $\mathbf{w}_-$ need be negative.

And so, for $x > 0$, $f(x) = \langle \mathbf{w}_+, \mathbf{v}_+ \rangle x$, while for $x \leq 0$, we have that $f(x) = \langle \mathbf{w}_-, \mathbf{v}_- \rangle x$. Or more succinctly,

$$f(x) = \langle \mathbf{w}_+, \mathbf{v}_+ \rangle x \, \mathbb{1}\{x > 0\} + \langle \mathbf{w}_-, \mathbf{v}_- \rangle x \, \mathbb{1}\{x \leq 0\}$$

Hence the gradient with respect to the parameters comes out to be:

$$\nabla_{\mathbf{w}} \ell = \delta \cdot (\mathbf{v}_+ x \, \mathbb{1}\{x > 0\} + \mathbf{v}_- x \, \mathbb{1}\{x \leq 0\}) \tag{54}$$

$$\nabla_{\mathbf{v}} \ell = \delta \cdot (\mathbf{w}_+ x \, \mathbb{1}\{x > 0\} + \mathbf{w}_- x \, \mathbb{1}\{x \leq 0\}) \tag{55}$$

where, $\delta = \langle \mathbf{w}_+, \mathbf{v}_+ \rangle x \, \mathbb{1}\{x > 0\} + \langle \mathbf{w}_-, \mathbf{v}_- \rangle x \, \mathbb{1}\{x \leq 0\} - y$. The second-order partial derivatives which will constitute the Hessian matrix turn out to be:

$$\nabla_{\mathbf{w},\mathbf{w}}^2 \ell = (\mathbf{v}_+ x \, \mathbb{1}\{x > 0\} + \mathbf{v}_- x \, \mathbb{1}\{x \leq 0\})(\mathbf{v}_+ x \, \mathbb{1}\{x > 0\} + \mathbf{v}_- x \, \mathbb{1}\{x \leq 0\})^\top$$

$$\nabla_{\mathbf{v},\mathbf{v}}^2 \ell = (\mathbf{w}_+ x \, \mathbb{1}\{x > 0\} + \mathbf{w}_- x \, \mathbb{1}\{x \leq 0\})(\mathbf{w}_+ x \, \mathbb{1}\{x > 0\} + \mathbf{w}_- x \, \mathbb{1}\{x \leq 0\})^\top$$

$$\nabla_{\mathbf{w},\mathbf{v}}^2 \ell = \frac{\partial^2 \ell}{\partial \mathbf{w} \partial \mathbf{v}}$$

$$= \text{diag}(\delta \, \mathbf{x}_\pm) + (\mathbf{v}_+ x \, \mathbb{1}\{x > 0\} + \mathbf{v}_- x \, \mathbb{1}\{x \leq 0\})(\mathbf{w}_+ x \, \mathbb{1}\{x > 0\} + \mathbf{w}_- x \, \mathbb{1}\{x \leq 0\})^\top$$

where, $\mathrm{diag}(\cdot)$ denotes a diagonal matrix formed by the corresponding vector, and $\mathbf{x}_{\pm}$ denotes the vector whose first $q$ coordinates contain equal $x\mathbb{1}\{x > 0\}$ and the other $m - q$ coordinates contain $x\mathbb{1}\{x \leq 0\}$. Said differently, we have that:

$$\mathrm{diag}(\delta\,\mathbf{x}_{\pm}) = \begin{pmatrix} \delta\,x\,\mathbb{1}\{x > 0\}\,\mathbf{I}_q & \mathbf{0} \\ \mathbf{0} & \delta\,x\,\mathbb{1}\{x \leq 0\}\,\mathbf{I}_{m-q} \end{pmatrix}$$

The expressions of the above components of the Hessian can be further simplified as,

$$\nabla^2_{\mathbf{w},\mathbf{w}}\ell = \mathbf{v}_+\mathbf{v}_+^\top x^2\,\mathbb{1}\{x > 0\} + \mathbf{v}_-\mathbf{v}_-^\top x^2\mathbb{1}\{x \leq 0\} \tag{56}$$

$$\nabla^2_{\mathbf{v},\mathbf{v}}\ell = \mathbf{w}_+\mathbf{w}_+^\top x^2\,\mathbb{1}\{x > 0\} + \mathbf{w}_-\mathbf{w}_-^\top x^2\mathbb{1}\{x \leq 0\} \tag{57}$$

$$\nabla^2_{\mathbf{w},\mathbf{v}}\ell = \mathrm{diag}(\delta\,\mathbf{x}_{\pm}) + \mathbf{v}_+\mathbf{w}_+^\top x^2\,\mathbb{1}\{x > 0\} + \mathbf{v}_-\mathbf{w}_-^\top x^2\mathbb{1}\{x \leq 0\} \tag{58}$$

**Averaging over Multiple datapoints.** Let us average the above Hessian components over multiple datapoints,

$$\nabla^2_{\mathbf{w},\mathbf{w}}\mathrm{L} = \frac{n_+}{n}\,\sigma^2_{x+}\,\mathbf{v}_+\mathbf{v}_+^\top + \frac{n_-}{n}\,\sigma^2_{x-}\,\mathbf{v}_-\mathbf{v}_-^\top \tag{59}$$

$$\nabla^2_{\mathbf{v},\mathbf{v}}\mathrm{L} = \frac{n_+}{n}\,\sigma^2_{x+}\,\mathbf{w}_+\mathbf{w}_+^\top + \frac{n_-}{n}\,\sigma^2_{x-}\,\mathbf{w}_-\mathbf{w}_-^\top \tag{60}$$

$$\nabla^2_{\mathbf{w},\mathbf{v}}\mathrm{L} = \mathrm{diag}(\overline{\delta\mathbf{x}_{\pm}}) + \frac{n_+}{n}\,\sigma^2_{x+}\,\mathbf{v}_+\mathbf{w}_+^\top + \frac{n_-}{n}\,\sigma^2_{x-}\,\mathbf{v}_-\mathbf{w}_-^\top \tag{61}$$

where, we have the the standard deviation of the positive and negative datapoints as, $\sigma_+ = \sqrt{\frac{1}{n_+}\sum_{i=1}^n x_i^2\,\mathbb{1}\{x_i > 0\}}$ and $\sigma_- = \sqrt{\frac{1}{n_-}\sum_{i=1}^n x_i^2\,\mathbb{1}\{x_i < 0\}}$ respectively, and besides, $n = n_+ + n_-$.

Besides, we let us denote the (uncentered) input-output covariance for the positive and negative cells respectively as follows,

$$\overline{yx}_+ = \frac{1}{n_+}\sum_{i=1}^n y_i x_i \mathbb{1}\{x_i > 0\}$$

$$\overline{yx}_- = \frac{1}{n_-}\sum_{i=1}^n y_i x_i \mathbb{1}\{x_i \leq 0\}$$

**Hessian decouples across ReLU cells.** If we look at the equations above carefully, we will notice that the coordinates for $\mathbf{v}_+, \mathbf{w}_+$ are mutually exclusive from the coordinates for $\mathbf{v}_-, \mathbf{w}_-$. And hence, up to column and row permutations the Hessian can be written as a block-diagonal matrix with Hessian for the individual cells respectively in these diagonal blocks. In fact, the column space itself is a direct sum of the column space across the cells. And, so the eigenvectors will also be non-zero on mutually exclusive coordinates. As a result, we can solve the Hessian spectrum separately for the two cells.

We can also express the above in the matrix form rather compactly as follows:

$$\frac{n_+}{n}\mathbf{H_L}^+ = \begin{matrix} \frac{\partial}{\partial\mathbf{v}_+} \\ \frac{\partial}{\partial\mathbf{w}_+} \end{matrix} \overset{\begin{matrix}\frac{\partial}{\partial\mathbf{v}_+^\top} & \frac{\partial}{\partial\mathbf{w}_+^\top}\end{matrix}}{\begin{pmatrix} \frac{n_+}{n}\,\sigma^2_{x+}\,\mathbf{w}_+\mathbf{w}_+^\top & \frac{n_+}{n}\,\sigma^2_{x+}\,\mathbf{w}_+\mathbf{v}_+^\top + \frac{n_+}{n}\,\overline{x\delta}_+\mathbf{I}_q \\ \frac{n_+}{n}\,\sigma^2_{x+}\,\mathbf{v}_+\mathbf{w}_+^\top + \frac{n_+}{n}\,\overline{x\delta}_+\mathbf{I}_q & \frac{n_+}{n}\,\sigma^2_{x+}\,\mathbf{v}_+\mathbf{v}_+^\top \end{pmatrix}}$$

where, $\overline{x\delta}_+ = \sigma^2_{x+}\,\langle\mathbf{w}_+,\mathbf{v}_+\rangle - \overline{yx}_+$. Likewise, we have that

$$\frac{n_-}{n}\mathbf{H_L}^- = \begin{matrix} \frac{\partial}{\partial \mathbf{v}_-} \\ \frac{\partial}{\partial \mathbf{w}_-} \end{matrix} \overset{\overset{\frac{\partial}{\partial \mathbf{v}_-^\top}}{}}{\begin{pmatrix} \frac{n_-}{n}\sigma_{x_-}^2\,\mathbf{w}_-\mathbf{w}_-^\top & \frac{n_-}{n}\sigma_{x_-}^2\,\mathbf{w}_-\mathbf{v}_-^\top + \frac{n_-}{n}\,\overline{x\delta}_-\,\mathbf{I}_{m-q} \\ \frac{n_-}{n}\sigma_{x_-}^2\,\mathbf{v}_-\mathbf{w}_-^\top + \frac{n_-}{n}\,\overline{x\delta}_-\,\mathbf{I}_{m-q} & \frac{n_-}{n}\sigma_{x_-}^2\,\mathbf{v}_-\mathbf{v}_-^\top \end{pmatrix}}$$

To emphasize, the Hessian is equivalent to:

$$\mathbf{H_L} = \begin{pmatrix} \frac{n_+}{n}\mathbf{H_L}^+ & \mathbf{0} \\ \mathbf{0} & \frac{n_-}{n}\mathbf{H_L}^- \end{pmatrix}$$

Essentially, the row and column permutations correspond to a different ordering of the parameters when forming the Hessian matrix.

Let us assume that $q$ coordinates of the parameter vector $\mathbf{v}$ are positive. Then we have that the spectrum in this ReLU case is a corollary of the Hessian decoupling result and the eigenvalues in the linear case.

When it comes to eigenvectors, we simply follow our strategy for the linear case, and since the Hessian for ReLU decouples into that along the positive and negative cell, we obtain the following set of eigenvectors for the outlier and bulk eigenvalues corresponding to the respective cells.

Let us briefly recall our shorthand from before, $\mathbf{w}_+ = \mathbf{w}\odot\mathbb{1}\{\mathbf{v} > 0\}$, $\mathbf{w}_- = \mathbf{w}\odot\mathbb{1}\{\mathbf{v} \le 0\}$, $\mathbf{v}_+ = \mathbf{v}\odot\mathbb{1}\{\mathbf{v} > 0\}$ and $\mathbf{v}_- = \mathbf{v}\odot\mathbb{1}\{\mathbf{v} \le 0\}$.

**Positive Cell Eigenvectors.** The outlier eigenvectors are given by,

$$\mathbf{z}_+ = \frac{\alpha}{\overline{x\delta}_+^2 - \lambda_+^2}\begin{pmatrix} \lambda_{\text{outlier}}^+\mathbf{w}_+ + \overline{x\delta}_+\mathbf{v}_+ \\ \overline{x\delta}_+\mathbf{w}_+ + \lambda_{\text{outlier}}^+\mathbf{v}_+ \end{pmatrix} \tag{62}$$

with,

$$\lambda_{\text{outlier}}^+ = \frac{n_+}{2n}(\sigma_+^2\|\mathbf{w}_+\|^2 + \sigma_+^2\|\mathbf{v}_+\|^2) \tag{63}$$

$$\pm \frac{n_+}{2n}\sqrt{(\sigma_+^2\|\mathbf{w}_+\|^2 + \sigma_+^2\|\mathbf{v}_+\|^2)^2 + 4(\overline{x\delta}_+^2 + 2\sigma_+^2\overline{x\delta}_+\langle\mathbf{w}_+, \mathbf{v}_+\rangle)}. \tag{64}$$

The bulk eigenvectors depend whether their eigenvalue is $\lambda_{\text{bulk}}^+ = \overline{x\delta}_+$ or $\lambda_{\text{bulk}}^+ = -\overline{x\delta}_+$. In the former case, we have eigenvectors of the form $\begin{pmatrix} \mathbf{z}_1 \\ \mathbf{z}_2 \end{pmatrix}$, where $\mathbf{z}_1 = \mathbf{z}_2$ is of the form $\left(\mathbf{I} - \frac{(\mathbf{w}_+ + \mathbf{v}_+)(\mathbf{w}_+ + \mathbf{v}_+)^\top}{\|\mathbf{w}_+ + \mathbf{v}_+\|^2}\right)\mathbf{c}$ for some vector $\mathbf{c}$, and they can simply be obtained by computing the eigenvectors of this matrix, corresponding to non-zero eigenvalues. In the latter case, we have that $\mathbf{z}_1 = -\mathbf{z}_2$ with, $\mathbf{z}_1$ of the form $\left(\mathbf{I} - \frac{(\mathbf{w}_+ - \mathbf{v}_+)(\mathbf{w}_+ - \mathbf{v}_+)^\top}{\|\mathbf{w}_+ - \mathbf{v}_+\|^2}\right)\mathbf{c}$ for some vector $\mathbf{c}$.

**Negative Cell Eigenvectors.** The outlier eigenvectors are given by,

$$\mathbf{z}_- = \frac{\alpha}{\overline{x\delta}_-^2 - \lambda_-^2}\begin{pmatrix} \lambda_{\text{outlier}}^-\mathbf{w}_- + \overline{x\delta}_-\mathbf{v}_- \\ \overline{x\delta}_-\mathbf{w}_- + \lambda_{\text{outlier}}^-\mathbf{v}_- \end{pmatrix} \tag{65}$$

with,

$$\lambda_{\text{outlier}}^- = \frac{n_-}{2n}(\sigma_-^2\|\mathbf{w}_-\|^2 + \sigma_-^2\|\mathbf{v}_-\|^2) \tag{66}$$

$$\pm \frac{n_-}{2n}\sqrt{(\sigma_-^2\|\mathbf{w}_-\|^2 + \sigma_-^2\|\mathbf{v}_-\|^2)^2 + 4(\overline{x\delta}_-^2 + 2\sigma_-^2\overline{x\delta}_-\langle\mathbf{w}_-, \mathbf{v}_-\rangle)}. \tag{67}$$

The bulk eigenvectors depend whether their eigenvalue is $\lambda_{\text{bulk}}^- = \overline{x\delta_-}$ or $\lambda_{\text{bulk}}^- = -\overline{x\delta_-}$. In the former case, we have eigenvectors of the form $\begin{pmatrix} \mathbf{z}_1 \\ \mathbf{z}_2 \end{pmatrix}$, where $\mathbf{z}_1 = \mathbf{z}_2$ is of the form $\left( \mathbf{I} - \frac{(\mathbf{w}_- + \mathbf{v}_-)(\mathbf{w}_- + \mathbf{v}_-)^\top}{\|\mathbf{w}_- + \mathbf{v}_-\|^2} \right)\mathbf{c}$ for some vector $\mathbf{c}$, and they can simply be obtained by computing the eigenvectors of this matrix, corresponding to non-zero eigenvalues. In the latter case, we have that $\mathbf{z}_1 = -\mathbf{z}_2$ with, $\mathbf{z}_1$ of the form $\left( \mathbf{I} - \frac{(\mathbf{w}_- - \mathbf{v}_-)(\mathbf{w}_- - \mathbf{v}_-)^\top}{\|\mathbf{w}_- - \mathbf{v}_-\|^2} \right)\mathbf{c}$ for some vector $\mathbf{c}$. $\qquad\square$

### B.3 LINEAR NETWORK WITH BIAS

**Theorem B.1.** For a linear network with bias given by $f(x) = \mathbf{w}^\top(\mathbf{v}x + \mathbf{b})$, assume that $\langle \mathbf{v}, \mathbf{b} \rangle = 0$ and $\|\mathbf{v}\| = \|\mathbf{b}\|$, as well as $\sigma_x^2 = 1$ and zero-mean data $\mathbb{E}[x] = 0, \mathbb{E}[y] = 0$ and $\overline{yx} = 0$. Then the spectrum consists of the following sets of outlier eigenvalues, the first being the following pair,

$$\lambda = \frac{1}{2}(\|\mathbf{v}\|^2 + \|\mathbf{w}\|^2) \pm \frac{1}{2}\sqrt{(\|\mathbf{v}\|^2 + \|\mathbf{w}\|^2)^2 + 12(\overline{\delta x}^2 + \overline{\delta}^2)} \tag{68}$$

and the second is the triple $\lambda_k = t_k + (\|\mathbf{v}\|^2 + \|\mathbf{w}\|^2)/3$, where $t_k$ for $k = 0, 1, 2$:

$$t_k = 2\sqrt{-\frac{p}{3}}\cos\left( \frac{1}{3}\arccos\left( \frac{3q}{2p}\sqrt{\frac{-3}{p}} \right) - k\frac{2\pi}{3} \right)$$

with, $p = -(\overline{\delta x}^2 + \overline{\delta}^2) - (\|\mathbf{v}\|^2 + \|\mathbf{w}\|^2)^2/3$ and $q = -\frac{2}{27}(\|\mathbf{v}\|^2 + \|\mathbf{w}\|^2)^3 - \frac{1}{3}\|\mathbf{v}\|^2(\overline{\delta x}^2 + \overline{\delta}^2) + \frac{2}{3}\|\mathbf{w}\|^2(\overline{\delta x}^2 + \overline{\delta}^2)$ and the rest being the bulk eigenvalues $\lambda_{\text{bulk}} = \pm\sqrt{\overline{\delta x}^2 + \overline{\delta}^2}$ and $0$ eigenvalues.

*Proof.*
Let us start with the linear case and assume that our network is now given by,

$$f(\mathbf{x}) = \mathbf{w}^\top(\mathbf{v}x + \mathbf{b}).$$

The expressions of the loss gradient are:

$$\nabla_{\mathbf{w}}\ell = \delta \cdot (\mathbf{v}x + \mathbf{b}) \tag{69}$$
$$\nabla_{\mathbf{v}}\ell = \delta \cdot \mathbf{w}x \tag{70}$$
$$\nabla_{\mathbf{b}}\ell = \delta \cdot \mathbf{w} \tag{71}$$

where, $\delta = f(\mathbf{x}) - y = \mathbf{w}^\top(\mathbf{v}x + \mathbf{b}) - y$. The Hessian terms are then as follows[5]:

$$\nabla_{\mathbf{w},\mathbf{w}}^2\ell = (\mathbf{v}x + \mathbf{b})(\mathbf{v}x + \mathbf{b})^\top \tag{72}$$
$$\nabla_{\mathbf{w},\mathbf{v}}^2\ell = \delta x\mathbf{I}_m + x(\mathbf{v}x + \mathbf{b})\mathbf{w}^\top \tag{73}$$
$$\nabla_{\mathbf{w},\mathbf{b}}^2\ell = \delta\mathbf{I}_m + (\mathbf{v}x + \mathbf{b})\mathbf{w}^\top \tag{74}$$
$$\nabla_{\mathbf{v},\mathbf{v}}^2\ell = x^2\mathbf{w}\mathbf{w}^\top, \quad \nabla_{\mathbf{v},\mathbf{b}}^2\ell = x\mathbf{w}\mathbf{w}^\top \tag{75}$$
$$\nabla_{\mathbf{b},\mathbf{b}}^2\ell = \mathbf{w}\mathbf{w}^\top \tag{76}$$

Next, let us aggregate the above expression over the entire dataset, with the assumption that the data is centered (which is anyways carried out in practice), i.e., $\mathbb{E}[x] = 0$, yielding :

---

[5]The term $\nabla_{\mathbf{v},\mathbf{b}}^2\ell = x\mathbf{w}\mathbf{w}^\top$ can be generalized to the case of having a 2-dimensional input with coordinates $x_1, x_2$ with $x_2 \neq 1$ and then we would get $\nabla_{\mathbf{v}_1,\mathbf{v}_2}^2\ell = x_1 x_2\mathbf{w}\mathbf{w}^\top$.

$$\nabla^2_{\mathbf{w},\mathbf{w}}\mathrm{L} = \sigma^2\mathbf{v}\mathbf{v}^\top + \mathbf{b}\mathbf{b}^\top \tag{77}$$

$$\nabla^2_{\mathbf{w},\mathbf{v}}\mathrm{L} = \overline{\delta x}\mathbf{I}_m + \sigma^2\mathbf{v}\mathbf{w}^\top \tag{78}$$

$$\nabla^2_{\mathbf{w},\mathbf{b}}\mathrm{L} = \overline{\delta}\mathbf{I}_m + \mathbf{b}\mathbf{w}^\top \tag{79}$$

$$\nabla^2_{\mathbf{v},\mathbf{v}}\mathrm{L} = \sigma^2\mathbf{w}\mathbf{w}^\top, \quad \nabla^2_{\mathbf{v},\mathbf{b}}\mathrm{L} = \mathbf{0} \tag{80}$$

$$\nabla^2_{\mathbf{b},\mathbf{b}}\mathrm{L} = \mathbf{w}\mathbf{w}^\top \tag{81}$$

In summary, the Hessian can be written as:

$$\mathbf{H}_{\mathrm{L}} = \begin{pmatrix} \sigma^2\mathbf{v}\mathbf{v}^\top + \mathbf{b}\mathbf{b}^\top & \overline{\delta x}\mathbf{I}_m + \sigma^2\mathbf{v}\mathbf{w}^\top & \overline{\delta}\mathbf{I}_m + \mathbf{b}\mathbf{w}^\top \\ \overline{\delta x}\mathbf{I}_m + \sigma^2\mathbf{w}\mathbf{v}^\top & \sigma^2\mathbf{w}\mathbf{w}^\top & \mathbf{0} \\ \overline{\delta}\mathbf{I}_m + \mathbf{w}\mathbf{b}^\top & \mathbf{0} & \mathbf{w}\mathbf{w}^\top \end{pmatrix} \tag{82}$$

For starters, assume $\sigma^2 = 1$. Then we need to solve the characteristic equation, i.e.,

$$\left| \begin{pmatrix} \mathbf{v}\mathbf{v}^\top + \mathbf{b}\mathbf{b}^\top - \lambda\mathbf{I}_m & \overline{\delta x}\mathbf{I}_m + \mathbf{v}\mathbf{w}^\top & \overline{\delta}\mathbf{I}_m + \mathbf{b}\mathbf{w}^\top \\ \overline{\delta x}\mathbf{I}_m + \mathbf{w}\mathbf{v}^\top & \mathbf{w}\mathbf{w}^\top - \lambda\mathbf{I}_m & \mathbf{0} \\ \overline{\delta}\mathbf{I}_m + \mathbf{w}\mathbf{b}^\top & \mathbf{0} & \mathbf{w}\mathbf{w}^\top - \lambda\mathbf{I}_m \end{pmatrix} \right| = 0 \tag{83}$$

Instead of computing the determinant using the Schur complement, let us first rewrite the matrix above in a simpler form:[6]

$$\mathbf{H}_{\mathrm{L}} - \lambda\mathbf{I}_{3m} = \begin{pmatrix} \mathbf{v} \\ \mathbf{w} \\ \mathbf{0} \end{pmatrix}\begin{pmatrix} \mathbf{v} \\ \mathbf{w} \\ \mathbf{0} \end{pmatrix}^\top + \begin{pmatrix} \mathbf{b} \\ \mathbf{0} \\ \mathbf{w} \end{pmatrix}\begin{pmatrix} \mathbf{b} \\ \mathbf{0} \\ \mathbf{w} \end{pmatrix}^\top + \begin{pmatrix} -\lambda & \overline{\delta x} & \overline{\delta} \\ \overline{\delta x} & -\lambda & 0 \\ \overline{\delta} & 0 & -\lambda \end{pmatrix} \otimes \mathbf{I}_m \tag{84}$$

$$= \begin{pmatrix} \mathbf{v} & \mathbf{b} \\ \mathbf{w} & \mathbf{0} \\ \mathbf{0} & \mathbf{w} \end{pmatrix}\begin{pmatrix} \mathbf{v} & \mathbf{b} \\ \mathbf{w} & \mathbf{0} \\ \mathbf{0} & \mathbf{w} \end{pmatrix}^\top + \begin{pmatrix} -\lambda & \overline{\delta x} & \overline{\delta} \\ \overline{\delta x} & -\lambda & 0 \\ \overline{\delta} & 0 & -\lambda \end{pmatrix} \otimes \mathbf{I}_m \tag{85}$$

Then using the fact that $|\mathbf{A}_{m\times m} + \mathbf{U}_{m\times n}\mathbf{V}^\top_{n\times m}| = |\mathbf{A}| \cdot |\mathbf{I}_n + \mathbf{V}^\top_{n\times m}\mathbf{A}^{-1}\mathbf{U}_{m\times n}|$. Let's apply this to the above matrix, with $\mathbf{A} = \begin{pmatrix} -\lambda & \overline{\delta x} & \overline{\delta} \\ \overline{\delta x} & -\lambda & 0 \\ \overline{\delta} & 0 & -\lambda \end{pmatrix} \otimes \mathbf{I}_m$ and $\mathbf{U} = \mathbf{V} = \begin{pmatrix} \mathbf{v} & \mathbf{b} \\ \mathbf{w} & \mathbf{0} \\ \mathbf{0} & \mathbf{w} \end{pmatrix}$. First, by using the fact $(\mathbf{C} \otimes \mathbf{D})^{-1} = \mathbf{C}^{-1} \otimes \mathbf{D}^{-1}$, we have that

$$\mathbf{A}^{-1} = -\frac{1}{\lambda\left(\lambda^2 - (\overline{\delta x}^2 + \overline{\delta}^2)\right)}\begin{pmatrix} \lambda^2 & \lambda\overline{\delta x} & \lambda\overline{\delta} \\ \lambda\overline{\delta x} & \lambda^2 - \overline{\delta}^2 & \overline{\delta}\,\overline{\delta x} \\ \lambda\overline{\delta} & \overline{\delta}\,\overline{\delta x} & \lambda^2 - \overline{\delta x}^2 \end{pmatrix} \otimes \mathbf{I}_m \tag{86}$$

Next, $\mathbf{A}^{-1}\mathbf{V}$ comes out to:

$$-\frac{1}{\lambda\left(\lambda^2 - (\overline{\delta x}^2 + \overline{\delta}^2)\right)}\begin{pmatrix} \lambda^2\mathbf{v} + \lambda\overline{\delta x}\mathbf{w} & \lambda^2\mathbf{b} + \lambda\overline{\delta}\mathbf{w} \\ \lambda\overline{\delta x}\mathbf{v} + (\lambda^2 - \overline{\delta}^2)\mathbf{w} & \lambda\overline{\delta x}\mathbf{b} + \overline{\delta}\,\overline{\delta x}\mathbf{w} \\ \lambda\overline{\delta}\mathbf{v} + \overline{\delta}\,\overline{\delta x}\mathbf{w} & \lambda\overline{\delta}\mathbf{b} + (\lambda^2 - \overline{\delta x}^2)\mathbf{w} \end{pmatrix} \tag{87}$$

Further, the quadratic form $\mathbf{V}^\top\mathbf{A}^{-1}\mathbf{V}$ is given by $-\frac{1}{\lambda\left(\lambda^2 - (\overline{\delta x}^2 + \overline{\delta}^2)\right)}\mathbf{B}$, with $\mathbf{B}$ being:

$$\mathbf{B} = \begin{pmatrix} (\|\mathbf{v}\|^2 + \|\mathbf{w}\|^2)\lambda^2 + 2\overline{\delta x}\langle\mathbf{w},\mathbf{v}\rangle\lambda - \overline{\delta}^2\|\mathbf{w}\|^2 & \langle\mathbf{v},\mathbf{b}\rangle\lambda^2 + \overline{\delta}\langle\mathbf{v},\mathbf{w}\rangle\lambda + \overline{\delta x}\langle\mathbf{b},\mathbf{w}\rangle\lambda + \overline{\delta}\,\overline{\delta x}\|\mathbf{w}\|^2 \\ \langle\mathbf{v},\mathbf{b}\rangle\lambda^2 + \overline{\delta}\langle\mathbf{v},\mathbf{w}\rangle\lambda + \overline{\delta x}\langle\mathbf{b},\mathbf{w}\rangle\lambda + \overline{\delta}\,\overline{\delta x}\|\mathbf{w}\|^2 & (\|\mathbf{b}\|^2 + \|\mathbf{w}\|^2)\lambda^2 + 2\overline{\delta}\langle\mathbf{w},\mathbf{b}\rangle\lambda - \overline{\delta x}^2\|\mathbf{w}\|^2 \end{pmatrix} \tag{88}$$

---

[6]If this does not work, I could start with assuming that the $\mathbf{v}$ and $\mathbf{b}$ are orthogonal.

Then, $|\mathbf{I}_2 - \frac{1}{\lambda\left(\lambda^2 - (\overline{\delta x}^2 + \overline{\delta}^2)\right)}\mathbf{B}| = \frac{(-1)^2}{\lambda^2\left(\lambda^2 - (\overline{\delta x}^2 + \overline{\delta}^2)\right)^2}|\mathbf{B} - \lambda\left(\lambda^2 - (\overline{\delta x}^2 + \overline{\delta}^2)\right)\mathbf{I}_2|$, and the determinant of the remaining $2 \times 2$ matrix is given by:

$$\left((\|\mathbf{v}\|^2 + \|\mathbf{w}\|^2)\,\lambda^2 + 2\overline{\delta x}\langle\mathbf{w},\mathbf{v}\rangle\lambda - \overline{\delta}^2\|\mathbf{w}\|^2 - \lambda\left(\lambda^2 - (\overline{\delta x}^2 + \overline{\delta}^2)\right)\right) \times$$

$$\left((\|\mathbf{b}\|^2 + \|\mathbf{w}\|^2)\,\lambda^2 + 2\overline{\delta}\langle\mathbf{w},\mathbf{b}\rangle\lambda - \overline{\delta x}^2\|\mathbf{w}\|^2 - \lambda\left(\lambda^2 - (\overline{\delta x}^2 + \overline{\delta}^2)\right)\right)$$

$$- \left(\langle\mathbf{v},\mathbf{b}\rangle\lambda^2 + \overline{\delta}\langle\mathbf{v},\mathbf{w}\rangle\lambda + \overline{\delta x}\langle\mathbf{b},\mathbf{w}\rangle\lambda + \overline{\delta}\;\overline{\delta x}\|\mathbf{w}\|^2\right)^2$$

In the case when $\sigma^2 = 1$, we have that $\overline{\delta x} = \langle\mathbf{w},\mathbf{v}\rangle - \overline{yx}$ and $\overline{\delta} = \langle\mathbf{w},\mathbf{b}\rangle - \overline{y}$.

Expanding the above equation and setting it to zero yields the following:

$$\lambda^6 + (\overline{\delta x}^2 + \overline{\delta}^2)^2\,\lambda^2 - 2(\overline{\delta x}^2 + \overline{\delta}^2)\lambda^4 - (\|\mathbf{v}\|^2 + \|\mathbf{b}\|^2 + 2\|\mathbf{w}\|^2)\lambda^5 \tag{89}$$

$$- 2(\overline{\delta x}\langle\mathbf{w},\mathbf{v}\rangle + \overline{\delta}\langle\mathbf{w},\mathbf{b}\rangle)\lambda^4 + (\overline{\delta x}^2 + \overline{\delta}^2)(\|\mathbf{v}\|^2 + \|\mathbf{b}\|^2 + 3\|\mathbf{w}\|^2)\,\lambda^3 \tag{90}$$

$$+ 2(\overline{\delta x}^2 + \overline{\delta}^2)(\overline{\delta x}\langle\mathbf{w},\mathbf{v}\rangle + \overline{\delta}\langle\mathbf{w},\mathbf{b}\rangle)\lambda^2 - (\overline{\delta x}^2 + \overline{\delta}^2)^2\|\mathbf{w}\|^2\lambda \tag{91}$$

$$+ (\|\mathbf{v}\|^2 + \|\mathbf{w}\|^2)(\|\mathbf{b}\| + \|\mathbf{w}\|^2)\lambda^4 + 4\overline{\delta x}\,\overline{\delta}\langle\mathbf{w},\mathbf{v}\rangle\langle\mathbf{w},\mathbf{b}\rangle\lambda^2 \cancel{+ \overline{\delta x}^2\overline{\delta}^2\|\mathbf{w}\|^4} \tag{92}$$

$$+ 2\overline{\delta}\langle\mathbf{w},\mathbf{b}\rangle(\|\mathbf{v}\|^2 + \|\mathbf{w}\|^2)\lambda^3 + 2\overline{\delta x}\langle\mathbf{w},\mathbf{v}\rangle(\|\mathbf{b}\|^2 + \|\mathbf{w}\|^2)\lambda^3 \tag{93}$$

$$- 2\overline{\delta x}^3\langle\mathbf{w},\mathbf{v}\rangle\|\mathbf{w}\|^2\lambda - 2\overline{\delta}^3\langle\mathbf{w},\mathbf{b}\rangle\|\mathbf{w}\|^2\lambda \tag{94}$$

$$- (\|\mathbf{v}\|^2 + \|\mathbf{w}\|^2)\|\mathbf{w}\|^2\overline{\delta x}^2\lambda^2 - (\|\mathbf{b}\|^2 + \|\mathbf{w}\|^2)\|\mathbf{w}\|^2\overline{\delta}^2\lambda^2 \tag{95}$$

$$- (\langle\mathbf{v},\mathbf{b}\rangle^2\lambda^4 + \langle\overline{\delta}\mathbf{v} + \overline{\delta x}\mathbf{b},\mathbf{w}\rangle^2\lambda^2 \cancel{+ \overline{\delta}^2\overline{\delta x}^2\|\mathbf{w}\|^4} \tag{96}$$

$$+ 2\langle\mathbf{v},\mathbf{b}\rangle\langle\overline{\delta}\mathbf{v} + \overline{\delta x}\mathbf{b},\mathbf{w}\rangle\lambda^3 + 2\langle\mathbf{v},\mathbf{b}\rangle\overline{\delta}\;\overline{\delta x}\|\mathbf{w}\|^2\lambda^2 \tag{97}$$

$$+ 2\langle\overline{\delta}\mathbf{v} + \overline{\delta x}\mathbf{b},\mathbf{w}\rangle\overline{\delta}\;\overline{\delta x}\|\mathbf{w}\|^2\lambda) \tag{98}$$

$$= 0 \tag{99}$$

The constant terms get cancelled, and rewriting the above expression by collecting the coefficients of various degree terms separately gives us:

$$\lambda^6 - (\|\mathbf{v}\|^2 + \|\mathbf{b}\|^2 + 2\|\mathbf{w}\|^2)\lambda^5$$

$$+ \left[-2(\overline{\delta x}^2 + \overline{\delta}^2) - 2(\overline{\delta x}\langle\mathbf{w},\mathbf{v}\rangle + \overline{\delta}\langle\mathbf{w},\mathbf{b}\rangle) + (\|\mathbf{v}\|^2 + \|\mathbf{w}\|^2)(\|\mathbf{b}\| + \|\mathbf{w}\|^2) - \langle\mathbf{v},\mathbf{b}\rangle^2\right]\lambda^4$$

$$+ \left[(\overline{\delta x}^2 + \overline{\delta}^2)(\|\mathbf{v}\|^2 + \|\mathbf{b}\|^2 + 3\|\mathbf{w}\|^2) + 2\overline{\delta}\langle\mathbf{w},\mathbf{b}\rangle(\|\mathbf{v}\|^2 + \|\mathbf{w}\|^2) + 2\overline{\delta x}\langle\mathbf{w},\mathbf{v}\rangle(\|\mathbf{b}\|^2 + \|\mathbf{w}\|^2)\right]\lambda^3$$

$$- \left[2\langle\mathbf{v},\mathbf{b}\rangle\langle\overline{\delta}\mathbf{v} + \overline{\delta x}\mathbf{b},\mathbf{w}\rangle\right]\lambda^3$$

$$+ \left[(\overline{\delta x}^2 + \overline{\delta}^2)^2 + 2(\overline{\delta x}^2 + \overline{\delta}^2)(\overline{\delta x}\langle\mathbf{w},\mathbf{v}\rangle + \overline{\delta}\langle\mathbf{w},\mathbf{b}\rangle) + 4\overline{\delta x}\,\overline{\delta}\langle\mathbf{w},\mathbf{v}\rangle\langle\mathbf{w},\mathbf{b}\rangle\right]\lambda^2$$

$$- \left[(\|\mathbf{v}\|^2 + \|\mathbf{w}\|^2)\|\mathbf{w}\|^2\overline{\delta x}^2 + (\|\mathbf{b}\|^2 + \|\mathbf{w}\|^2)\|\mathbf{w}\|^2\overline{\delta}^2 + \langle\overline{\delta}\mathbf{v} + \overline{\delta x}\mathbf{b},\mathbf{w}\rangle^2 + 2\langle\mathbf{v},\mathbf{b}\rangle\overline{\delta}\;\overline{\delta x}\|\mathbf{w}\|^2\right]\lambda^2$$

$$- \left[(\overline{\delta x}^2 + \overline{\delta}^2)^2\|\mathbf{w}\|^2 + 2\overline{\delta x}^3\langle\mathbf{w},\mathbf{v}\rangle\|\mathbf{w}\|^2 + 2\overline{\delta}^3\langle\mathbf{w},\mathbf{b}\rangle\|\mathbf{w}\|^2 + 2\langle\overline{\delta}\mathbf{v} + \overline{\delta x}\mathbf{b},\mathbf{w}\rangle\overline{\delta}\;\overline{\delta x}\|\mathbf{w}\|^2\right]\lambda^1$$

$$= 0$$

Let us try to first solve the above equation in a simpler setting. To do so, we assume: $\overline{yx} = 0$ and $\overline{y} = 0$. This means that: $\overline{\delta x} = \langle\mathbf{w},\mathbf{v}\rangle$ and $\overline{\delta} = \langle\mathbf{w},\mathbf{b}\rangle$. We get the following simplified form:

$$\lambda^6 - (\|\mathbf{v}\|^2 + \|\mathbf{b}\|^2 + 2\|\mathbf{w}\|^2)\lambda^5$$

$$+ \left[ -2(\overline{\delta x}^2 + \overline{\delta}^2) - 2(\overline{\delta x}^2 + \overline{\delta}^2) + (\|\mathbf{v}\|^2 + \|\mathbf{w}\|^2)(\|\mathbf{b}\| + \|\mathbf{w}\|^2) - \langle \mathbf{v}, \mathbf{b}\rangle^2 \right] \lambda^4$$

$$+ \left[ (\overline{\delta x}^2 + \overline{\delta}^2)(\|\mathbf{v}\|^2 + \|\mathbf{b}\|^2 + 3\|\mathbf{w}\|^2) + 2\overline{\delta}^2(\|\mathbf{v}\|^2 + \|\mathbf{w}\|^2) + 2\overline{\delta x}^2(\|\mathbf{b}\|^2 + \|\mathbf{w}\|^2) \right] \lambda^3$$

$$- \left[ 4\langle \mathbf{v}, \mathbf{b}\rangle \,\overline{\delta x}\,\overline{\delta} \right] \lambda^3$$

$$+ \left[ (\overline{\delta x}^2 + \overline{\delta}^2)^2 + 2(\overline{\delta x}^2 + \overline{\delta}^2)^2 + 4\overline{\delta x}^2\overline{\delta}^2 \right] \lambda^2$$

$$- \left[ (\|\mathbf{v}\|^2 + \|\mathbf{w}\|^2)\|\mathbf{w}\|^2\overline{\delta x}^2 + (\|\mathbf{b}\|^2 + \|\mathbf{w}\|^2)\|\mathbf{w}\|^2\overline{\delta}^2 + 4\overline{\delta x}^2\overline{\delta}^2 + 2\langle \mathbf{v}, \mathbf{b}\rangle\overline{\delta}\,\overline{\delta x}\|\mathbf{w}\|^2 \right] \lambda^2$$

$$- \left[ (\overline{\delta x}^2 + \overline{\delta}^2)^2\|\mathbf{w}\|^2 + 2\overline{\delta x}^4\|\mathbf{w}\|^2 + 2\overline{\delta}^4\|\mathbf{w}\|^2 + 4\overline{\delta}^2\,\overline{\delta x}^2\|\mathbf{w}\|^2 \right] \lambda^1$$

$$= 0$$

which can be further simplified as,

$$\lambda^6 - (\|\mathbf{v}\|^2 + \|\mathbf{b}\|^2 + 2\|\mathbf{w}\|^2)\lambda^5$$

$$+ \left[ -4(\overline{\delta x}^2 + \overline{\delta}^2) + (\|\mathbf{v}\|^2 + \|\mathbf{w}\|^2)(\|\mathbf{b}\| + \|\mathbf{w}\|^2) - \langle \mathbf{v}, \mathbf{b}\rangle^2 \right] \lambda^4$$

$$+ \left[ (\overline{\delta x}^2 + \overline{\delta}^2)(\|\mathbf{v}\|^2 + \|\mathbf{b}\|^2 + 3\|\mathbf{w}\|^2) + 2\overline{\delta}^2(\|\mathbf{v}\|^2 + \|\mathbf{w}\|^2) + 2\overline{\delta x}^2(\|\mathbf{b}\|^2 + \|\mathbf{w}\|^2) - 4\langle \mathbf{v}, \mathbf{b}\rangle \,\overline{\delta x}\,\overline{\delta} \right] \lambda^3$$

$$+ \left[ 3(\overline{\delta x}^2 + \overline{\delta}^2)^2 - (\|\mathbf{v}\|^2 + \|\mathbf{w}\|^2)\|\mathbf{w}\|^2\overline{\delta x}^2 - (\|\mathbf{b}\|^2 + \|\mathbf{w}\|^2)\|\mathbf{w}\|^2\overline{\delta}^2 - 2\langle \mathbf{v}, \mathbf{b}\rangle\overline{\delta}\,\overline{\delta x}\|\mathbf{w}\|^2 \right] \lambda^2$$

$$- 3(\overline{\delta x}^2 + \overline{\delta}^2)^2\|\mathbf{w}\|^2\lambda^1$$

$$= 0$$

We effectively have a quintic equation above and quintic equations, in general, do not have roots in radicals due to the Abel-Ruffini theorem. Although this specific equation might still be solvable in radicals. Let's assume that the underlying quintic is of the form:

$$(\lambda^2 + A\lambda + B)(\lambda^3 + C\lambda^2 + D\lambda + E)$$

which we can be expressed as:

$$\lambda^5 + (A + C)\lambda^4 + (B + AC + D)\lambda^3 + (E + AD + BC)\lambda^2 + (AE + BD)\lambda + BE$$

Matching this with the above equation with $\lambda$ factored out, we get the following series of equations:

$$A + C = -(\|\mathbf{v}\|^2 + \|\mathbf{b}\|^2 + 2\|\mathbf{w}\|^2)$$

$$B + AC + D = -4(\overline{\delta x}^2 + \overline{\delta}^2) + (\|\mathbf{v}\|^2 + \|\mathbf{w}\|^2)(\|\mathbf{b}\| + \|\mathbf{w}\|^2) - \langle \mathbf{v}, \mathbf{b}\rangle^2$$

$$E + AD + BC = (\overline{\delta x}^2 + \overline{\delta}^2)(\|\mathbf{v}\|^2 + \|\mathbf{b}\|^2 + 3\|\mathbf{w}\|^2) + 2\overline{\delta}^2(\|\mathbf{v}\|^2 + \|\mathbf{w}\|^2) + 2\overline{\delta x}^2(\|\mathbf{b}\|^2 + \|\mathbf{w}\|^2) - 4\langle \mathbf{v}, \mathbf{b}\rangle \,\overline{\delta x}\,\overline{\delta}$$

$$AE + BD = 3(\overline{\delta x}^2 + \overline{\delta}^2)^2 - (\|\mathbf{v}\|^2 + \|\mathbf{w}\|^2)\|\mathbf{w}\|^2\overline{\delta x}^2 - (\|\mathbf{b}\|^2 + \|\mathbf{w}\|^2)\|\mathbf{w}\|^2\overline{\delta}^2 - 2\langle \mathbf{v}, \mathbf{b}\rangle\overline{\delta}\,\overline{\delta x}\|\mathbf{w}\|^2$$

$$BE = -3(\overline{\delta x}^2 + \overline{\delta}^2)^2\|\mathbf{w}\|^2$$

And to remind again the above quintic is:

$$\lambda^5 - (\|\mathbf{v}\|^2 + \|\mathbf{b}\|^2 + 2\|\mathbf{w}\|^2)\lambda^4$$

$$+ \left[ -4(\overline{\delta x}^2 + \overline{\delta}^2) + (\|\mathbf{v}\|^2 + \|\mathbf{w}\|^2)(\|\mathbf{b}\| + \|\mathbf{w}\|^2) - \langle \mathbf{v}, \mathbf{b} \rangle^2 \right] \lambda^3$$

$$+ \left[ (\overline{\delta x}^2 + \overline{\delta}^2)(\|\mathbf{v}\|^2 + \|\mathbf{b}\|^2 + 3\|\mathbf{w}\|^2) + 2\overline{\delta}^2(\|\mathbf{v}\|^2 + \|\mathbf{w}\|^2) + 2\overline{\delta x}^2(\|\mathbf{b}\|^2 + \|\mathbf{w}\|^2) - 4\langle \mathbf{v}, \mathbf{b} \rangle \overline{\delta x}\, \overline{\delta} \right] \lambda^2$$

$$+ \left[ 3(\overline{\delta x}^2 + \overline{\delta}^2)^2 - (\|\mathbf{v}\|^2 + \|\mathbf{w}\|^2)\|\mathbf{w}\|^2\overline{\delta x}^2 - (\|\mathbf{b}\|^2 + \|\mathbf{w}\|^2)\|\mathbf{w}\|^2\overline{\delta}^2 - 2\langle \mathbf{v}, \mathbf{b} \rangle \overline{\delta}\, \overline{\delta x}\|\mathbf{w}\|^2 \right] \lambda^1$$

$$- 3(\overline{\delta x}^2 + \overline{\delta}^2)^2\|\mathbf{w}\|^2$$

$$= 0$$

Based on the above equations, let us employ two additional assumptions, i.e., $\langle \mathbf{v}, \mathbf{b} \rangle = 0$ and $\|\mathbf{v}\| = \|\mathbf{b}\|$. We have as the choice of the coefficients::

$$A = -(\|\mathbf{v}\|^2 + \|\mathbf{w}\|^2) \tag{100}$$

$$B = -3(\overline{\delta x}^2 + \overline{\delta}^2) \tag{101}$$

$$C = -(\|\mathbf{v}\|^2 + \|\mathbf{w}\|^2) \tag{102}$$

$$D = -(\overline{\delta x}^2 + \overline{\delta}^2) \tag{103}$$

$$E = (\overline{\delta x}^2 + \overline{\delta}^2)\|\mathbf{w}\|^2 \tag{104}$$

We get as a factored quadratic:

$$\lambda^2 - (\|\mathbf{v}\|^2 + \|\mathbf{w}\|^2)\lambda - 3(\overline{\delta x}^2 + \overline{\delta}^2) = 0$$

$$\lambda = \frac{1}{2}(\|\mathbf{v}\|^2 + \|\mathbf{w}\|^2) \pm \frac{1}{2}\sqrt{(\|\mathbf{v}\|^2 + \|\mathbf{w}\|^2)^2 + 12(\overline{\delta x}^2 + \overline{\delta}^2)} \tag{105}$$

This strictly generalizes the case without bias, where we had the solution:

$$\lambda = \frac{1}{2}(\|\mathbf{v}\|^2 + \|\mathbf{w}\|^2) \pm \frac{1}{2}\sqrt{(\|\mathbf{v}\|^2 + \|\mathbf{w}\|^2)^2 + 12\overline{\delta x}^2} \tag{106}$$

Remember, under our assumptions $\overline{\delta x} = \langle \mathbf{w}, \mathbf{v} \rangle$ and $\overline{\delta} = \langle \mathbf{w}, \mathbf{b} \rangle$. Hence this forms for an interesting generalization in the case with multi-dimensional input, with term in the square root being the sum of squares of the inner-product of the second layer weight $\mathbf{w}$ and the columns of the first layer matrix.

**Solving the cubic.** Other than the solutions from the above quadratic, we should also check for solutions in:

$$\lambda^3 - (\|\mathbf{v}\|^2 + \|\mathbf{w}\|^2)\lambda^2 - (\overline{\delta x}^2 + \overline{\delta}^2)\lambda + (\overline{\delta x}^2 + \overline{\delta}^2)\|\mathbf{w}\|^2 = 0$$

We can put this into depressed form, i.e., $t^3 + pt + q = 0$, by making the substitution:

$$\lambda = t + \frac{(\|\mathbf{v}\|^2 + \|\mathbf{w}\|^2)}{3}.$$

Here, $p$ and $q$ come out to be:

$$p = -(\overline{\delta x}^2 + \overline{\delta}^2) - \frac{(\|\mathbf{v}\|^2 + \|\mathbf{w}\|^2)^2}{3} \tag{107}$$

$$q = \frac{1}{27} \left[ -2(\|\mathbf{v}\|^2 + \|\mathbf{w}\|^2)^3 - 9(\|\mathbf{v}\|^2 + \|\mathbf{w}\|^2)(\overline{\delta x}^2 + \overline{\delta}^2) + 27(\overline{\delta x}^2 + \overline{\delta}^2)\|\mathbf{w}\|^2 \right] \tag{108}$$

The coefficient $q$ can be further simplified as:

$$q = -\frac{2}{27}(\|\mathbf{v}\|^2 + \|\mathbf{w}\|^2)^3 - \frac{1}{3}\|\mathbf{v}\|^2(\overline{\delta x}^2 + \overline{\delta}^2) + \frac{2}{3}\|\mathbf{w}\|^2(\overline{\delta x}^2 + \overline{\delta}^2) \tag{109}$$

Using Viète's trigonometric expression of the roots in three-real roots case, we have that the solutions to $t$ are given by $t_k$ for $k = 0, 1, 2$:

$$t_k = 2\sqrt{-\frac{p}{3}}\cos\left(\frac{1}{3}\arccos\left(\frac{3q}{2p}\sqrt{\frac{-3}{p}}\right) - k\frac{2\pi}{3}\right) \tag{110}$$

Since $cos \in [-1, 1]$, we can upper and lower bound the above solutions to the cubic as:

$$\frac{1}{3}(\|\mathbf{v}\|^2 + \|\mathbf{w}\|^2) - t' \leq \lambda_{0,1,2} \leq \frac{1}{3}(\|\mathbf{v}\|^2 + \|\mathbf{w}\|^2) + t'$$

with $t' = 2\sqrt{\frac{1}{3}(\overline{\delta x}^2 + \overline{\delta}^2) + \frac{1}{9}(\|\mathbf{v}\|^2 + \|\mathbf{w}\|^2)^2} = \frac{2}{3}\sqrt{(\|\mathbf{v}\|^2 + \|\mathbf{w}\|^2)^2 + 3(\overline{\delta x}^2 + \overline{\delta}^2)}$.

Notice, that $\|\mathbf{v}\|^2 + \|\mathbf{w}\|^2$ is $1/2 \cdot \text{tr}(\mathbf{H}_{\text{L}})$. So, the two solutions to the quadratic on the page before are centered at $1/4 \cdot \text{tr}(\mathbf{H}_{\text{L}})$, while the three solutions to the cubic get centered around $1/6 \cdot \text{tr}(\mathbf{H}_{\text{L}})$. $\qquad\square$

### B.4 LINEAR NETWORK BEYOND UNIDIMENSIONAL CASE

#### B.4.1 EXTENSION TO MULTI-DIMENSIONAL INPUTS

We now consider an extension of our closed-form results in the setting of multi-dimensional input $\mathbf{x} \in \mathbb{R}^D$, where such a task still remains amenable. In particular, our network is given as follows:

$$f(\mathbf{x}) = \sum_{d=1}^{D}\langle\mathbf{w}_d, \mathbf{v}_d\rangle x_d. \tag{111}$$

The parameters that are being learned are these weight vectors for each of the coordinates, i.e., $\Theta = \{\mathbf{w}_1, \mathbf{v}_1, \cdots \mathbf{w}_D, \mathbf{v}_D\}$, amounting to a total parameter count $p = 2mD$. Alternatively, we can rewrite the above network as a special case of a one-hidden layer linear network,

$$f(\mathbf{x}) = \mathbf{w}^\top \mathbf{V}\mathbf{x}, \text{ where } \mathbf{w} = \begin{pmatrix}\mathbf{w}_1^\top & \cdots & \mathbf{w}_D^\top\end{pmatrix}^\top \in \mathbb{R}^{mD}, \text{ and } \mathbf{V} = \begin{pmatrix}\mathbf{v}_1 & \cdots & \mathbf{0}\\ \vdots & \ddots & \vdots\\ \mathbf{0} & \cdots & \mathbf{v}_d\end{pmatrix} \in \mathbb{R}^{mD\times D}.$$

But, in essence, this can still be thought of as a regression scenario, $f(\mathbf{x}) = \boldsymbol{\alpha}^\top\mathbf{x}$, where the $d$-th coordinate of $\boldsymbol{\alpha}$ is parameterized as $\boldsymbol{\alpha}_d = \langle\mathbf{w}_d, \mathbf{v}_d\rangle$.

**Corollary B.1.** For the above setting of a multi-dimensional one-hidden layer linear network $f(\mathbf{x}) = \sum_{d=1}^{D}\langle\mathbf{w}_d, \mathbf{v}_d\rangle x_d$ with $2mD$ parameters and where the input has been standardized such that $\sigma_{ij} := \mathbb{E}[x_i x_j] = 0$, the Hessian $\mathbf{H}_{\text{L}}$ spectrum decouples across the parameter groups for each input dimension $\{\mathbf{w}_d, \mathbf{v}_d\}$, thereby resulting in a spectrum which has a set of $2(m-1)D$ bulk eigenvalues $\lambda_{\text{bulk}}^d = \pm\overline{\delta x_d} = \pm(\langle\mathbf{w}_d, \mathbf{v}_d\rangle\sigma_d^2 - \overline{yx_d})$, where each is repeated $m-1$ times with both positive and negative sign, and a set of $2D$ paired outlying eigenvalues defined by the expression:

$$\lambda_{\text{outlier}_{1,2}}^d = \frac{1}{2}(\sigma_d^2\|\mathbf{w}_d\|^2 + \sigma_d^2\|\mathbf{v}_d\|^2) \tag{112}$$

$$\pm \frac{1}{2}\sqrt{(\sigma_d^2\|\mathbf{w}_d\|^2 + \sigma_d^2\|\mathbf{v}_d\|^2)^2 + 4(\overline{\delta x_d}^2 + 2\sigma_d^2\overline{\delta x_d}\langle\mathbf{w}_d, \mathbf{v}_d\rangle)} \tag{113}$$

As a consequence of the above theorem, we see signs of a richer phenomenology arising in the spectrum. In particular, we observe that the input coordinate $d$ which has the highest variance will, in general, lead to eigenvalues with the largest magnitudes. However, this is mediated by the residue of target $y$ left after being explained by the $d$-th coordinate feature $\langle \mathbf{w}_d, \mathbf{v}_d \rangle x_d$, but weighted by the input $x_d$ — in other words $\overline{\delta x_d}$. Hence, we can expect to witness a tension between these two aspects, and especially, across the different coordinates, based on whether a particular coordinate feature gets picked up early in learning, or which coordinate races ahead of other coordinates in explaining the target. We will also see a similar behaviour in the change of signs in the second outlying eigenvalue based on the progress of the $d$-th coordinate, i.e., how $\langle \mathbf{w}_d, \mathbf{v}_d \rangle$ compares to $\beta_d^* = \overline{yx_d}/\sigma_d^2$.

**Diagonal Linear Networks and their Mixtures.** Note that the oft-employed setting of diagonal linear networks is actually a special case of the above setup, where $\beta_d = w_d\, v_d$, i.e., with $m = 1$. In which case the network function reduces to: $f(\mathbf{x}) = \boldsymbol{\beta}^\top \mathbf{x}$, with $\boldsymbol{\beta} = \begin{pmatrix} w_1\, v_1 & \cdots & w_D\, v_D \end{pmatrix}^\top$.

However, in the general case where $m > 1$, we can equivalently write the above network in terms of a mixture of $m$ diagonal linear networks. Define $\boldsymbol{\beta}^{(i)} = \begin{pmatrix} \mathbf{w}_{1,i}\, \mathbf{v}_{1,i} & \cdots & \mathbf{w}_{D,i}\, \mathbf{v}_{D,i} \end{pmatrix}^\top$. Then we have that:

$$f(\mathbf{x}) = \sum_{d=1}^{D} \langle \mathbf{w}_d, \mathbf{v}_d \rangle x_d = \sum_{d=1}^{D} \sum_{i=1}^{m} \mathbf{w}_{d,i} \mathbf{v}_{d,i}\, x_d = \sum_{i=1}^{m} \langle \boldsymbol{\beta}^{(i)}, \mathbf{x} \rangle. \tag{114}$$

In the case, above we consider all the experts of equal importance, and since we only have a scalar output we omit the usual softmax operation.

### B.4.2 EXTENSION TO MULTI-DIMENSIONAL OUTPUTS

Now, let us try to analyze how the results would generalize in the setting of $K > 1$ outputs. The simplest possible extension is to have a network with separate heads for each of the output dimensions, i.e., $f^k(x) = \langle \mathbf{w}_k, \mathbf{v}_k \rangle x$ for all $k \in [K]$, but which are related to each through the loss. If we consider mean-squared loss, we have:

$$\ell(\boldsymbol{\theta}) = \frac{1}{2n} \sum_{i=1}^{n} \sum_{k=1}^{K} \left( f^k(x_i) - y_i^k \right)^2 \quad \text{where,} \quad \boldsymbol{\theta} = \{\mathbf{w}_1, \mathbf{v}_1, \cdots, \mathbf{w}_K, \mathbf{v}_K\} \tag{115}$$

Alternatively, we can rewrite the above network as a special case of a one-hidden layer linear network,

$$f(\mathbf{x}) = \mathbf{W}\mathbf{v}x\,, \text{ where } \mathbf{v} = \begin{pmatrix} \mathbf{v}_1^\top & \cdots & \mathbf{v}_K^\top \end{pmatrix}^\top \in \mathbb{R}^{mK}\,, \text{ and } \mathbf{W} = \begin{pmatrix} \mathbf{w}_1^\top & \cdots & \mathbf{0} \\ \vdots & \ddots & \vdots \\ \mathbf{0} & \cdots & \mathbf{w}_K^\top \end{pmatrix} \in \mathbb{R}^{K \times mK}\,.$$

Moving on, the gradient comes out to be,

$$\frac{\partial \ell(\boldsymbol{\theta})}{\partial \mathbf{w}_k} = \frac{1}{n} \sum_{i=1}^{n} (\langle \mathbf{w}_k, \mathbf{v}_k \rangle x_i - y_i^k)\, x_i\, \mathbf{v}_k\,, \quad \frac{\partial \ell(\boldsymbol{\theta})}{\partial \mathbf{v}_k} = \frac{1}{n} \sum_{i=1}^{n} (\langle \mathbf{w}_k, \mathbf{v}_k \rangle x_i - y_i^k)\, x_i\, \mathbf{w}_k\,,$$

which can be equivalently written down as, $\nabla_{\mathbf{w}_k} \ell(\boldsymbol{\theta}) = \overline{\delta_k x}\, \mathbf{v}_k$ and $\nabla_{\mathbf{v}_k} \ell(\boldsymbol{\theta}) = \overline{\delta_k x}\, \mathbf{w}_k$, where, $\overline{\delta_k x} := \frac{1}{n} \sum_{i=1}^{n} (\langle \mathbf{w}_k, \mathbf{v}_k \rangle x_i - y_i^k)\, x_i = \langle \mathbf{w}_k, \mathbf{v}_k \rangle \sigma^2 - \overline{y_k x}$ denotes the residual for the $k$-th output dimension weighted by the input.

Consequently, the Hessian is zero on the off-diagonal blocks that comprise of parameters pertaining to different output dimensions. On the other hand, along the diagonal block for the $k$-th output dimension, the Hessian can be written down as:

$$\mathbf{H}_{\mathrm{L}}^k = \begin{array}{c} \frac{\partial}{\partial \mathbf{v}_k} \\ \frac{\partial}{\partial \mathbf{w}_k} \end{array} \overset{\begin{array}{cc} \frac{\partial}{\partial \mathbf{v}_k^\top} & \frac{\partial}{\partial \mathbf{w}_k^\top} \end{array}}{\begin{pmatrix} \sigma_x^2\, \mathbf{w}_k \mathbf{w}_k^\top & \sigma_x^2\, \mathbf{w}_k \mathbf{v}_k^\top + \overline{\delta_k x} \mathbf{I}_m \\ \sigma_x^2\, \mathbf{v}_k \mathbf{w}_k^\top + \overline{\delta_k x} \mathbf{I}_m & \sigma_x^2\, \mathbf{v}_k \mathbf{v}_k^\top \end{pmatrix}} \tag{116}$$

Therefore the overall spectrum is just the direct sum of the spectrum of the diagonal blocks, for which we can fall back to the case of our results from the univariate case.

**Corollary B.2.** For the above setting of one-hidden layer linear network $f(\mathbf{x}) = \mathbf{W}\mathbf{v}x$ with $2mK$ parameters, the Hessian $\mathbf{H}_{\mathrm{L}}$ spectrum decouples across the parameter groups for each output dimension $\{\mathbf{w}_k, \mathbf{v}_k\}$, thereby resulting in a spectrum which has a set of $2(m-1)K$ bulk eigenvalues $\lambda_{\mathrm{bulk}}^k = \pm \overline{\delta_k x} = \pm(\langle \mathbf{w}_k, \mathbf{v}_k\rangle \sigma^2 - \overline{y_k x})$, where each is repeated $m-1$ times with both positive and negative sign, and a set of $2K$ paired outlying eigenvalues defined by the expression:

$$\lambda_{\mathrm{outlier}_{1,2}}^k = \frac{1}{2}(\sigma^2\|\mathbf{w}_k\|^2 + \sigma^2\|\mathbf{v}_k\|^2) \tag{117}$$

$$\pm \frac{1}{2}\sqrt{(\sigma^2\|\mathbf{w}_k\|^2 + \sigma^2\|\mathbf{v}_k\|^2)^2 + 4(\overline{\delta_k x}^2 + 2\sigma^2\,\overline{\delta_k x}\langle \mathbf{w}_k, \mathbf{v}_k\rangle)} \tag{118}$$

Based on the above form of the eigenvalues, we can see the second outlier will converge to zero in the order of the speed of learning across the various output dimensions. The output dimension where learning is retarded will still contribute towards the paired outlier with the negative sign, and thus to negative eigenvalues. As another corollary, we have that the eigenvector structure in this case is given by:

**Corollary B.3.** For the above setting, the Hessian eigenvectors corresponding to the outlying eigenvalues, determined up to scaling and sign, take the form, for $i \in \{1, 2\}$, given below:

$$\mathbf{z}_{\mathrm{outlier}_i}^k = \lambda_{\mathrm{outlier}_i}^k \begin{pmatrix} \mathbf{0} \\ \mathbf{w}_k \\ \mathbf{v}_k \\ \mathbf{0} \end{pmatrix} + \overline{\delta_k x} \begin{pmatrix} \mathbf{0} \\ \mathbf{v}_k \\ \mathbf{w}_k \\ \mathbf{0} \end{pmatrix} \tag{119}$$

## B.5 SELF-ATTENTION

**Definition B.1.** Let $\mathbf{A} \in \mathbb{R}^{L \times L}$ be an attention matrix and $\mathbf{x} \in \mathbb{R}^L$ the input sequence. Following Ormaniec et al. (2024), let us define the first attention vector as

$$\mathbf{m}_1 := \mathbf{A}\mathbf{x} \in \mathbb{R}^L.$$

Similarly, the second and third central moment matrices are

$$\mathbf{m}_2 := \left[\sum_{j=1}^L \mathbf{A}_{i,j}\left(\mathbf{x}_j - [\mathbf{m}_1]_i\right)^2\right]_{1 \le i \le L} \in \mathbb{R}^L$$

$$\mathbf{m}_3 := \left[\sum_{j=1}^L \mathbf{A}_{i,j}\left(\mathbf{x}_j - [\mathbf{m}_1]_i\right)^3\right]_{1 \le i \le L} \in \mathbb{R}^L.$$

**Theorem 3.5.** For the above setting, the Hessian has $d_K - 1$ pairs of bulk eigenvalues $\pm\gamma$. Moreover, there are three outlier eigenvalues $\lambda_{\mathrm{outlier}_{1,2,3}}$, that satisfy

$$\frac{1}{3}(\alpha + \zeta\|\mathbf{w}_{\mathrm{K}}\|^2 + \zeta\|\mathbf{w}_{\mathrm{Q}}\|^2) - \xi \le \lambda_{\mathrm{outlier}_{1,2,3}} \le \frac{1}{3}(\alpha + \zeta\|\mathbf{w}_{\mathrm{K}}\|^2 + \zeta\|\mathbf{w}_{\mathrm{Q}}\|^2) + \xi, \text{ where}$$

$$\xi = \frac{2}{3}\sqrt{3\gamma^2 + 6\zeta\gamma\langle \mathbf{w}_{\mathrm{K}}, \mathbf{w}_{\mathrm{Q}}\rangle + 3(\beta^2 - \alpha\zeta)\left(\|\mathbf{w}_{\mathrm{K}}\|^2 + \|\mathbf{w}_{\mathrm{Q}}\|^2\right) + (\zeta\|\mathbf{w}_{\mathrm{K}}\| + \zeta\|\mathbf{w}_{\mathrm{Q}}\| + \alpha)^2}.$$

*Proof.*
**Stating the Hessian.** Let us start by noting that the setting we consider here can be obtained from the one studied by (Ormaniec et al., 2024) by setting their embedding dimension to $d_{\mathrm{V}} = 1$ and multiplying their loss by a factor of $\frac{1}{2}$. Hence, we recall the Hessian expression for the considered model and loss (Theorem 1 and Theorem 2)

$$\mathbf{H}_{\mathrm{L}} =$$

$$
\begin{array}{c}
\quad\quad \frac{\partial}{\partial w_{\mathrm{V}}} \quad\quad\quad \frac{\partial}{\partial \mathbf{w}_{\mathrm{Q}}^{\top}} \quad\quad\quad\quad\quad\quad \frac{\partial}{\partial \mathbf{w}_{\mathrm{K}}^{\top}} \\
\begin{array}{c} \frac{\partial}{\partial w_{\mathrm{V}}} \\ \frac{\partial}{\partial \mathbf{w}_{\mathrm{Q}}} \\ \frac{\partial}{\partial \mathbf{w}_{\mathrm{K}}} \end{array}
\left(
\begin{array}{ccc}
\frac{\mathbf{m}_1^{\top}\mathbf{m}_1}{L} & \frac{(w_{\mathrm{V}}\mathbf{m}_1+\delta_{\mathbf{x},\mathbf{y}})^{\top}\mathbf{z}_1}{L\sqrt{d_K}}\mathbf{w}_{\mathrm{K}}^{\top} & \frac{(w_{\mathrm{V}}\mathbf{m}_1+\delta_{\mathbf{x},\mathbf{y}})^{\top}\mathbf{z}_1}{L\sqrt{d_K}}\mathbf{w}_{\mathrm{Q}}^{\top} \\
\cdot & \frac{w_{\mathrm{V}}{}^2\mathbf{z}_1{}^{\top}\mathbf{z}_1+w_{\mathrm{V}}\delta_{\mathbf{x},\mathbf{y}}^{\top}\mathbf{z}_2}{Ld_K}\mathbf{w}_{\mathrm{K}}\mathbf{w}_{\mathrm{K}}^{\top} & \frac{w_{\mathrm{V}}{}^2\mathbf{z}_1{}^{\top}\mathbf{z}_1+w_{\mathrm{V}}\delta_{\mathbf{x},\mathbf{y}}^{\top}\mathbf{z}_2}{Ld_K}\mathbf{w}_{\mathrm{K}}\mathbf{w}_{\mathrm{Q}}^{\top} + \frac{w_{\mathrm{V}}\delta_{\mathbf{x},\mathbf{y}}^{\top}\mathbf{z}_1}{L\sqrt{d_K}}\mathbf{I}_{d_K} \\
\cdot & \cdot & \frac{w_{\mathrm{V}}{}^2\mathbf{z}_1{}^{\top}\mathbf{z}_1+w_{\mathrm{V}}\delta_{\mathbf{x},\mathbf{y}}^{\top}\mathbf{z}_2}{Ld_K}\mathbf{w}_{\mathrm{Q}}\mathbf{w}_{\mathrm{Q}}^{\top}
\end{array}
\right),
\end{array}
$$

where we replaced the symmetric terms with $\cdot$. We note that there are some repeating scalars in the Hessian, so we define:

$$\alpha := \frac{\|\mathbf{m}_1\|^2}{L}$$

$$\beta := \frac{(w_{\mathrm{V}}\mathbf{m}_1 + \delta_{\mathbf{x},\mathbf{y}})^{\top}\mathbf{z}_1}{L\sqrt{d_K}} = \frac{(w_{\mathrm{V}}\mathbf{m}_1 + \delta_{\mathbf{x},\mathbf{y}})^{\top}(\mathbf{x}\odot\mathbf{m}_2)}{L\sqrt{d_K}}$$

$$\gamma := \frac{w_{\mathrm{V}}\delta_{\mathbf{x},\mathbf{y}}^{\top}\mathbf{z}_1}{L\sqrt{d_K}} = \frac{w_{\mathrm{V}}\delta_{\mathbf{x},\mathbf{y}}^{\top}(\mathbf{x}\odot\mathbf{m}_2)}{L\sqrt{d_K}}$$

$$\zeta := \frac{w_{\mathrm{V}}{}^2\mathbf{z}_1{}^{\top}\mathbf{z}_1 + w_{\mathrm{V}}\delta_{\mathbf{x},\mathbf{y}}^{\top}\mathbf{z}_2}{Ld_K} = \frac{w_{\mathrm{V}}{}^2\|\mathbf{x}\odot\mathbf{m}_2\|^2 + w_{\mathrm{V}}\delta_{\mathbf{x},\mathbf{y}}^{\top}(\mathbf{x}\odot\mathbf{x}\odot\mathbf{m}_3)}{Ld_K},$$

where we express the vectors representing the first and the second derivative of the softmax $\mathbf{z}_1$ and $\mathbf{z}_2$ in terms of the Hadamard products ($\odot$) between the input sequence and the attention moment matrices (definition B.1) according to Remark 3.1 from Ormaniec et al. (2024).

After substituting the expressions for $\alpha, \beta, \gamma, s$ the Hessian simplifies to

$$\mathbf{H}_{\mathrm{L}} = \begin{pmatrix} \alpha & \beta\mathbf{w}_{\mathrm{K}}^{\top} & \beta\mathbf{w}_{\mathrm{Q}}^{\top} \\ \beta\mathbf{w}_{\mathrm{K}} & \zeta\mathbf{w}_{\mathrm{K}}\mathbf{w}_{\mathrm{K}}^{\top} & \zeta\mathbf{w}_{\mathrm{K}}\mathbf{w}_{\mathrm{Q}}^{\top} + \gamma\mathbf{I}_{d_K} \\ \beta\mathbf{w}_{\mathrm{Q}} & \zeta\mathbf{w}_{\mathrm{Q}}\mathbf{w}_{\mathrm{K}}^{\top} + \gamma\mathbf{I}_{d_K} & \zeta\mathbf{w}_{\mathrm{Q}}\mathbf{w}_{\mathrm{Q}}^{\top} \end{pmatrix}.$$

**Solving the eigenvalues.** The next step is to solve the characteristic equation, namely $|\mathbf{H}_{\mathrm{L}} - \lambda\mathbf{I}_{2d_K+1}| = 0$ or equivalently

$$\left| \begin{pmatrix} \alpha - \lambda & \beta\mathbf{w}_{\mathrm{K}}^{\top} & \beta\mathbf{w}_{\mathrm{Q}}^{\top} \\ \beta\mathbf{w}_{\mathrm{K}} & s\mathbf{w}_{\mathrm{K}}\mathbf{w}_{\mathrm{K}}^{\top} - \lambda\mathbf{I}_{d_K} & \zeta\mathbf{w}_{\mathrm{K}}\mathbf{w}_{\mathrm{Q}}^{\top} + \gamma\mathbf{I}_{d_K} \\ \beta\mathbf{w}_{\mathrm{Q}} & \zeta\mathbf{w}_{\mathrm{Q}}\mathbf{w}_{\mathrm{K}}^{\top} + \gamma\mathbf{I}_{d_K} & \zeta\mathbf{w}_{\mathrm{Q}}\mathbf{w}_{\mathrm{Q}}^{\top} - \lambda\mathbf{I}_{d_K} \end{pmatrix} \right| = 0.$$

To simplify the derivations, let's define some helper terms

$$\mathbf{v} := \begin{pmatrix} \frac{\beta}{\zeta} \\ \mathbf{w}_{\mathrm{K}} \end{pmatrix},$$

$$\mathbf{E} := \begin{pmatrix} \lambda + \frac{\beta^2}{\zeta} - \alpha & \mathbf{0}^{\top} \\ \mathbf{0} & \lambda\mathbf{I}_{d_K} \end{pmatrix},$$

$$\mathbf{G} := \begin{pmatrix} \mathbf{0} & \gamma\mathbf{I}_{d_K} \end{pmatrix}.$$

Using the above definitions, we can equivalently represent our problem as

$$\left| \begin{pmatrix} \zeta\mathbf{v}\mathbf{v}^{\top} - \mathbf{E} & \zeta\mathbf{v}\mathbf{w}_{\mathrm{Q}}^{\top} + \mathbf{G}^{\top} \\ \zeta\mathbf{w}_{\mathrm{Q}}\mathbf{v}^{\top} + \mathbf{G} & \zeta\mathbf{w}_{\mathrm{Q}}\mathbf{w}_{\mathrm{Q}}^{\top} - \lambda\mathbf{I}_{d_K} \end{pmatrix} \right| = 0.$$

Via the Schur complement we have $\left| \begin{pmatrix} \mathbf{A} & \mathbf{B} \\ \mathbf{C} & \mathbf{D} \end{pmatrix} \right| = |\mathbf{A}| |\mathbf{D} - \mathbf{C}\mathbf{A}^{-1}\mathbf{B}|$. We can apply this formula to the above equation, where $\mathbf{A} = \zeta\mathbf{v}\mathbf{v}^{\top} - \mathbf{E}$, which is invertible as long as $\lambda \neq 0$ and $\lambda \neq \alpha - \frac{\beta^2}{\zeta}$.

Hence, the determinant of $\mathbf{D}$ is not zero, and the roots of the equation above (which yields the eigenvalues) will come from the other term, $\left|\mathbf{D} - \mathbf{C}\mathbf{A}^{-1}\mathbf{B}\right| = 0$. Let us then calculate it:

$$\left|\zeta\mathbf{w}_{\mathrm{Q}}\mathbf{w}_{\mathrm{Q}}^{\top} - \lambda\mathbf{I}_{d_K} + \left(\zeta\mathbf{v}\mathbf{w}_{\mathrm{Q}}^{\top} + \mathbf{G}^{\top}\lambda\mathbf{I}_{d_K}\right)\left(\mathbf{E} - \zeta\mathbf{v}\mathbf{v}^{\top}\right)^{-1}\left(\zeta\mathbf{w}_{\mathrm{Q}}\mathbf{v}^{\top} + \mathbf{G}\lambda\mathbf{I}_{d_K}\right)\right| = 0.$$

Using the Woodbury matrix identity, we invert the matrix $\mathbf{E} - \zeta\mathbf{v}\mathbf{v}^{\top}$.

$$\left(\mathbf{E} - \zeta\mathbf{v}\mathbf{v}^{\top}\right)^{-1} = \mathbf{E}^{-1} + \mathbf{E}^{-1}\mathbf{v}\left(\frac{1}{\zeta} - \mathbf{v}^{\top}\mathbf{E}^{-1}\mathbf{v}\right)^{-1}\mathbf{v}^{\top}\mathbf{E}^{-1}$$

$$= \mathbf{E}^{-1} + \mathbf{E}^{-1}\mathbf{v}\left(\frac{1}{\zeta} - \|\mathbf{e}\|^2\right)^{-1}\mathbf{v}^{\top}\mathbf{E}^{-1}$$

$$= \mathbf{E}^{-1} + \frac{\zeta}{1 - \zeta\|\mathbf{e}\|^2}\mathbf{E}^{-1}\mathbf{v}\mathbf{v}^{\top}\mathbf{E}^{-1},$$

where we substituted $\mathbf{e} = \mathbf{E}^{-\frac{1}{2}}\mathbf{v}$.

Additionally, after defining $\mathbf{K} = \mathbf{G}\mathbf{E}^{-\frac{1}{2}}$, we simplify appendix B.5 to

$$\left|\zeta\mathbf{w}_{\mathrm{Q}}\mathbf{w}_{\mathrm{Q}}^{\top} - \lambda\mathbf{I}_{d_K} + \zeta^2\|\mathbf{e}\|^2\mathbf{w}_{\mathrm{Q}}\mathbf{w}_{\mathrm{Q}}^{\top} + \frac{\zeta^3\|\mathbf{e}\|^4}{1 - \zeta\|\mathbf{e}\|^2}\mathbf{w}_{\mathrm{Q}}\mathbf{w}_{\mathrm{Q}}^{\top}\right.$$

$$+\zeta\mathbf{w}_{\mathrm{Q}}\mathbf{e}^{\top}\mathbf{K}^{\top} + \frac{\zeta^2\|\mathbf{e}\|^2}{1 - \zeta\|\mathbf{e}\|^2}\mathbf{w}_{\mathrm{Q}}\mathbf{e}^{\top}\mathbf{K}^{\top} + \zeta\mathbf{K}\mathbf{e}\mathbf{w}_{\mathrm{Q}}^{\top} + \frac{\zeta^2\|\mathbf{e}\|^2}{1 - \zeta\|\mathbf{e}\|^2}\mathbf{K}\mathbf{e}\mathbf{w}_{\mathrm{Q}}^{\top} + \qquad (120)$$

$$\left.\mathbf{K}\mathbf{K}^{\top} + \frac{\zeta}{1 - \zeta\|\mathbf{e}\|^2}\mathbf{K}\mathbf{e}\mathbf{e}^{\top}\mathbf{K}^{\top}\right| = 0.$$

Let's now note some simplifications that can be made regarding $\mathbf{K}$ and $\mathbf{e}$. First, their product turns out to be just rescaled $\mathbf{w}_{\mathrm{K}}$

$$\mathbf{K}\mathbf{e} = \mathbf{G}\mathbf{E}^{-1}\mathbf{v} = \begin{pmatrix} \mathbf{0} & \gamma\mathbf{I}_{d_K} \end{pmatrix}\begin{pmatrix} (\lambda + \frac{\beta^2}{\zeta} - \alpha)^{-1} & \mathbf{0}^{\top} \\ \mathbf{0} & \lambda^{-1}\mathbf{I}_{d_K} \end{pmatrix}\begin{pmatrix} \frac{\beta}{\zeta} \\ \mathbf{w}_{\mathrm{K}} \end{pmatrix} = \gamma\lambda^{-1}\mathbf{w}_{\mathrm{K}}.$$

Second,

$$\mathbf{K}\mathbf{K}^{\top} = \mathbf{G}\mathbf{E}^{-1}\mathbf{G}^{\top} = \begin{pmatrix} \mathbf{0} & \gamma\mathbf{I}_{d_K} \end{pmatrix}\begin{pmatrix} (\lambda + \frac{\beta^2}{\zeta} - \alpha)^{-1} & \mathbf{0}^{\top} \\ \mathbf{0} & \lambda^{-1}\mathbf{I}_{d_K} \end{pmatrix}\begin{pmatrix} \mathbf{0} \\ \gamma\mathbf{I}_{d_K} \end{pmatrix} = \gamma^2\lambda^{-1}\mathbf{I}_{d_K}.$$

This allows us to further simplify eq. (120) into

$$\left|\zeta\mathbf{w}_{\mathrm{Q}}\mathbf{w}_{\mathrm{Q}}^{\top} - \lambda\mathbf{I}_{d_K} + \zeta^2\|\mathbf{e}\|^2\mathbf{w}_{\mathrm{Q}}\mathbf{w}_{\mathrm{Q}}^{\top} + \frac{\zeta^3\|\mathbf{e}\|^4}{1 - \zeta\|\mathbf{e}\|^2}\mathbf{w}_{\mathrm{Q}}\mathbf{w}_{\mathrm{Q}}^{\top} + \zeta\gamma\lambda^{-1}\mathbf{w}_{\mathrm{Q}}\mathbf{w}_{\mathrm{K}}^{\top}\right.$$

$$+\frac{\zeta^2\gamma\lambda^{-1}\|\mathbf{e}\|^2}{1 - \zeta\|\mathbf{e}\|^2}\mathbf{w}_{\mathrm{Q}}\mathbf{w}_{\mathrm{K}}^{\top} + \zeta\gamma\lambda^{-1}\mathbf{w}_{\mathrm{K}}\mathbf{w}_{\mathrm{Q}}^{\top} + \frac{\zeta^2\gamma\lambda^{-1}\|\mathbf{e}\|^2}{1 - \zeta\|\mathbf{e}\|^2}\mathbf{w}_{\mathrm{K}}\mathbf{w}_{\mathrm{Q}}^{\top} + \gamma^2\lambda^{-1}\mathbf{I}_{d_K}$$

$$\left.+\frac{\zeta\gamma^2\lambda^{-2}}{1 - \zeta\|\mathbf{e}\|^2}\mathbf{w}_{\mathrm{K}}\mathbf{w}_{\mathrm{K}}^{\top}\right| = 0.$$

Similarly, as in the case of the two-layer MLP, we can group the scalars in front of $\mathbf{w}_{\mathrm{Q}}\mathbf{w}_{\mathrm{Q}}^{\top}$, $\mathbf{w}_{\mathrm{K}}\mathbf{w}_{\mathrm{Q}}^{\top}$, $\mathbf{w}_{\mathrm{Q}}\mathbf{w}_{\mathrm{K}}^{\top}$, and $\mathbf{I}_{d_K}$. This further reduces the equation to

$$\left|\frac{\zeta}{1 - \zeta\|\mathbf{e}\|^2}\mathbf{w}_{\mathrm{Q}}\mathbf{w}_{\mathrm{Q}}^{\top} + \frac{\zeta\gamma\lambda^{-1}}{1 - \zeta\|\mathbf{e}\|^2}\left(\mathbf{w}_{\mathrm{Q}}\mathbf{w}_{\mathrm{K}}^{\top} + \mathbf{w}_{\mathrm{K}}\mathbf{w}_{\mathrm{Q}}^{\top}\right)\right.$$

$$\left.+\frac{\zeta\gamma^2\lambda^{-2}}{1 - \zeta\|\mathbf{e}\|^2}\mathbf{w}_{\mathrm{K}}\mathbf{w}_{\mathrm{K}}^{\top} - \left(\lambda - \gamma^2\lambda^{-1}\right)\mathbf{I}_{d_K}\right| = 0.$$

Further, multiplying both sides by $\lambda^2\left(1 - \zeta\|\mathbf{e}\|^2\right) \neq 0$, the equation becomes

$$\left|\zeta\lambda^{-2}\mathbf{w}_{\mathrm{Q}}\mathbf{w}_{\mathrm{Q}}^{\top} + \zeta\gamma\lambda\left(\mathbf{w}_{\mathrm{Q}}\mathbf{w}_{\mathrm{K}}^{\top} + \mathbf{w}_{\mathrm{K}}\mathbf{w}_{\mathrm{Q}}^{\top}\right) + \zeta\gamma^2\mathbf{w}_{\mathrm{K}}\mathbf{w}_{\mathrm{K}}^{\top} - \lambda\left(1 - \zeta\|e\|^2\right)\left(\lambda^2 - \gamma^2\right)\mathbf{I}_{d_K}\right| = 0.$$

Defining, $\mathbf{w} := \lambda \mathbf{w}_Q + t \mathbf{w}_K$ and $\nu := \lambda \left(1 - \zeta \|e\|^2\right) \left(\lambda^2 - \gamma^2\right)$, we can view the equation as

$$\left| \zeta \mathbf{w} \mathbf{w}^\top - \nu \mathbf{I}_{d_K} \right| = 0.$$

This is a determinant of a matrix of the same form as in the two-layer linear MLP case, so again we know that

$$\left| \zeta \mathbf{w} \mathbf{w}^\top - \nu \mathbf{I}_{d_K} \right| = \nu^{d_K - 1} \left( \zeta \|\mathbf{w}\|^2 - \nu \right) = 0.$$

Hence, we get $d_K - 1$ repeat roots of $\nu = 0$

$$\lambda = \pm \gamma.$$

> **Remark B.2.** Note that if we consider only the block of the Hessian corresponding to the query
> and key weight matrices, the Hessian has the same form as that of a two-layer linear MLP, just
> with different scalars and vectors. Hence, we know that the bulk eigenvalues of the query-key
> Hessian block are also given by $\lambda = \pm \gamma$.

The remaining three eigenvalues correspond to the solutions of the following equation in $\lambda$

$$\zeta \|\mathbf{w}\|^2 - \nu = \zeta \lambda^2 \|\mathbf{w}_Q\|^2 + 2 \zeta \gamma \lambda \langle \mathbf{w}_Q, \mathbf{w}_K \rangle + \zeta \gamma^2 \|\mathbf{w}_K\|^2 - \lambda \left(1 - \zeta \|e\|^2\right) \left(\lambda^2 - \gamma^2\right) = 0.$$

To solve it, we need first to disentangle $\|\mathbf{e}\|^2$

$$\|\mathbf{e}\|^2 = \mathbf{v}^\top \mathbf{E}^{-1} \mathbf{v} = \begin{pmatrix} \frac{\beta}{\zeta} & \mathbf{w}_K^\top \end{pmatrix} \left( \begin{matrix} \left( \lambda + \frac{\beta^2}{\zeta} - \alpha \right)^{-1} & \mathbf{0}^\top \\ \mathbf{0} & \lambda^{-1} \mathbf{I}_{d_K} \end{matrix} \right) \begin{pmatrix} \frac{\beta}{\zeta} \\ \mathbf{w}_K \end{pmatrix}$$

$$= \frac{\beta^2}{\zeta^2} \left( \lambda + \frac{\beta^2}{\zeta} - \alpha \right)^{-1} + \lambda^{-1} \|\mathbf{w}_K\|^2$$

$$= \frac{\beta^2}{\zeta \left( \zeta \lambda + \beta^2 - \alpha \zeta \right)} + \lambda^{-1} \|\mathbf{w}_K\|^2.$$

This lets us simplify the last term in our equation of interest

$$\lambda \left(1 - \zeta \|\mathbf{e}\|^2\right) \left(\lambda^2 - \gamma^2\right) = \left( \lambda - \frac{\beta^2 \lambda}{\zeta \lambda + \beta^2 - \alpha \zeta} - \zeta \|\mathbf{w}_K\|^2 \right) \left(\lambda^2 - \gamma^2\right)$$

$$= \lambda^3 - \lambda \gamma^2 + \frac{\beta^2 \gamma^2 \lambda - \beta^2 \lambda^3}{\zeta \lambda + \beta^2 - \alpha \zeta} - \zeta \lambda^2 \|\mathbf{w}_K\|^2 + \zeta \gamma^2 \|\mathbf{w}_K\|^2$$

$$\zeta \lambda^2 \|\mathbf{w}_Q\|^2 + 2 \zeta \gamma \lambda \langle \mathbf{w}_Q, \mathbf{w}_K \rangle + \cancel{\zeta \gamma^2 \|\mathbf{w}_K\|^2} - \lambda^3 + \lambda \gamma^2$$

$$- \frac{\beta^2 \gamma^2 \lambda - \beta^2 \lambda^3}{\zeta \lambda + \beta^2 - \alpha \zeta} + \zeta \lambda^2 \|\mathbf{w}_K\|^2 - \cancel{\zeta \gamma^2 \|\mathbf{w}_K\|^2} = 0.$$

Dividing both sides by $\lambda$, we get the equation

$$\zeta \lambda \|\mathbf{w}_Q\|^2 + 2 \gamma \zeta \langle \mathbf{w}_Q, \mathbf{w}_K \rangle - \lambda^2 + \gamma^2 - \frac{\beta^2 \gamma^2 - \beta^2 \lambda^2}{\zeta \lambda + \beta^2 - \alpha \zeta} + \zeta \lambda \|\mathbf{w}_K\|^2 = 0.$$

Finally, multiplying by $\zeta \lambda + \beta^2 - \alpha \zeta$ and dividing by $\zeta$ we arrive at the third-order polynomial

$$-\lambda^3 + \left(\alpha + \zeta(\|\mathbf{w}_K\|^2 + \|\mathbf{w}_Q\|^2)\right)\lambda^2 + \left((\beta^2 - \alpha \zeta)(\|\mathbf{w}_K\|^2 + \|\mathbf{w}_Q\|^2)\right)$$

$$+ 2\gamma \zeta \langle \mathbf{w}_K, \mathbf{w}_Q \rangle + \gamma^2)\lambda + \left(2\gamma(\beta^2 - \alpha \zeta)\langle \mathbf{w}_K, \mathbf{w}_Q \rangle - \alpha \gamma^2\right) = 0.$$

> **Remark B.3.** When $\gamma = 0$, the third-order polynomial can be simplified to a quadratic equation
> with solutions given by
>
> $$\lambda_{1,2} =$$
> $$\frac{1}{2}(\alpha + \zeta \|\mathbf{w}_K\|^2 + \zeta \|\mathbf{w}_Q\|^2) \pm \frac{1}{2}\sqrt{(\alpha + \zeta \|\mathbf{w}_K\|^2 + \zeta \|\mathbf{w}_Q\|^2)^2 + 4(\beta^2 - \alpha \zeta)(\|\mathbf{w}_K\|^2 + \|\mathbf{w}_Q\|^2)}.$$

We can put the cubic into its depressed form .i.e., $\tilde{\lambda}^3 + p\tilde{\lambda} + q = 0$, by making the substitution

$$\lambda = \tilde{\lambda} + \frac{\alpha + \zeta(\|\mathbf{w}_K\|^2 + \|\mathbf{w}_Q\|^2)}{3},$$

with $p$ and $q$ given by

$$p = -\frac{1}{3}\left(3\gamma^2 + 6\gamma\zeta\langle\mathbf{w}_K, \mathbf{w}_Q\rangle + 3(\beta^2 - \alpha\zeta)\left(\|\mathbf{w}_K\|^2 + \|\mathbf{w}_Q\|^2\right) + \left(\zeta\|\mathbf{w}_K\|^2 + \zeta\|\mathbf{w}_Q\|^2 + \alpha\right)^2\right),$$

$$\begin{aligned}
q = -\frac{1}{27}\big(&2(\alpha + \zeta(\|\mathbf{w}_K\|^2 + \|\mathbf{w}_Q\|^2))^3 \\
&+ 9(\alpha + \zeta(\|\mathbf{w}_K\|^2 + \|\mathbf{w}_Q\|^2))((\beta^2 - \alpha\zeta)(\|\mathbf{w}_K\|^2 + \|\mathbf{w}_Q\|^2) + 2\gamma\zeta\langle\mathbf{w}_K, \mathbf{w}_Q\rangle + \gamma^2) \\
&+ 27(2\gamma(\beta^2 - \alpha\zeta)\langle\mathbf{w}_K, \mathbf{w}_Q\rangle - \alpha\gamma^2)\big).
\end{aligned}$$

Using Viete's trigonometric expression of the roots in the three-real roots case, we have that

$$\tilde{\lambda}_k = 2\sqrt{-\frac{p}{3}}\cos\zeta\left(\frac{1}{3}\arccos\zeta\left(\frac{3q}{2p}\sqrt{-\frac{3}{p}}\right) - k\frac{2\pi}{3}\right),$$

for $k \in \{0, 1, 2\}$, which gives us closed-form expressions for the outlier eigenvalues.

Since $\cos\zeta \in [-1, 1]$, we can upper and lower bound the solutions to the cubic, we are looking for, as

$$\frac{\alpha + \zeta(\|\mathbf{w}_K\|^2 + \|\mathbf{w}_Q\|^2)}{3} - \xi \leq \lambda_{0,1,2} \leq \frac{\alpha + \zeta(\|\mathbf{w}_K\|^2 + \|\mathbf{w}_Q\|^2)}{3} + \xi,$$

where

$$\begin{aligned}
\xi &= 2\sqrt{-\frac{p}{3}} \\
&= \frac{2}{3}\sqrt{3\gamma^2 + 6\gamma\zeta\langle\mathbf{w}_K, \mathbf{w}_Q\rangle + 3(\beta^2 - \alpha\zeta)\left(\|\mathbf{w}_K\|^2 + \|\mathbf{w}_Q\|^2\right) + (\zeta\|\mathbf{w}_K\| + \zeta\|\mathbf{w}_Q\| + \alpha)^2}.
\end{aligned}$$

Moreover, note that since for any $p, q$, $\frac{1}{3}\arccos\zeta\left(\frac{3q}{2p}\sqrt{-\frac{3}{p}}\right) \in [0, \frac{\pi}{3}]$, the possible values of the cosine for $k \in \{0, 1, 2\}$ are in range $[\frac{1}{2}, 1]$, $[-\frac{1}{2}, \frac{1}{2}]$, and $[-1, -\frac{1}{2}]$ respectively. This implies the lower and upper bounds on the largest Hessian eigenvalue stated in remark B.4.

**Remark B.4.** The largest Hessian eigenvalue can be bounded by

$$\underbrace{\frac{\alpha + \zeta(\|\mathbf{w}_K\|^2 + \|\mathbf{w}_Q\|^2)}{3}}_{\text{tr}(\mathbf{H}_L)/3} + \frac{1}{2}\xi \leq \lambda_{\max} \leq \underbrace{\frac{\alpha + \zeta(\|\mathbf{w}_K\|^2 + \|\mathbf{w}_Q\|^2)}{3}}_{\text{tr}(\mathbf{H}_L)/3} + \xi. \tag{121}$$

The lower bound from remark B.4 and the upper bound on the absolute value of the lowest magnitude eigenvalue through the positive bulk directly imply 3.1.

## C    Insights on Learning Phases

**Three Learning Phases.** By noting that the trace of the Hessian in the setting from theorem 3.1 is given by $\operatorname{tr}(\mathbf{H}_{\mathrm{L}}) = \sigma_{\mathbf{x}}^2\|\mathbf{w}\|^2 + \sigma_{\mathbf{x}}^2\|\mathbf{v}\|^2$, the expression for the outlying eigenvalues from eq. (1) can be reformulated as

$$\lambda_{\mathrm{outlier}_{1,2}} = \frac{1}{2}\operatorname{tr}(\mathbf{H}_{\mathrm{L}}) \pm \sqrt{\frac{1}{4}\operatorname{tr}^2(\mathbf{H}_{\mathrm{L}}) - \lambda_{\mathrm{outlier}_1}\lambda_{\mathrm{outlier}_2}}\,. \tag{122}$$

Further, the expression $-\lambda_{\mathrm{outlier}_1}\lambda_{\mathrm{outlier}_2} = (\overline{\delta x}^2 + 2\sigma_{\mathbf{x}}^2\overline{\delta x}\langle\mathbf{w},\mathbf{v}\rangle)$ present in eq. (122) can be rewritten as, $(\langle\mathbf{w},\mathbf{v}\rangle\sigma_{\mathbf{x}}^2 - \overline{yx})(3\langle\mathbf{w},\mathbf{v}\rangle\sigma_{\mathbf{x}}^2 - \overline{yx})$ which, as a quadratic function of $\langle\mathbf{w},\mathbf{v}\rangle$, is negative if $\langle\mathbf{w},\mathbf{v}\rangle < \overline{yx}/\sigma_{\mathbf{x}}^2$ and $\langle\mathbf{w},\mathbf{v}\rangle > \overline{yx}/3\sigma_{\mathbf{x}}^2$ and non-negative otherwise. Note, $\overline{yx}/\sigma_{\mathbf{x}}^2 = \mathbb{E}\left[yx\right]/\mathbb{E}\left[x^2\right] =: \theta^{\star}$ is precisely the closed-form solution for the scalar parameter $\theta$ in the linear regression $\mathbf{y} = \theta\,\mathbf{x}$. Effectively, the linear network under consideration is nothing but a parameterization of this scalar in the form of an inner-product between the layer weights $\mathbf{w}, \mathbf{v}$.

So, we categorize the phases of learning in three kinds:

- *early phase*: when the vectors at initialization are not correlated, and hence $\langle\mathbf{w},\mathbf{v}\rangle \leq \theta^{\star}/3$,

- *late phase*: when $\theta^{\star}/3 \leq \langle\mathbf{w},\mathbf{v}\rangle \leq \theta^{\star}$,

- *divergent phase*: when $\theta^{\star} \leq \langle\mathbf{w},\mathbf{v}\rangle$.

In early and divergent phases, $\lambda_1 \geq \operatorname{tr}(\mathbf{H}_{\mathrm{L}})$ and $\lambda_2 \leq 0$. In the late phase, we have that $\lambda_1 \leq \operatorname{tr}(\mathbf{H}_{\mathrm{L}})$ and $\lambda_2 \geq 0$. Upon reaching the solution, the Hessian is just rank one, with its spectrum given by $\lambda_1 = \operatorname{tr}(\mathbf{H}_{\mathrm{L}}) = \sigma_{\mathbf{x}}^2\|\mathbf{w}\|^2 + \sigma_{\mathbf{x}}^2\|\mathbf{v}\|^2$, and with the rest of the eigenvalues being 0.

# D  EXPERIMENTS

## D.1  EXPERIMENTAL SETUP

Unless mentioned otherwise, all eigenvalues and spectra were obtained using curvlinops library (Dangel et al., 2025).

### D.1.1  EXPERIMENTS ACCOMPANYING THEORETICAL RESULTS IN SECTION 3

**Figure 7** For varying $\sigma_{\mathbf{x}}^2 \in (0, 20)$, we draw a random sample $\mathbf{x}$ of size 100000 and random standard normal targets. We initialize the parameters of the unidimensional, two-layer, linear network from a zero-centered Gaussian with variance as in standard parametrization (see appendix D.1.3), and compute all MSE loss Hessian eigenvalues.

**Figures 2a to 2c** We consider a single uni-dimensional self-attention layer with $d_K = 1000$, batch size and sequence length equal to 10. For figs. 2a and 2b we set the parameter that is not changing to have norm 1. We find the bulk eigenvalues by looking for the ones that repeat multiple times in the spectrum.

**Figure 9** We consider a self-attention layer with embedding dimension $\mathbf{d} = 10$, $d_K = 100$, batch size and sequence length 10. We plot the repeating eigenvalues as the bulk and limit the outliers to the largest 5%, due to the large number of eigenvalues.

### D.1.2  PAIRED NATURE OF OUTLIER EIGENVALUES

**Model.** To obtain results from fig. 3, we consider a two-layer linear model with arbitrary input size $D$ and output size $K = 1$. We consider two model initialization strategies: orthonormal initialization and initialization according to standard parametrization (see appendix D.1.3 for details). We consider two network widths $m \in \{10, 150\}$ and multiple values for the input size $D \leq m$. For every point in fig. 3 we sample 100 models.

**Data.** We consider two types of datasets. The first one, which we call decorrelated, is inspired by the theoretical assumptions required to derive the spectrum of the linear network with bias. In this dataset we ensure that the empirical correlation between the features and between any feature and the target it equal to zero and that all the features and the target are normalized. The second dataset is simply normalized. We draw samples of size 102, and 152 for width 10 and 150 respectively.

**Finding Paired Outliers.** To find the paired outliers, we compute all Hessian eigenvalues, and we look for a pair of eigenvalues that sums up to a value closest to $\frac{1}{D} \operatorname{tr}(\mathbf{H})$. Specifically, we find the quantity $\min(|\lambda_1 + \lambda_2 - \frac{1}{D} \operatorname{tr}(\mathbf{H}_{\mathrm{L}})|)$ and the next smallest difference $\min_2(|\lambda_1 + \lambda_2 - \frac{1}{D} \operatorname{tr}(\mathbf{H}_{\mathrm{L}})|)$.

### D.1.3  CORRELATION BETWEEN LARGEST HESSIAN EIGENVALUE AND NORM/ALIGNMENT OF WEIGHTMATRICES

**Matrix Norms and Alignment.** In section 4.2, figs. 4 and 15, we study three matrix norms and three matrix alignment metrics that in the vector case should reduce to, respectively, Euclidean norm and inner product that we observe in the formulas for the largest Hessian eigenvalues. Specifically, for any weight matrix $\mathbf{W} \in \mathbb{R}^{n \times m}$, we consider the following:

$$\text{Frobenius norm}: \|\mathbf{W}\|_{\mathrm{F}} = \sqrt{\sum_{i=1}^{n} \sum_{j=1}^{m} \mathbf{W}_{n,m}^2} = \sqrt{\operatorname{tr}(\mathbf{W}^{\top} \mathbf{W})},$$

$$\text{Spectral norm}: \|\mathbf{W}\|_2 = \sigma_{\max}(\mathbf{W}),$$

$$\text{Nuclear norm}: \|\mathbf{W}\|_{\sigma} = \sum_{i=1}^{\min\{n,m\}} \sigma_i(\mathbf{W}).$$

We also consider an inner product between matrices. To be more precise, we define "inner-product-like" metrics corresponding to the matrix norms considered above. Specifically, for weight matrices $\mathbf{W} \in \mathbb{R}^{m,n}$ and $\mathbf{V} \in \mathbb{R}^{m,n}$, we define the following:

$$\text{Frobenius alignment}: \langle \mathbf{W}, \mathbf{V} \rangle_{\mathrm{F}} = \sum_{i=1}^{n} \sum_{j=1}^{m} \mathbf{W}_{n,m} \mathbf{V}_{n,m} = \mathrm{tr}\left(\mathbf{W}^{\top} \mathbf{V}\right),$$

$$\text{Spectral alignment}: \langle \mathbf{W}, \mathbf{V} \rangle_{2} = \sigma_{\max}(\mathbf{W}^{\top} \mathbf{V})$$

$$\text{Nuclear alignment}: \langle \mathbf{W}, \mathbf{V} \rangle_{\sigma} = \sum_{i=1}^{\min\{n,m\}} \sigma_i\left(\mathbf{W}^{\top} \mathbf{V}\right)$$

Note that, contrary to the Frobenius inner product, the spectral and nuclear products defined above are not real inner products, because they are not bilinear. In the vector case the all reduce to the standard vector inner product.

**Model.** We consider a two-layer linear network with arbitrary but equal input and output sizes $D = K$, $\boldsymbol{f}_{\mathbf{W}_2,\mathbf{W}_1}(\mathbf{x}) = \mathbf{W}_2 \mathbf{W}_1 \mathbf{x}$, where $\mathbf{W}_2 \in \mathbb{R}^{D \times m}$ and $\mathbf{W}_1 \in \mathbb{R}^{D \times m}$. We set the width of the model $m = 100$, and consider input/output sizes $D \in \{1, \ldots, 100\}$

**Initialization Schemes.** We consider three initialization schemes. The first is the standard parametrization (SP) available as the default parametrization for linear layers in PyTorch (He et al., 2015; LeCun et al., 2002). This is by far the most popular parametrization scheme in deep learning community with fan-in dependent initialization variance and layer independent learning rate, namely $\sigma_\ell^2 = \Theta(1/n_{\ell-1})$ and $\eta_\ell = \Theta(1)$, where $n_{\ell-1}$ is the fan-in of the weight matrix at layer $\ell$.

The second parameterization considered here is the neural tangent kernel parametrization (NTP) (Jacot et al., 2018). In this parametrization, the output of every layer is multiplied by a scalar $1/\sqrt{n_\ell}$, every layer is initialized from a standard normal distribution, and the learning rate is layer-independent.

Finally, we also consider maximal update parametrization ($\mu$P) (Yang et al., 2022; 2023). Here, the weight matrices are initialized with variance according to $\sigma_\ell^2 = \Theta(n_\ell/n_{\ell-1})$ and we also use layer-dependent learning rates $\eta_\ell = \Theta(1/n_{\ell-1})$.

**Data.** For experiments with linear networks, we sample a random dataset of (input, target) pairs by drawing an input vector $\mathbf{x} \in \mathbb{R}^d$ from a standard normal distribution and then setting $\mathbf{y} = \mathbf{W}\mathbf{x} + \epsilon$ for a fixed matrix $\mathbf{W} \in \mathbb{R}^{d \times d}$ and again a zero-mean Gaussian random vector $\epsilon \in \mathbb{R}^d$ with variance 0.01. For experiments with ReLU networks, we again draw an input vector $\mathbf{x} \in \mathbb{R}^d$ from a standard normal distribution, pass it through a single layer ReLU random teacher and add a zero-mean Gaussian random noise vector with variance 0.01. We fit the networks and compute the largest eigenvalue on a sample of size 1024.

**Training.** We fit the models such that the training loss is very close to zero. To do that, we train the networks using batched SGD (batch size= 128) starting with a fixed base learning rate $\eta = 10^{-2}$ (with the only exception of $\mu$P networks with large input/output sizes where we set initial $\eta = 1$) until the loss stops improving and then decrease the learning rate by dividing it by 10. We repeat this until a pre-defined minimum learning rate $\eta_{\min} = 10^{-5}$ is reached.

**Computing correlations.** For every configuration (so every point in figs. 4 and 15) we sample and fit 200 neural networks, compute the largest loss Hessian eigenvalue together with the consider parameter matrix statistics at initialization and after training. Then we compute Pearson correlation coefficient (PCC) between the largest Hessian eigenvalue and the considered quantity.

### D.1.4 CORRELATION BETWEEN LARGEST HESSIAN EIGENVALUE AND NORM OF WEIGHT MATRICES/PRODUCT OF WEIGHT MATRICES

**Model and Initialization.** In section 4.2, figs. 17, 17c, 18, 19 and 20, we analyze models of a form $\boldsymbol{f}_{\{\mathbf{W}_b\}_{b:1\leq i\leq B}} = \mathbf{W}_b \cdot \ldots \cdot \mathbf{W}_1 \mathbf{x}$, with $\mathbf{W}_b \in \mathbb{R}^{m \times m}$ for $b \notin \{1, B\}$, $\mathbf{W}_1 \in \mathbb{R}^{m \times D}$, and $\mathbf{W}_B \in \mathbb{R}^{D \times m}$. We consider models of width $m = 100$, depth $B \in \{2, 3, 4, 5, 6\}$ and input/output size $D \in \{1, 10\}$, and three model parametrizations: SP, $\mu$P and NTP (see appendix D.1.3 for details).

**Norms of Weight Matrix Products.** We consider two weight-matrix-based expressions that in the two-layer, unidimensional setting degenerate to simple vector norms, present in the expressions for the largest Hessian eigenvalue in eq. (1). Specifically, we consider simply the weight matrices $\mathbf{W}_i$ and a product of matrices with a single matrix removed, which we denote as $\widehat{\mathbf{W}}_i = \prod_{1 \leq j \leq B, j \neq i} \mathbf{W}_j$. For the correlation study, we consider squared norms of the resulting matrices, vectors and scalars. When we compute the norm of a matrix we resort to using spectral norm, knowing from fig. 4 that its square correlates the most with the largest Hessian eigenvalue.

**Data, Training and Correlation Study.** We sample the data, train the model and perform the correlation study in the same manner as we described in appendix D.1.3. The only exception are the starting base learning rates. For the input/output size $D = 1$, we start with learning rate $\eta = 10^{-2}$ for all settings except for deeper networks parametrized with SP and $\mu$P, where we set $\eta = 10^{-3}$.

### D.1.5 APPROXIMATING LARGEST HESSIAN EIGENVALUE WITH THE LARGEST GGN EIGENVALUE IN A GPT2 MODEL

To obtain appendix D.3.5 and figs. 5 and 24, we pre-train a 12-layer GPT-2 model with 124M parameters from scratch on the OpenWebText dataset for 10,000 steps, and compare it to the same architecture with attention layers removed and replaced by six additional MLP blocks. We follow the setup of Karpathy (2022) and repeat all experiments across three random seeds.

We measure the largest eigenvalues every 10 steps at the beginning of training, then every 100, and eventually every 1,000 steps. At each checkpoint, we use a random sample of 8 sequences from the training dataset and record the largest eigenvalue of both the Hessian and the GGN, the loss on these sequences and the average norm of the residual (the derivative of the loss with respect to the model output).

In appendix D.3.5, we show the loss and the relative approximation error averaged across seeds together with the standard error. In figs. 5 and 24, we plot the average approximation error as a function of the loss and the average residual. For each seed, we first interpolate the corresponding curves and then report the mean and standard error across seeds.

### D.1.6 INFLUENCE OF ATTENTION SINKS ON A HESSIAN SPECTRUM OF A GPT2 MODEL

In fig. 6 we study how the presence of an attention sink influences the Hessian spectrum. We conduct the experiment on the fitted GPT2 (124M) model from the nanoGPT repository (Karpathy (2022)).

**Introducing the attention sink.** We introduce the attention sinks across all Transformer blocks in the first attention heads, by adding a bias term on the first token to the pre-softmax attention scores. We add enough bias to make the average attention score on the first token and the first attention head across the Transformer blocks and the data used to compute the spectrum to jump from the original 0.32 to 0.81.

**Computing the spectrum.** To compute the whole spectrum we again use the curvlinops library Dangel et al. (2025), which implements the Lanczos spectrum approximation method Papyan (2020). We set the grid to have 1024 points, use 1024 vector Hessian products and set $\kappa = 3$.

### D.2 PROGRESSIVE SHARPENING AND THE EDGE OF STABILITY IN SELF-ATTENTION MODEL

We consider the same model as described in section 3.3 with $d_K = 2$. We initialize the weights from a zero-centered normal distribution with standard deviation 0.1. We train the model with gradient descent with learning rate set to $\eta = 2.3$. We demonstrate the results for this single learning rate but both progressive sharpening and EoS can be observed for a range of learning rates around the chosen $\eta$. The task we consider is sequence reversal. Specifically, for an input sequence $\mathbf{x} = (x_1 \ x_2)^\top$, the target sequence is $\mathbf{y} = (x_2 \ x_1)^\top$.

### D.3 ADDITIONAL RESULTS

#### D.3.1 EXPERIMENTS ACCOMPANYING THEORETICAL RESULTS IN SECTION 3

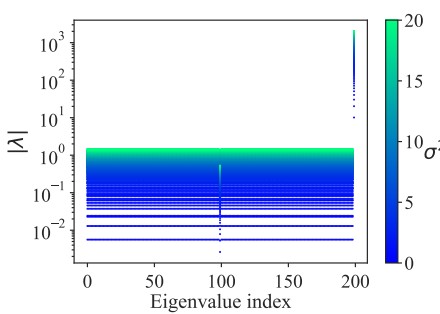

Figure 7: Impact of the data scale on the Hessian spectrum at initialization in the random Gaussian regression task. The eigenvalues are ordered from the most negative to the most positive. We distinguish two outliers and the bulk as per theorem 3.1.

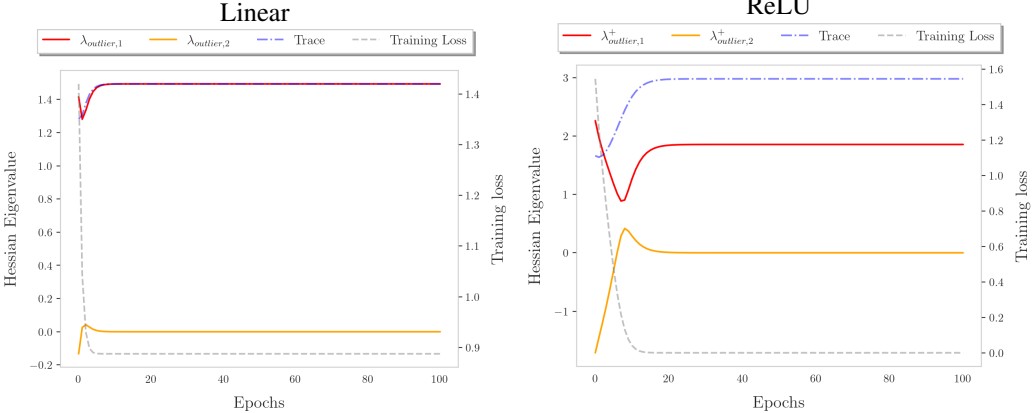

Figure 8: The paired nature of outlying eigenvalues throughout training. For the ReLU network, the outlying pair of eigenvalues corresponding to the positive cell are shown. Both the cases are with gradient descent on a synthetic wedge dataset, with a single hidden layer network of width 10 using a learning rate of 0.18 and momentum 0.9.

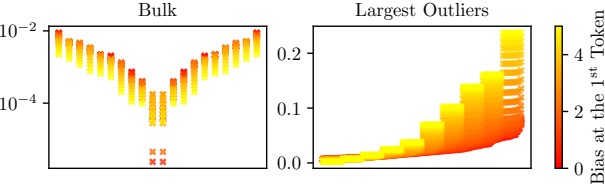

Figure 9: The larger the attention sink, the smaller the bulk and the larger the outlier eigenvalues, also for embedding dimension larger than 1.

#### D.3.2 PAIRED NATURE OF OUTLIER EIGENVALUES

**Complementary Results for Two-layer Linear Networks**

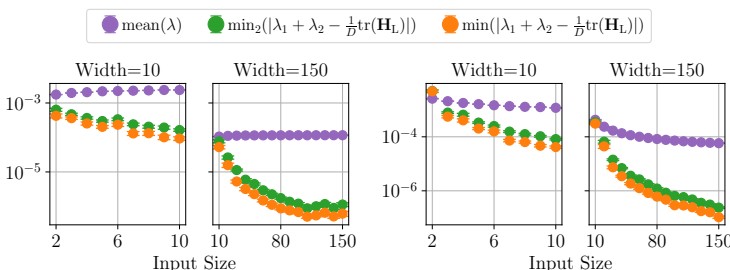

(a) Orthogonal parametrization, normalized data.

(b) Standard parametrization, decorrelated data

Figure 10: Smallest differences between the paired eigenvalues and $\frac{1}{D}\operatorname{tr}(\mathbf{H}_L)$ compared to the mean eigenvalue across varying input sizes in two-layer linear networks.

**Four-Layer Linear Networks**

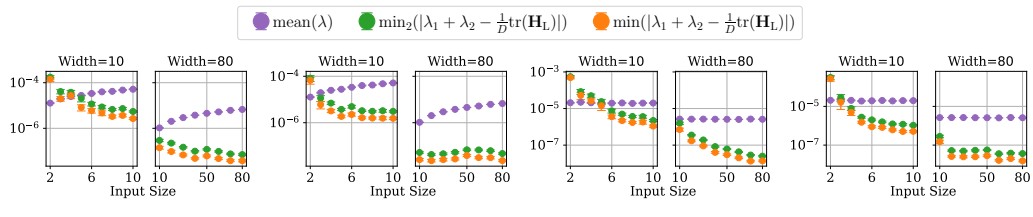

(a) Orthogonal param., decorrelated data.

(b) Orthogonal param., normalized data.

(c) Standard param., decorrelated data.

(d) Standard param., normalized data.

Figure 11: Smallest differences between the paired eigenvalues and $\frac{1}{D}\operatorname{tr}(\mathbf{H}_L)$ compared to the mean eigenvalue across varying input sizes in four-layer linear networks.

**Two-Layer ReLU Networks**

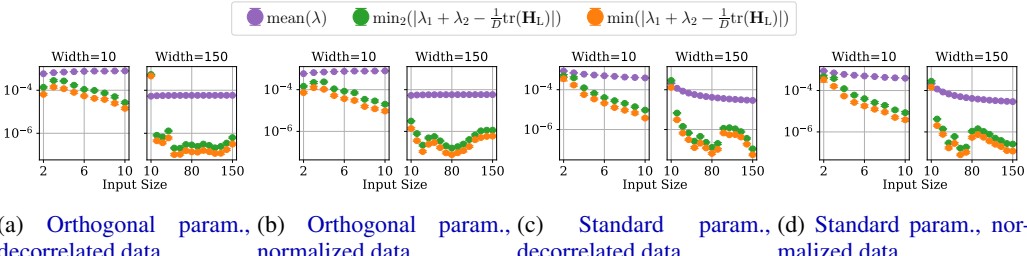

(a) Orthogonal param., decorrelated data.

(b) Orthogonal param., normalized data.

(c) Standard param., decorrelated data.

(d) Standard param., normalized data.

Figure 12: Smallest differences between the paired eigenvalues and $\frac{1}{D}\operatorname{tr}(\mathbf{H}_L)$ compared to the mean eigenvalue across varying input sizes in two-layer ReLU networks.

**Four-Layer ReLU Networks**

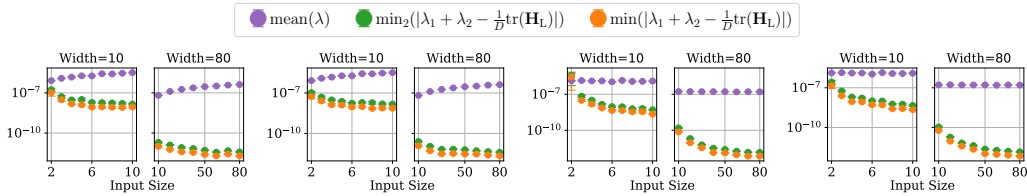

(a) Orthogonal param., decorrelated data. (b) Orthogonal param., normalized data. (c) Standard param. decorrelated data. (d) Standard param., normalized data.

Figure 13: Smallest differences between the paired eigenvalues and $\frac{1}{D} \operatorname{tr}(\mathbf{H}_L)$ compared to the mean eigenvalue across varying input sizes in four-layer ReLU networks.

### D.3.3 CORRELATION BETWEEN LARGEST LOSS HESSIAN EIGENVALUE AND THE SQUARED WEIGHT MATRIX NORM AND WEIGHT MATRIX ALIGNMENT

**Linear Networks**

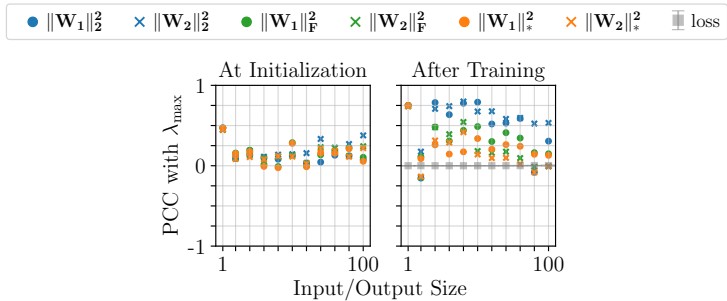

Figure 14: Pearson correlation between the largest Hessian eigenvalue and the squared norm (spectral, Frobenius, nuclear) of the weight matrices at initialization and at the end of training for linear networks with neural tangent parametrisation (NTP). We again note that it is the spectral norm that correlates most with the largest Hessian eigenvalue, and depending on paramterization it is a different matrix that drives the correlation.

**Why Squared Spectral Norm?** It is possible to formally reason why the spectral norm correlates the most with the largest eigenvalue if we assume that the Hessian is block-diagonal dominant, meaning that we can neglect the cross-layer blocks of the Hessian. Note that it has been established in the literature that the loss Hessian can become block-diagonal dominant, especially by the end of training (Dong et al., 2025). The success of block-diagonal preconditioners, like KFAC (Martens & Grosse, 2015a), also suggests this approximate block-diagonal dominance.

Let us consider a network $\boldsymbol{f}_{\{\mathbf{W}_1, \mathbf{W}_2\}} = \mathbf{W}_2 \mathbf{W}_1 \mathbf{x}$, with $\mathbf{W}_1 \in \mathbb{R}^{m \times D}$ and $\mathbf{W}_2 \in \mathbb{R}^{D \times m}$ and an MSE loss. According to Singh et al. (2021), the blocks on the Hessian diagonal are given by

$$\mathbf{H}_L^{11} = \frac{\partial^2 \ell}{\partial \mathbf{W}_1 \partial \mathbf{W}_1} = \mathbf{W}_2^\top \mathbf{W}_2 \otimes \Sigma_{\mathbf{xx}}$$

$$\mathbf{H}_L^{22} = \frac{\partial^2 \ell}{\partial \mathbf{W}_2 \partial \mathbf{W}_2} = \mathbf{I}_D \otimes \mathbf{W}_1 \Sigma_{\mathbf{xx}} \mathbf{W}_1^\top,$$

where $\Sigma_{\mathbf{xx}} = \frac{1}{n} \sum_{i=1}^n \mathbf{x}_i \mathbf{x}_i^\top \in \mathbb{R}^{D \times D}$. If we further assume that the input data is decorrelated, so $\Sigma_{\mathbf{xx}} = \mathbf{I}_D$, and note that the set of eigenvalues of a block diagonal matrix is given by the union of

the blocks eigenvalues, we obtain that when the Hessian is block-diagonal,

$$
\begin{aligned}
\lambda_{\max}(\mathbf{H}_{\mathrm{L}}) &= \max\left\{\lambda_{\max}\left(\mathbf{W}_2^\top \mathbf{W}_2 \otimes \mathbf{I}_D\right), \lambda_{\max}\left(\mathbf{I}_D \otimes \mathbf{W}_1 \mathbf{W}_1^\top\right)\right\} \\
&= \max\left\{\lambda_{\max}\left(\mathbf{W}_2^\top \mathbf{W}_2\right), \lambda_{\max}\left(\mathbf{W}_1 \mathbf{W}_1^\top\right)\right\} \\
&= \max\left\{\sigma_{\max}^2\left(\mathbf{W}_2\right), \sigma_{\max}^2\left(\mathbf{W}_1\right)^2\right\} = \max\left\{\|\mathbf{W}_2\|_2^2, \|\mathbf{W}_1\|_2^2\right\}.
\end{aligned}
$$

Note, even if the Hessian is not block-diagonal, it will at least be lower-bounded by the squared spectral norm of the weight matrices.

Overall, this formally suggests that the largest Hessian eigenvalue for a linear two-layer network is driven by the squared spectral norm of the weight matrices.

**Impact of Initialization** In section 4.2 we established that from all considered norms, it is the squared spectral norm that correlates most with the sharpness and which matrix drives this correlation depends on the chosen network parametrization. We hypothesize that, at initialization, the discrepancy in magnitude of the entries in the weights drives the correlation. To see that, note that whenever the first weight matrix is initialized with larger magnitude values than the second matrix, which happens for SP and $\mu$P for small $D$, it is the norm of the first matrix that exhibits a significant correlation with the largest eigenvalue. Similarly, when both matrices are initialized with similar magnitude values (NTP, as well as $\mu$P and SP for the larger D), none of them correlates strongly with the largest Hessian eigenvalue, but we observe a larger correlation with weight matrix alignment (see fig. 15).

At convergence, for the SP it is the second weight matrix that drives the correlation, while for the maximal update parametrization ($\mu$P) it is the second one. For the neural tangent kernel parametrization (NTP) the largest singular values of both matrices correlate well with the largest eigenvalue. In networks with scalar output Wang et al. (2022) theoretically observed and empirically confirmed that the largest eigenvalue of the NTK and the norm of the output layer weight vector change in the same direction most of the time along the gradient descent trajectory, i.e. they both increase or decrease at the same time. We extend this observation because we consider networks of any output size; we also note that it can be the norm of both matrices that strongly correlates with the largest eigenvalue and that the initialization influences which one it is.

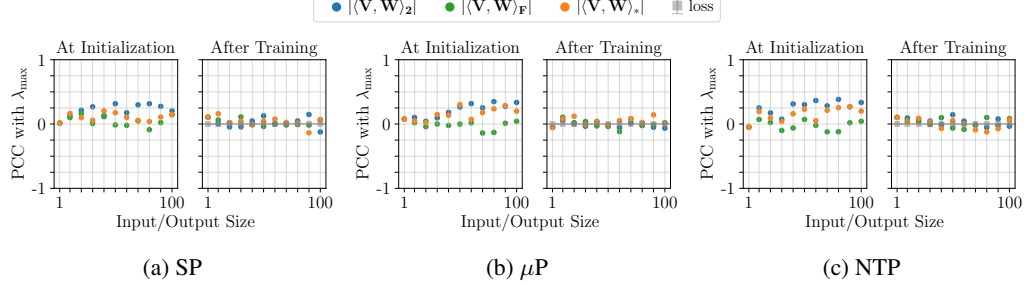

(a) SP      (b) $\mu$P      (c) NTP

Figure 15: Pearson correlation between the largest Hessian eigenvalue and the absolute alignment (spectral, Frobenius, nuclear) of the weight matrices at initialization and at the end of training across three model parametrizations in linear networks.

**ReLU Networks**

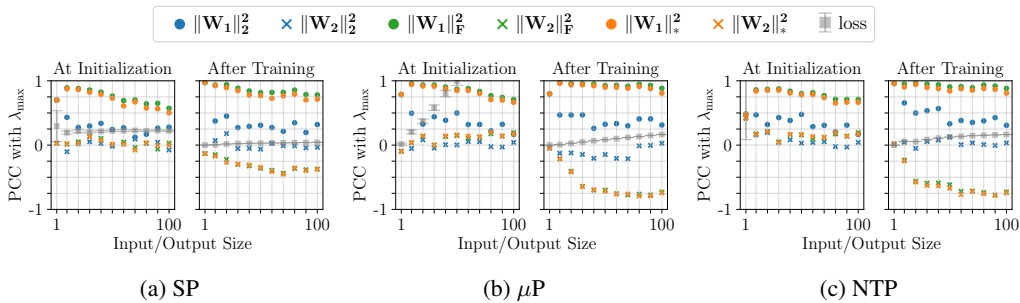

(a) SP

(b) $\mu$P

(c) NTP

Figure 16: Pearson correlation between the largest Hessian eigenvalue and the squared weight matrix norms (spectral, Frobenius, nuclear) of the weight matrices at initialization and at the end of training across three model parametrizations in ReLU networks.

### D.3.4 Correlation between largest loss Hessian eigenvalue and squared weight matrix norms in deep linear MLPs

**Linear Networks**

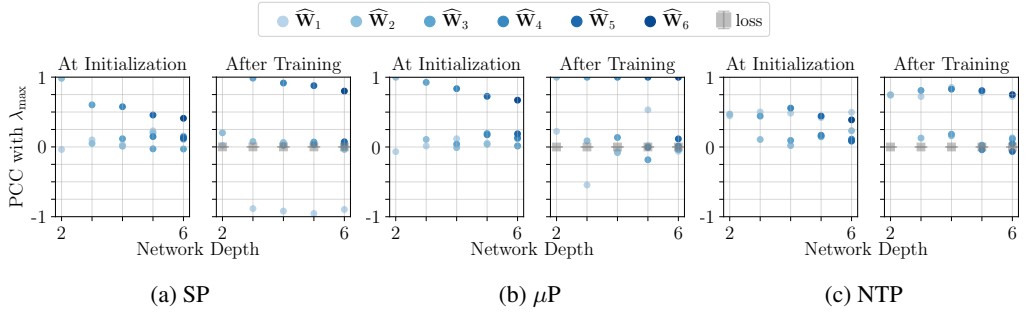

(a) SP

(b) $\mu$P

(c) NTP

Figure 17: Pearson correlation coefficient between sharpness and the squared norm of the weight matrix products $\widehat{\mathbf{W}}_b$ under SP, $\mu$P, NTP for linear networks of depth $B \in \{2, 3, 4, 5, 6\}$ and input/output size $D = K = 1$. Squared norm of the products of parameter matrices excluding the last one correlate well with the largest Hessian eigenvalues across different depths and model parametrizations.

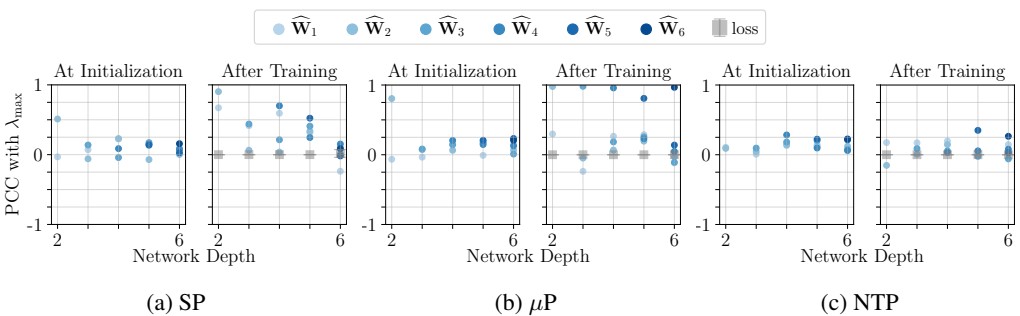

(a) SP

(b) $\mu$P

(c) NTP

Figure 18: Pearson correlation coefficient between sharpness and the squared norm of the weight matrix products $\widehat{\mathbf{W}}_b$ under SP, $\mu$P, NTP for linear networks of depth $B \in \{2, 3, 4, 5, 6\}$ and input/output size $D = K = 10$. Squared norm of the products of parameter matrices excluding the last one correlate well with the largest Hessian eigenvalues across different depths and model parametrizations.

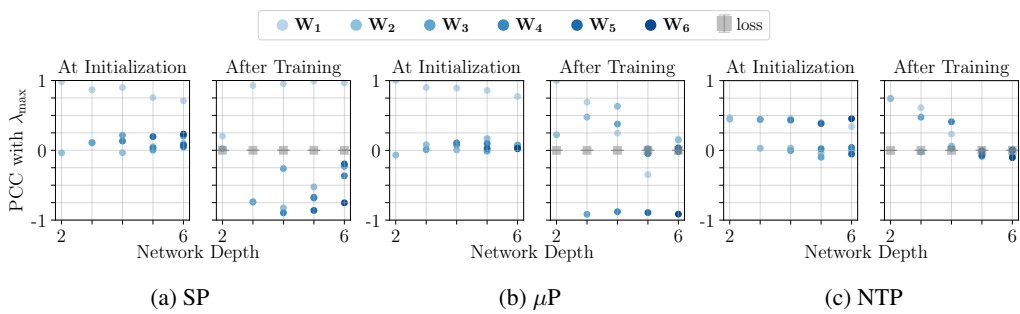

Figure 19: Pearson correlation coefficient between sharpness and the squared spectral norm of the weight matrices $\mathbf{W}_b$ under SP, $\mu$P, NTP for linear networks of depth $B \in \{2, 3, 4, 5, 6\}$ and input/output size $D = K = 1$.

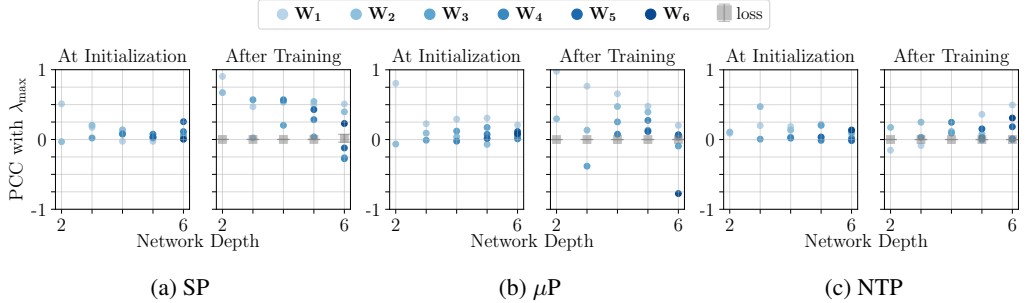

Figure 20: Pearson correlation coefficient between sharpness and the squared spectral norm of the weight matrices $\mathbf{W}_b$ under SP, $\mu$P, NTP for linear networks of depth $B \in \{2, 3, 4, 5, 6\}$ and input/output size $D = K = 10$.

**ReLU Networks**

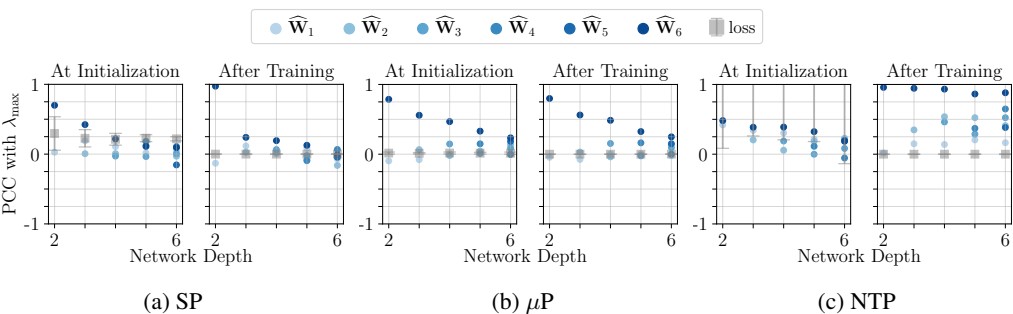

Figure 21: Pearson correlation coefficient between sharpness and the squared norm of the weight matrix products $\widehat{\mathbf{W}}_b$ under SP, $\mu$P, NTP for ReLU networks of depth $B \in \{2, 3, 4, 5, 6\}$ and input/output size $D = K = 1$.

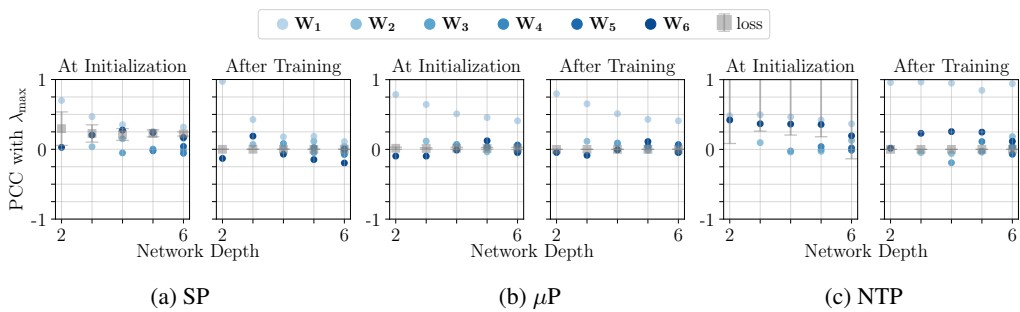

(a) SP $\qquad$ (b) $\mu$P $\qquad$ (c) NTP

Figure 22: Pearson correlation coefficient between sharpness and the squared Frobenius norm of the weight matrices $\mathbf{W}_b$ under SP, $\mu$P, NTP for ReLU networks of depth $B \in \{2, 3, 4, 5, 6\}$ and input/output size $D = K = 1$.

### D.3.5 PRETRAINING CURVES FOR GPT2 AND GPT2 WITHOUT ATTENTION LAYERS

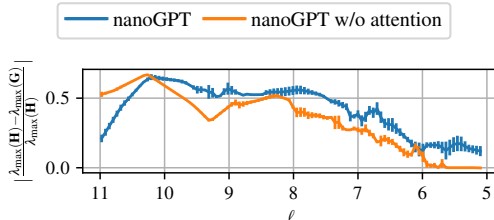

Figure 23: Relative approx. error vs the loss value.

Figure 24: Relative approximation error of the largest Hessian eigenvalue by the largest GGN eigenvalue while pretraining GPT2 (124M) and similarly-sized model without attention layers. The curves show mean and standard error averaged across 3 pretraining runs.

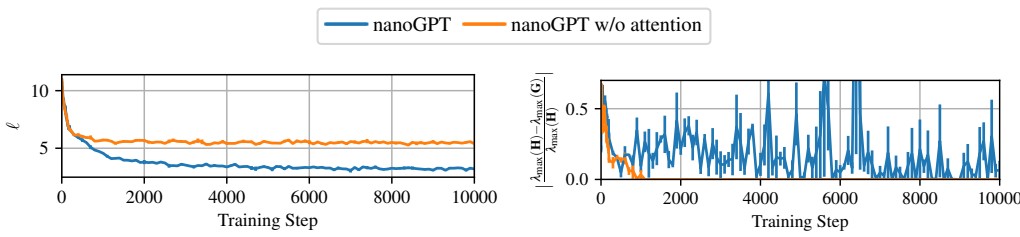

Figure 25: Loss value and relative approximation error of the largest Hessian eigenvalue by the largest GGN eigenvalue while pretraining GPT2 (124M) and similarly-sized model without attention layers. The curves show mean and standard error averaged acrosss 3 pretraining runs.

### D.4 CORRELATION BETWEEN LOSS HESSIAN TRACE AND THE SQUARED WEIGHT MATRIX NORM

In this section we discuss which matrix correlates the most with the loss Hessian trace in the two-layer linear networks with arbitrary input and output dimension. Specifically, we consider a network $f_{\{\mathbf{W}_1, \mathbf{W}_2\}} = \mathbf{W}_2 \mathbf{W}_1 \mathbf{x}$, with $\mathbf{W}_1 \in \mathbb{R}^{m \times D}$ and $\mathbf{W}_2 \in \mathbb{R}^{D \times m}$ and an MSE loss. Let us note that the loss Hessian trace depends only on the trace of its diagonal blocks $\mathbf{H}_L{}^{11}$ and $\mathbf{H}_L{}^{22}$.

According to Singh et al. (2021), these blocks are given by

$$\mathbf{H}_{\mathrm{L}}^{11} = \frac{\partial^2 \ell}{\partial \mathbf{W}_1 \partial \mathbf{W}_1} = \mathbf{W}_2^\top \mathbf{W}_2 \otimes \Sigma_{\mathbf{xx}}$$

$$\mathbf{H}_{\mathrm{L}}^{22} = \frac{\partial^2 \ell}{\partial \mathbf{W}_2 \partial \mathbf{W}_2} = \mathbf{I}_D \otimes \mathbf{W}_1 \Sigma_{\mathbf{xx}} \mathbf{W}_1^\top,$$

where $\Sigma_{\mathbf{xx}} = \frac{1}{n} \sum_{i=1}^n \mathbf{x}_i \mathbf{x}_i^\top \in \mathbb{R}^{DD}$. Assuming that the features are decorrelated, so $\Sigma_{\mathbf{xx}} \simeq \mathbf{I}_D$, we can approximate the Hessian trace as

$$\mathrm{tr}\left(\mathbf{H}_{\mathrm{L}}\right) = \mathrm{tr}\left(\mathbf{H}_{\mathrm{L}}^{11}\right) + \mathrm{tr}\left(\mathbf{H}_{\mathrm{L}}^{22}\right) \simeq \mathrm{tr}\left(\mathbf{W}_2^\top \mathbf{W}_2 \otimes \mathbf{I}_D\right) + \mathrm{tr}\left(\mathbf{I}_D \otimes \mathbf{W}_1 \mathbf{W}_1^\top\right)$$

$$= D\left(\mathrm{tr}\left(\mathbf{W}_2^\top \mathbf{W}_2\right) + \mathrm{tr}\left(\mathbf{W}_1 \mathbf{W}_1^\top\right)\right) = D\left(\|\mathbf{W}_1\|_{\mathrm{F}}^2 + \|\mathbf{W}_2\|_{\mathrm{F}}^2\right).$$

In fig. 26, we experimentally demonstrate that, while all considered squared weight norms correlate strongly with Hessian trace, it is indeed the squared Frobeenius norm that correlates the most.

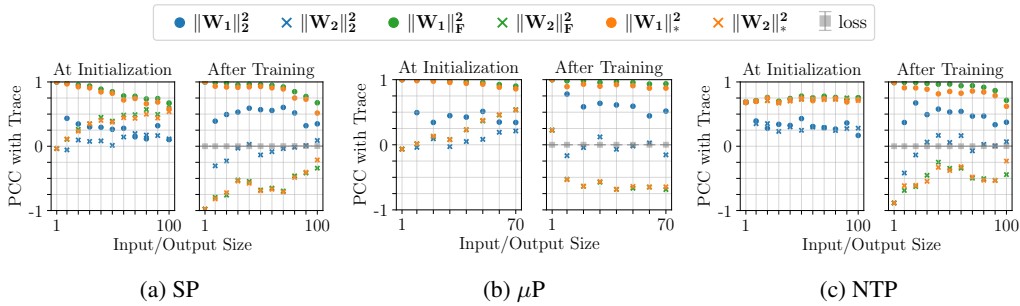

(a) SP           (b) $\mu$P           (c) NTP

Figure 26: Pearson correlation between the Hessian trace and the square weight matrix norms (spectral, Frobenius, nuclear) of the weight matrices at initialization and at the end of training across three model parametrizations in linear networks.

### D.5 PROGRESSIVE SHARPENING AND THE EDGE OF STABILITY IN SELF-ATTENTION MODEL

In this section we demonstrate preliminary numerical results concerning progressive sharpening and the edge of stability (EoS) (Cohen et al., 2020; Jastrzębski et al., 2018), that we observe in a model consisting of a single self-attention layer with embedding dimension equal to 1 (see section 3.3).

Progressive sharpening is the phenomenon in deep learning optimization where the sharpness (largest loss Hessian eigenvalue) consistently increases during the initial stages of training with gradient descent. The EoS is the subsequent regime where the sharpness reaches and then oscillates around the critical threshold of $2/\eta$, where $\eta$ is the learning rate. Simultaneously the loss to continues decreasing over long timescales despite local non-monotonic behavior.

In fig. 27 we demonstrate both phenomena in the model discussed in section 3.3. Detailed experimental setup can be found in appendix D.2. We note that the progressive sharpening phase coincides with growing trace of the Hessian (bottom-left panel of fig. 27), which is the first summand in the sharpness upper and lower bound from eq. (121). Similar progressive growth can be observed in the norms of the parameter vectors and self-attention moment vectors (bottom- and middle-right panels of fig. 27), which define the Hessian trace (again see eq. (121)).

Interestingly, during the EoS the oscillations of sharpness and trace are in a completely opposite phases, suggesting that the oscillations are driven by terms defining $\xi$ from eq. (121). This could include the alignment between query and key vectors and the loss residual. Indeed the amplitude of the oscillations seems to be decreasing similarly to the norm of the loss residual (top-right panel of fig. 27). We leave a detailed mathematical description of these dynamics under gradient descent to future work.

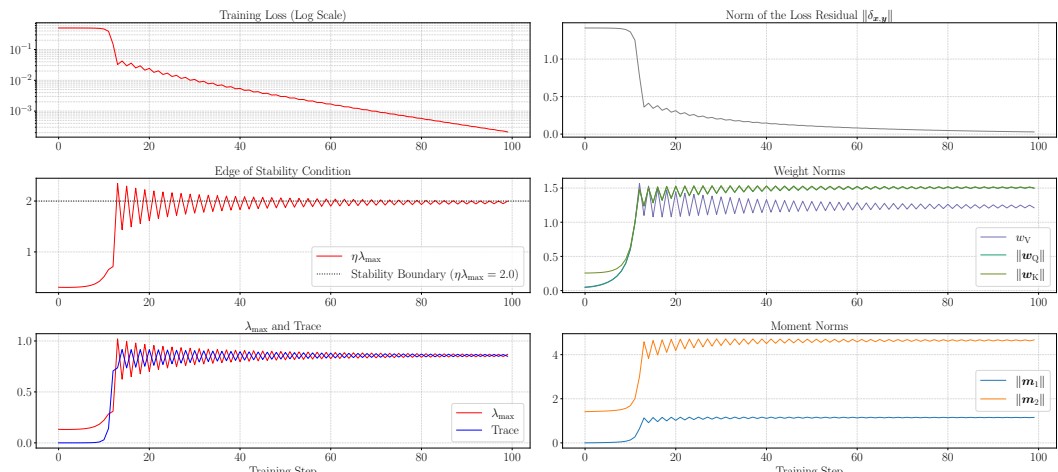

Figure 27: Model consisting of a self-attention layer with input dimension 1 with MSE loss exhibits progressive sharpening and the edge of stability phenomena. Experimental setup can be found in appendix D.2.

