# OpenReview forum: "Cracking the Hessian: Closed-Form Hessian Spectra for Fundamental Neural Networks"
_ICLR.cc/2026/Conference — Submitted to ICLR 2026_

### Official Review · Reviewer_iw5q · 2025-10-15

**Soundness:** 3
**Presentation:** 3
**Contribution:** 3
**Rating:** 4
**Confidence:** 3

**Summary:**

This paper derives closed-form expressions for Hessian eigenvalues and eigenvectors for two-layer linear and ReLU networks with scalar input, and provides closed-form eigenvalues for a single self-attention layer. The work reveals a previously undiscovered paired structure of outlier eigenvalues and explores correlations between weight matrix norms and Hessian eigenvalues in more complex architectures.

**Strengths:**

- The theoretical contributions represent the first successful derivation of complete, exact Hessian spectra for non-trivial neural networks without approximations. The closed-form results for linear networks (Theorem 3.1), ReLU networks (Theorem 3.4), and self-attention layers (Theorem 3.5) provide valuable analytical tools for understanding loss landscapes, with the paired eigenvalue structure offering new insights into Hessian behavior throughout training.
  - The empirical validation extends beyond theoretical assumptions to practical architectures, including GPT2 models. The correlation studies between weight matrix spectral norms and Hessian eigenvalues (Section 4.2) provide actionable insights that appear to hold across different parametrizations and network depths, while experiments on attention sinks offer concrete guidance for Transformer architecture design.

**Weaknesses:**

- The theoretical results are restricted to highly simplified settings (scalar input/output for linear and ReLU networks, single self-attention layers), which limits their direct applicability. While the paper attempts to extend results to multi-dimensional cases in the appendix (Section B.4), these extensions require strong structural assumptions on weight matrices (e.g., block diagonal or specific factorizations) that may rarely hold in practice. The gap between the theoretical models (e.g., f(x) = ⟨w,v⟩x) and practical deep networks is substantial, and the paper would benefit from more explicit discussion of when these assumptions are reasonable approximations versus when they fundamentally break down.
  - The connection between the closed-form results and the empirical observations in Section 4 lacks rigor. For instance, the correlation studies in Section 4.2 show that spectral norms of weight matrices correlate with Hessian eigenvalues, but there is no formal analysis explaining why this correlation exists or under what conditions it should hold. Similarly, the claim that GGN provides worse approximations for Transformers than MLPs (Section 4.3) is based only on GPT2 experiments with limited analysis of the underlying mechanisms. Adding theoretical justification or at least formal conjectures would strengthen these empirical findings.
  - The paired eigenvalue structure (Section 4.1), while interesting, receives insufficient analysis regarding its practical implications. The paper demonstrates that eigenvalue pairs sum to approximately (1/D)tr(H) but does not explain what this means for optimization dynamics, model training, or practical algorithm design. For example: Does this pairing affect convergence rates? Does it inform choices of learning rates or optimization algorithms? Could it be exploited for better Hessian approximation methods? The paper would benefit from exploring these questions or at least providing concrete examples where the paired structure matters for practical applications.
  - The experiments validating the paired eigenvalue structure beyond theoretical assumptions (Figure 3) show that the pairing becomes approximate rather than exact. The paper reports differences on the order of 10^(-3) to 10^(-5) but does not analyze when or why these approximations degrade. More importantly, there is no investigation of how network depth, nonlinearities, or other architectural choices affect the quality of this approximation. Understanding these failure modes would be valuable for practitioners and could guide future theoretical work. I will reconsider my score in the rebuttal.

**Questions:**

see weaknesses

---

> ### Author Response · Authors · 2025-11-22
> **Response (1/3) to Reviewer iw5q**
>
> We thank the reviewer for the helpful feedback. We answer specific points and questions below, and mark changes introduced in the manuscript in blue.
>
> > The theoretical results are restricted to highly simplified settings [...], which limits their direct applicability.
>
> We address this point in detail in our general response to all reviewers. Here, we would like to highlight that we empirically observed that the conclusions drawn from the simple models do, at least approximately, transfer to the more advanced settings. This includes but is not limited to the observation about the nature of the paired eigenvalues, the dependence of the largest eigenvalue on the parameter norm, insights about relationship between the outlier eigenvalues and attention sinks, etc.. We can also see empirical observations from prior work reflected in our expressions, for instance [1] note that batch normalization, which acts as a normalization within a network, can suppress outlier eigenvalues. In eq. 2, we can explicitly see that data normalization would indeed contribute to controlling the magnitude of the largest eigenvalue.
>
> We also want to stress that analysis of simplified settings to draw conclusions about more complex systems is a common approach in deep learning theory [2,3,4]. In the context of the Hessian eigenvalues, due to the complexity of the problem, significantly simpler models than ours, like 4-parameter scalar “neural networks”, have been considered [5].
>
>
> > The gap between the theoretical models (e.g., f(x) = ⟨w,v⟩x) and practical deep networks is substantial, and the paper would benefit from more explicit discussion of when these assumptions are reasonable approximations versus when they fundamentally break down.
>
> It is difficult to state a precise but general rule for when these simplified systems serve as reasonable approximations and when not. Our results can be viewed as a solvable baseline: when a researcher observes a Hessian spectral phenomenon in a deep network (e.g., progressive sharpening or EoS), they should first verify if the same phenomenon exists in our simple models. If it does, the derived expressions provide mechanistic, precise insight into the phenomenon's fundamental drivers. To give an example of this, we have verified that progressive sharpening and edge of stability [3] are present in the self-attention layer we study in section 3.3. This verification has been added to the Appendix D.5.
>
> However, we believe our results could be particularly useful for studying the effects of initialization, width, and data characteristics on the neural network landscape, as well as the role of softmax/activations in Transformers. These concepts are explicitly captured by the models considered in this manuscript. While our models do not explicitly capture network depth, some core phenomena, like the effect of data normalization on the spectrum [1], the dependence of the eigenvalues on the network parameters (section 4.2), and now also the approximate paired eigenvalues (see our answer to your request below) persist in deeper architectures.
>
> In lines 510-519 we have extended the conclusion to include this discussion, as suggested by the reviewer.
>
> > The correlation studies in Section 4.2 show that spectral norms of weight matrices correlate with Hessian eigenvalues, but there is no formal analysis explaining why this correlation exists or under what conditions it should hold.
>
> In Appendix D.3.3 (around line 2413), we now provide a formal explanation for why the spectral norm of the weights correlates with the largest eigenvalue of the linear MLP. Our derivation relies on the approximation that the Hessian structure is block-diagonal, an assumption frequently made when designing preconditioners [6, 7]. The appearance of this per-layer block-dominant structure has also recently been studied in [8] and is especially likely by the end of training.
>
> To complete the empirical analysis, we have also extended the experiments from Section 4.2 to include ReLU networks (see Appendix D.3.3). We find that although the squared spectral norm shows non-negligible correlation with the largest Hessian eigenvalue (between $0.25$ and $0.75$ for the first weight matrix across different parametrizations), the squared Frobenius norm correlates most strongly. This suggests an interesting research direction regarding how activations influence the most-correlating norm, which we leave as future work. To make it clear that section 4.2 discusses results specific for linear networks, we changed its title accordingly and provided a link to the ReLU results in line 432.

---

> ### Author Response · Authors · 2025-11-22
> **Response (2/3) to Reviewer iw5q**
>
> > The claim that GGN provides worse approximations for Transformers than MLPs (Section 4.3) is based only on GPT2 experiments with limited analysis of the underlying mechanisms.
>
> We emphasize that this hypothesis originated from the theoretical analysis in Section 3.3 (lines 347–354), specifically where we compare the largest eigenvalue of a single self-attention layer to that of a simple linear/ReLU network. We observe that, unlike linear or ReLU networks, the expression for the largest eigenvalue in self-attention depends heavily on the residual $\delta_{x, y}$. These terms arise from the 'remainder matrix' (the difference between the full Hessian and the GGN)—which we term the functional Hessian—and are responsible for the reduced approximation quality.
>
> Intuitively, because Softmax is highly non-linear compared to standard activations like ReLU, it results in a 'richer' structure of the network’s second order derivative, which is fully captured only by the functional Hessian. This leads to a larger contribution from the functional Hessian and a weaker contribution from the GGN. We summarised this intuition in lines 462-466 of the updated manuscript.
>
> >  Adding theoretical justification or at least formal conjectures would strengthen these empirical findings.
>
> As mentioned in the previous two points, in the updated version of the manuscript, we now provide preliminary justification for the correlation of the largest eigenvalue with the squared spectral norm of the weights in linear networks. We also provide further justification for GGN providing worse eigenvalue approximation for Transformers than food MLPs.
>
> > The paired eigenvalue structure (Section 4.1), while interesting, receives insufficient analysis regarding its practical implications. The paper demonstrates that eigenvalue pairs sum to approximately (1/D)tr(H) but does not explain what this means for optimization dynamics, model training, or practical algorithm design. For example: Does this pairing affect convergence rates? Does it inform choices of learning rates or optimization algorithms? Could it be exploited for better Hessian approximation methods? The paper would benefit from exploring these questions or at least providing concrete examples where the paired structure matters for practical applications.
>
> * In Appendix C, we do provide a discussion of how the ranges of these eigenvalue-pairs demarcate a notion of early, late, and divergent phase in training, with these phases also accompanied by changes in the landscape conditioning. The condition number can be lower bounded through the ratio of their magnitudes, which can be calculated cheaply through the link with the trace. Furthermore, this provides a cheap way to check if the network might be stuck at a saddle point or if it is in a convex basin. We’re happy to add this discussion to the main text if the reviewer thinks that would strengthen the paper.
> * At a more fundamental level, this also shows an instance of tight dependencies between outlier eigenvalues that could better inform the choice of RMT models used to analyze the Hessian.
>
>
> > The experiments validating the paired eigenvalue structure beyond theoretical assumptions (Figure 3) show that the pairing becomes approximate rather than exact. The paper reports differences on the order of 10^(-3) to 10^(-5) but does not analyze when or why these approximations degrade.
>
> We want to clarify that in Appendix D.3.2, we empirically analyze how initialization and data preprocessing procedures affect the quality of the approximation. These correspond to the theoretical assumptions satisfied in Figure 3a but relaxed in Figure 3b. By studying these factors separately, we find that both are crucial for exactness; however, the data decorrelation assumption is the most critical, as its violation leads to a significantly larger deviation
>
> > [.. ] there is no investigation of how network depth, nonlinearities, or other architectural choices affect the quality of this approximation.
>
> We thank the reviewer for this suggestion. In appendix D.3.2 we now provide the results of the paired-eigenvalue experiments for ReLU networks as well as deeper (4-layer) linear and ReLU networks. We observe that the quality of the approximation improves with larger depth, width and input/output size. The presence of ReLU also does not degrade the quality of the approximation.

---

> ### Author Response · Authors · 2025-11-22
> **Response (3/3) to Reviewer iw5q**
>
> We hope this resolves all reviewer’s concerns, and we are happy to address any other comments and questions. We believe the manuscript has improved significantly thanks to reviewer’s feedback and we kindly invite the reviewer to reevaluate the paper.
>
> References:
>
> [1] Ghorbani et al., An investigation into neural net optimization via hessian eigenvalue density, 2019, https://arxiv.org/abs/1901.10159
>
> [2] Saxe et al. Exact solutions to the nonlinear dynamics of learning in deep linear neural networks, ICLR 2014, https://arxiv.org/abs/1312.6120
>
> [3] Cohen et al., Gradient Descent on Neural Networks Typically Occurs at the Edge of Stability, ICLR 2021, https://arxiv.org/abs/2103.00065
>
> [4] von Oswald et al., Transformers Learn In-Context by Gradient Descent, ICML 2023, https://arxiv.org/abs/2212.07677
>
> [5]  Zhu et al., Understanding Edge-of-stability Training Dynamics with a Minimalist Example, ICLR 2023, https://openreview.net/pdf?id=p7EagBsMAEO
>
> [6] Martens & Grosse, Optimizing Neural Networks with Kronecker-factored Approximate Curvature, 2015, https://arxiv.org/abs/1503.05671
>
> [7]  Yao et al., AdaHessian: An Adaptive Second Order Optimizer for Machine Learning, 2021, https://arxiv.org/pdf/2006.00719
>
> [8] Dong et al., Towards Quantifying the Hessian Structure of Neural Networks, 2025, https://arxiv.org/pdf/2505.02809

---

### Official Review · Reviewer_Lt5D · 2025-10-28

**Soundness:** 3
**Presentation:** 3
**Contribution:** 3
**Rating:** 4
**Confidence:** 3

**Summary:**

This paper first derivied a closed-form formulae for the complete Hessian spectrum of Linear, RELU and self-attention neural networks using a proof technique. The study extended the formulae towards a `paired' structure of outlier eigenvalues, and the correlation between the largest eigenvalue and the spectral norm of weight matrices.

**Strengths:**

1. This paper extended previous studies on the Hessian spectrum structure to several novel theoretical assumptions and insights verified by empirical studies, demonstrating a comprehensive in-depth understanding.
2. The theorem proof is clear and intriguing, with a new perspevtive on the two paired outlying eigenvalues persisting across different neural architectures.

**Weaknesses:**

The major theorem on closed-form Hessian is derived upon input/output or embedding dimension equal to 1; while extended in the Appendix and verified with empirical observations beyond these assumptions, the strength of the theoretical explanation remains vague.

**Questions:**

1. Upon the paired eigenvalue, would their value or sum correlates stronger to the spectral norm, as a measure of sharpness than only the largest eigenvalue, as theorem 3.1 suggested?
2. In recent empirical works , roughly ~10 eigenvalue outliers is observed in teh complete Hessian spectrum across a variety of models, neigher the top-2 paired eigenvalues does not add up to the Hessian trace, could you elaborate your theorem on their observations?
3. Is it possible to derive a two-layer neural network Hessian Lemma under the assumption that the input and/or output dimension equal to 2, or the theorem holds for dimension k+1 if proved for dimension k?

---

> ### Author Response · Authors · 2025-11-22
> **Response to Reviewer Lt5D**
>
> We thank the reviewer for their feedback. We have addressed the weakness in our general response to all reviewers, and we provide a point-by-point response to specific questions below. Please note that all changes in the manuscript are highlighted in blue.
>
> > Upon the paired eigenvalue, would their value or sum correlates stronger to the spectral norm, as a measure of sharpness than only the largest eigenvalue, as theorem 3.1 suggested?
>
> Since the sum should be quite close to a fraction of Hessian trace, as suggested by our theoretical results and experiments from section 4.1, we can answer this question by finding the norm with which the Hessian trace should correlate the most.
>
> It turns out, that if we consider a two-layer linear network of arbitrary input and output dimension $D$, and assume that the empirical correlation matrix of the input samples is close to identity, we can show that the Hessian trace is given by $D (\|W_1\|^2_F + \|W_2\|^2_F)$. We present this derivation together with a numerical experiment confirming this insight in appendix D4 of the updated version of the manuscript. While the correlation with the squared Frobenius norm dominates in this case, we still empirically observe a strong correlation with the squared spectral norm.
>
> > In recent empirical works , roughly ~10 eigenvalue outliers is observed in teh complete Hessian spectrum across a variety of models, neigher the top-2 paired eigenvalues does not add up to the Hessian trace, could you elaborate your theorem on their observations?
>
> We would like to point out that our claim is not that two outliers should sum up to the trace — this is generally true for all eigenvalues, i.e. all Hessian eigenvalues sum to its trace. Our theory suggests that there are two outlier eigenvalues that sum to $\frac{tr(H_L)}{D}$ where $D$ is the input and output dimension of the network.
>
> We are not sure which empirical works the reviewer is referencing but we would like to stress that our outlier eigenvalues are not necessarily the largest magnitude eigenvalues which we suspect may be the nomenclature followed in these papers. In our theoretical results, we distinguish outliers from the bulk based on the fact that their algebraic formula is distinct from the majority of eigenvalues (the bulk). We have added this clarification in the footnote in theorem 3.1.
>
> > Is it possible to derive a two-layer neural network Hessian Lemma under the assumption that the input and/or output dimension equal to 2?
>
> Yes, it is absolutely possible to derive such a lemma using our proposed proof technique. In fact, we have effectively already solved a specific instance of the $D=2$ case in Appendix B.3 (Linear Network with Bias).
>
> Mathematically, a network with a bias term is equivalent to a network with input dimension
> $D=2$ using homogeneous coordinates (where the second input dimension is fixed to
> $x_2=1$. Our derivation in Appendix B.3 demonstrates that the Schur-complement-based technique extends to this setting. While the resulting characteristic polynomial increases in degree (leading to the cubic/quartic solutions found there) and complexity, the "Hessian Lemma" approach remains valid. For a general $D=2$ case with variable inputs, most of the proof technique should transfer but with some appropriate adjustments.
>
> > Is it possible [that] the theorem holds for dimension k+1 if proved for dimension k?
>
> The answer depends on whether one refers to deriving the characteristic polynomial or deriving the explicit roots (eigenvalues).
>
> * Deriving the Polynomial (Yes):
> Our proof technique—using the Schur complement to reduce the Hessian determinant—is indeed inductive in nature. As shown in Appendix B.4 (Corollary B.1), for a multi-dimensional input $D$, the characteristic equation maintains a consistent structural form. Moving from dimension k to k+1 simply adds additional block terms to the determinant formulation.
>
> * Solving for Eigenvalues (No, due to Abel-Ruffini):
> While the equation generalizes, finding the closed-form roots does not. As the input dimension
> $D$ increases (without assuming decorrelation), the degree of the polynomial governing the outliers increases. According to the Abel-Ruffini theorem (which we cite in the Introduction), general polynomial equations of degree ≥5 cannot be solved in closed form using radicals.
>
> In summary, while the method holds inductively, the ability to write down an explicit formula for the eigenvalues breaks down as it starts going over and beyond a quintic, without further assumptions.
>
> We hope that our response satisfactorily addresses the points raised and we are happy to provide further details if needed. We believe the paper has benefited significantly from the extended discussion inspired by this review, so we kindly invite the reviewer to reevaluate the paper and reconsider the final score in light of these clarifications and improvements.

---

### Official Review · Reviewer_UcKL · 2025-10-29

**Soundness:** 2
**Presentation:** 3
**Contribution:** 2
**Rating:** 4
**Confidence:** 2

**Summary:**

The authors provide explicit closed form expression for eigenvalues and eigenvectors of simple neural networks:
- 2 layer linear network from R to R, with 2m parameters (and later they extend this to larger input and output dimensions)
- 2 layer ReLU network from R to R, with 2m parameters
- 1 layer of softmax attention (embedding dimension 1, length n, d_k+1 parameters)

The authors find that in linear models there exists a pair of eigenvalues that sum to the trace of the matrix, and verify that this is approximately true also for larger input and output dimensions (Section 4.1).

They then discuss (numerically) which matrix norm of the weights correlates most with the largest hessian eigenvalue in linear networks, and find that it is the spectral norm. The authors discuss this issue also in transformers.

**Strengths:**

The authors obtain very explicit solutions for the spectrum of the Hessian, which allows for direct interpretability.

**Weaknesses:**

The setting the authors consider is very simple, looking at D=1 inputs and outputs (and relaxing this requirement only in a somewhat technical way in line 358-360) and linear architectures. They also consider the smallest (dimension-wise) possible attention layer, which is nice but one should also remember that softmax is morally an invertible function (modulo an overall shift), so it is not clear to me how much more non-linear this setting is.
The experiments in subsequent sections do not highlight key properties that are both known to be important in more complicated networks, and are already captured by the simple models, putting into question how much general insight we can gain from the presented results.

Section 4.2: it is unclear to me what is the purpose of this section. It seems to me to test quantities/correlations that are not inspired by the theoretical results. Also, the authors limit themselves to numerical analysis of still very simple networks, while they could perform similar analysis on nets that are at least non-linear.
Similar comments hold for Section 4.3, first paragraph.

**Questions:**

line 107: there is a full line of works studying Hessians in spin glass models and simple models of learning, see for e.g. https://arxiv.org/pdf/2006.06997, https://arxiv.org/pdf/2202.04509 and references therein. How does your work relate to those, and in particular, is the statement that no work has obtained closed form expression still valid?

Can you motivate better what is the realtionship between the numerical experiments in Section 4.2 and the previous theoretical analysis?

---

> ### Author Response · Authors · 2025-11-22
> **Response (1/3) to Reviewer UcKL**
>
> We thank the reviewer for the insightful feedback and questions. We address reviewer’s points and questions one by one below. We also mark the changes introduced in the manuscript in blue.
>
> > The setting the authors consider is very simple, looking at D=1 inputs and outputs [...] and linear architectures.
>
> We respectfully note that while the model settings appear simple, deriving these results is non-trivial. The fact that it can be done even for these models is quite surprising due to the reasons described in lines 46-53 of the introduction (equivalence to finding the characteristic polynomial and then solving a high-degree polynomial in closed form). Furthermore, we highlight that our analysis is not limited to linear architectures; Section 3.2 explicitly considers a two-layer ReLU network.
>
> Finally, we would like to invite the reviewer to read our general response, where we argue that studying these simple models is of significant relevance for the DL community.
>
> > They also consider the smallest (dimension-wise) possible attention layer, which is nice but one should also remember that softmax is morally an invertible function (modulo an overall shift), so it is not clear to me how much more non-linear this setting is.
>
> The reviewer is correct that the softmax function is invertible up to a shift. However, we want to argue that this is an interesting setting to study.
>
> 1. **Invertibility is not synonymous with linearity** A function can be both perfectly invertible and highly non-linear, for example tanh or sigmoid activations are invertible, yet they make MLPs nonlinear.
> 2. **Softmax models information selection instead of information loss** Unlike non-invertible ReLU, softmax in self-attention enables data-dependent information selection. However, it can still effectively discard information when attention weights are zero.
> 3. **Softmax attention introduces context-dependent curvature that is non-constant across tokens** In softmax, the normalization term $\sum_j e^{z_j}$ introduces a context-dependent non-linearity where every element of the output sequence is a nonlinear function of the entire input sequence. We believe this makes it substantially different from the other settings considered in the paper.
> 4. **Relevance for the LLM and Transformer optimization community** Softmax is the dominant activation in LLMs. It is also central to the observed optimization-related issues like rank collapse and vanishing gradients in Transformers [1, 2]. Therefore, a theoretical analysis of the softmax-attention Hessian can offer pertinent insights for the optimization community.
>
> Finally, we would like to note that one could apply our proof technique to derive the Hessian spectrum for a self-attention layer with any twice-differentiable activation as long as the activation is applied per-row. We added this information to section 3.3 of the manuscript.
>
>
> > The experiments in subsequent sections do not highlight key properties that are both known to be important in more complicated networks, and are already captured by the simple models
>
> The goal of the experiment section was to verify to what extent the conclusions drawn from the simple models for which we have the closed-form spectra expressions extend to more advanced models. However, we do agree with the reviewer that it is interesting to investigate whether some of the phenomena that are of interest for the deep learning community are already captured by the models considered in this paper.
>
> Inspired by the reviewer’s suggestion, we investigated the progressive sharpening and edge of stability (EoS) phenomena [3] in the self-attention layer (analyzed in section 3.3 of the manuscript). We believe these phenomena to be important for the deep learning optimization community.
>
> In appendix D5 of the updated version of the manuscript, we experimentally demonstrate that both progressive sharpening and EoS happen in this system (see fig. 27). We also analyze the key components of the largest eigenvalue expressions. We note that the progressive sharpening phase coincides with a growing magnitude of the weights and attention moments while the magnitude of the oscillations in the EoS phase matches the magnitude of the loss residual.
>
> What we present here is a preliminary empirical result aiming to demonstrate that these phenomena occur in the system studied in this manuscript. Using our expressions, one could conduct precise mathematical analysis of the phenomena as in [4] but this time for this more realistic network instead of a 4-parameter scalar network. We believe that such an analysis in our simple model of Transformer would be useful for the deep learning theory community.

---

> ### Author Response · Authors · 2025-11-22
> **Response (2/3) to Reviewer UcKL**
>
> > Section 4.2: it is unclear to me what is the purpose of this section. [...] Can you motivate better what is the relationship between the numerical experiments in Section 4.2 and the previous theoretical analysis?
>
> As stated at the beginning of Section 4, the aim of the experiments in this section is to verify the extent to which the closed-form Hessian expressions for the eigenvalues and the ensuing insights extend beyond our theoretical assumptions.
>
> Specifically, Section 4.2 is dedicated to checking if the largest eigenvalue of the Hessian (which is of significant interest to the deep learning community) still depends on the general weight matrix norms if we drop our assumptions on the input/output size and depth. Such dependence is suggested by our theory in Section 3.1 for input/output size equal to 1. So looking at the expression from eq. 2, we try to extrapolate what this expression could look like for a network with larger input/output size and depth.
>
> Since the $\ell_2$ vector norm can generalize to various matrix norms when we increase the input and output size beyond 1, we start by considering the correlations between various weight matrix norms.
>
> After finding the best-correlating norm in the two-layer case, we proceed to check these correlations for deeper networks. The increase in depth, however, introduces another ambiguity into the expression we are trying to extrapolate: Specifically, when considering deeper models, and assuming that a similar expression would hold, it is not immediately clear whether the eigenvalue expressions should depend on norms of weights or norms of products of weights with one weight removed. Note that in the case of a two-layer network, both of these formulations happen to be mathematically equivalent.
>
> We extended the explanation of our reasoning behind these experiments in lines 405 and 427. We hope this has clarified the purpose of this section. Please let us know if this is not the case.
>
> > Similar comments hold for Section 4.3, first paragraph.
>
> The experiments in Section 4.3 serve to verify whether the conclusions drawn from the explicit Hessian spectra of a single self-attention layer extend to a full Transformer model. Specifically, in Section 3.3 (lines 335–346), we discuss a link between attention sinks and loss Hessian eigenvalues of the self-attention model; we empirically verify this connection in a full Transformer in Figure 6. Furthermore, in lines 347–354, we hypothesize that the GGN provides a poorer approximation of the Hessian's largest eigenvalue in Transformers compared to MLPs, based on the structure of the self-attention Hessian eigenvalue expressions. We present the experiment validating this hypothesis in Figure 5.
>
> We modified the first paragraph of section 4.3 to make the purpose of this section clearer and more connected to section 3.3.
>
>
> >  the authors limit themselves to numerical analysis of still very simple networks, while they could perform similar analysis on nets that are at least non-linear.
>
> We focused our numerical analysis on linear networks because they offered clear potential candidates for how the eigenvalues might depend on the weight matrices, which we could then use to extrapolate the expressions to the multi-dimensional input/output setting.
> For ReLU networks, even with an input dimension equal to one, the largest eigenvalue depends on a data-dependent subset of entries from the weight matrices. Theorem 3.4 does not suggest any data-independent expression that we could hypothesize to correlate with the largest eigenvalue.
>
> Nevertheless, inspired by the reviewer's comment, we repeated the experiments from Section 4.2 for the ReLU networks as well. These results can be found in Appendix D.3.3 and D.3.4 of the updated manuscript. While the squared spectral norm of the weight matrices still shows significant correlation with the largest eigenvalue, we found that for two-layer ReLU networks, it is the squared Frobenius norm that correlates the strongest.

---

> ### Author Response · Authors · 2025-11-22
> **Response (3/3) to Reviewer UcKL**
>
> > there is a full line of works studying Hessians in spin glass models and simple models of learning, see for e.g. https://arxiv.org/pdf/2006.06997, https://arxiv.org/pdf/2202.04509 and references therein. How does your work relate to those, and in particular, is the statement that no work has obtained closed form expression still valid?
>
> We appreciate the reviewer highlighting these connections. We have expanded our discussion of spin glass models and Random Matrix Theory (RMT) in Appendix A (lines 855-859, 865-866), specifically addressing the references you have mentioned [6,7].
>
> Regarding the validity of our claim: Yes, our statement stands! There is a fundamental mathematical distinction between the statistical descriptions derived in spin glass literature and the exact, algebraic closed-form solutions we present.
> 1. **Exact Eigenvalues vs. Limiting Spectral Densities**
> The works cited (e.g., [6, 7]) operate in the thermodynamic limit (#parameters -> $\infty$). They utilize RMT to derive a Limiting Spectral Density (LSD), a continuous probability distribution describing the shape of the spectrum (e.g., a semicircle law), or its extremal values.
>
>     In contrast, our work derives exact formulas for all individual eigenvalues and eigenvectors for networks of finite width and arbitrary sample size. We do not describe the statistical distribution of the spectrum; we solve for the spectrum itself which is a much more exacting task.
> 2. **Structural Fidelity vs. Probabilistic Proxies**
> Crucially, to apply RMT tools, previous works must often replace the exact Hessian structure with highly simplified probabilistic proxies. For instance, [6] analyzes phase retrieval (effectively one-layer neural networks) by modeling components of the Hessian as Wigner matrices. This does not even however describe the spectrum of the whole Hessian, as “free-independence” assumptions are needed to handle the sum of the two terms that make up the Hessian [5].
>
>     While useful for macroscopic analysis, these assumptions effectively "wash out" the precise structural constraints imposed by the neural network architecture. Our derivation requires no such substitution; we analyze the Hessian structure exactly as defined by the network's connectivity, preserving the correlations that RMT assumptions often discard.
>
> 3. **Deterministic vs. Stochastic Validity**
> Our formulas are deterministic, meaning they hold for any specific configuration of weights and data, not just in the limit. (Note that one can still use our deterministic formulas to study the limiting distributions)
>
> We trust that the revisions and clarifications provided above effectively resolve reviewer’s concerns, though we remain available to answer any further questions. As reviewer’s feedback has been instrumental in strengthening the manuscript, we respectfully invite the reviewer to reevaluate the work and consider updating the score.
>
> References:
>
> [1] Noci et al., Signal Propagation in Transformers: Theoretical Perspectives and the Role of Rank Collapse, NeurIPS 2022, https://arxiv.org/pdf/2206.03126
>
> [2] Anonymous, Two failure modes of deep transformers and how to avoid them: a unified theory of signal propagation at initialisation, under review at ICLR 2026 https://openreview.net/forum?id=utSqpxQHXq
>
> [3] Cohen et al., Gradient Descent on Neural Networks Typically Occurs at the Edge of Stability, ICLR 2021, https://arxiv.org/abs/2103.00065
>
> [4] Zhu et al., Understanding Edge-of-stability Training Dynamics with a Minimalist Example, ICLR 2023, https://openreview.net/pdf?id=p7EagBsMAEO
>
> [5] Pennington and Bahri, 2017, Geometry of Neural Network Loss Surfaces via Random Matrix Theory, (ICML 2017) https://proceedings.mlr.press/v70/pennington17a/pennington17a.pdf
>
> [6] d’Ascoli et al., Optimal learning rate schedules in high-dimensional non-convex optimization problems, 2022, https://arxiv.org/pdf/2202.04509
>
> [7] Sarao Mannelli et al., Complex dynamics in simple neural networks: Understanding gradient
> flow in phase retrieval, NeurIPS 2020,  https://arxiv.org/abs/2006.06997

---

> > ### Comment · Reviewer_UcKL · 2025-11-25
> >
> > I thank the authors for engaging so nicely with my comments. They addressed most of my comments in detail and to a satisfactory extent. While I maintain my confidence score to 2, I will update my grading to 6.

---

### Official Review · Reviewer_wW7M · 2025-11-03

**Soundness:** 3
**Presentation:** 3
**Contribution:** 3
**Rating:** 6
**Confidence:** 2

**Summary:**

The paper derives closed-form expressions for the complete Hessian spectra (eigenvalues and eigenvectors) of fundamental neural networks, specifically two-layer linear and ReLU MLPs with scalar input/output, arbitrary hidden width, and MSE loss over any samples, plus a single-head self-attention layer with arbitrary sequence length and embedding dim=1. It uncovers a paired outlier eigenvalue structure summing to a trace fraction, cell-wise spectral decomposition for ReLU, Hessian condition number sensitivity to query/key matrix norms, and attention sinks amplifying outliers. Tested empirically on deeper linear networks with varying depths/widths/dims/parametrizations, and GPT2-124M pre-training on OpenWebText. Results show strong sharpness-spectral norm correlations, worse GGN approximation in Transformers vs. MLPs, and paired structure persistence. Overall, it bridges the gap between approximations and exact Hessian analysis, enhancing insights into optimization, generalization, and loss landscapes.

Strength:
- Introduces a novel proof technique yielding exact, interpretable closed-forms for non-trivial NNs, enabling precise theoretical insights where prior work relied on bounds or numerics.

Weaknesses:
- The theoretical results for self-attention are derived under highly restrictive assumptions, such as embedding dimension d=1 and single-head attention, which do not capture the multi-head, high-dimensional nature of real Transformers. This limits generalization, as dimensional interactions could fundamentally alter the spectrum, yet no analysis of higher dimensions is provided.
- The proof technique, while novel, depends on specific structural assumptions (e.g., block-diagonal weights for multi-dimensional extensions, orthogonal v and b in bias cases) that are unjustified in general deep learning settings; failure to discuss scalability or applicability to non-linear, multi-layer architectures risks overstating the method's utility.-

**Strengths:**

- Introduces a novel proof technique yielding exact, interpretable closed-forms for non-trivial NNs, enabling precise theoretical insights where prior work relied on bounds or numerics.

**Weaknesses:**

- The theoretical results for self-attention are derived under highly restrictive assumptions, such as embedding dimension d=1 and single-head attention, which do not capture the multi-head, high-dimensional nature of real Transformers. This limits generalization, as dimensional interactions could fundamentally alter the spectrum, yet no analysis of higher dimensions is provided.
- The proof technique, while novel, depends on specific structural assumptions (e.g., block-diagonal weights for multi-dimensional extensions, orthogonal v and b in bias cases) that are unjustified in general deep learning settings; failure to discuss scalability or applicability to non-linear, multi-layer architectures risks overstating the method's utility.-

**Questions:**

N/A

---

> ### Author Response · Authors · 2025-11-22
> **Response to Reviewer wW7M**
>
> We thank the reviewer for their feedback. We address the points one by one below. In the updated manuscript we mark changes in blue.
>
> >The theoretical results for self-attention are derived under highly restrictive assumptions, such as embedding dimension d=1 and single-head attention, which do not capture the multi-head, high-dimensional nature of real Transformers. [...] no analysis of higher dimensions is provided.
>
> We do acknowledge that these are simplifications of the Transformer architecture. However, our primary goal was to derive exact, closed-form expressions for the Hessian spectrum—a task that is analytically intractable for non-linear architectures without structural assumptions. By isolating the fundamental unit of the Transformer (a single head), we uncover spectral properties that are otherwise obscured by the complexity of full models.
>
> Moreover, our approach is fairly standard in theoretical deep learning literature, where tractable, simplified settings are used to derive insights that govern high-dimensional behavior. For instance,
> von Oswald et al. (2023) https://proceedings.mlr.press/v202/von-oswald23a/von-oswald23a.pdf: To prove the equivalence between In-Context Learning and Gradient Descent, the authors restrict their analysis to a **single-**head linear self-attention layer and simple regression tasks.
>
>
> Tarzanagh et al. (2023) https://arxiv.org/abs/2308.16898: the authors analyze the optimization geometry of a single self-attention layer to formally prove that attention heads maximize the margin of token separation.
>
> Critically, these assumptions do not limit the generalization of our insights. As noted in our experiments (Section 4), the key insights that we gained from our theoretical analysis, such as the role of attention sinks—persist in the full GPT-2 model (d >> 1, and multi-head). This provides strong empirical evidence that the spectral structures that we identify do not fundamentally alter in the general case.
>
>
> >The proof technique, while novel, depends on specific structural assumptions [...] that are unjustified in general deep learning settings; failure to discuss scalability or applicability to non-linear, multi-layer architectures risks overstating the method's utility.
>
> While the specific structural assumptions (such as block-diagonal weights and orthogonality) are simplifications, they essentially serve to isolate and emphasize the block-diagonal structure of the Hessian. It is well-established in prior work [1, 2, 3] that the Hessian is block-diagonally dominant; therefore, our closed-form analysis preserves the most critical facets of it.
>
> Furthermore, the orthogonality assumption is often a standard tool in theoretical analysis, and is also present in past theoretical work on the Hessian, such as Pennington & Bahri (2017) "Geometry of Neural Network Loss Surfaces".
>
> Also, we would like to emphasize that in Section 3.3, we also provide precise insights into how ReLU non-linearities shape the spectrum over linear networks.
>
> To mitigate the risk of overstating our method's utility, we have updated lines 493–498 of the manuscript to explicitly state that the technique is not directly applicable to deep models with arbitrary activations without introducing additional assumptions. We also highlight that finding a right set of assumptions that would allow us to apply the technique to a wider set of models represents a promising direction for future work.
>
> We hope this addresses the reviewer's concerns. We are happy to discuss these assumptions more explicitly in the paper and to further emphasize the simplified nature of these models.
>
> References:
>
> [1] Martens & Grosse, Optimizing Neural Networks with Kronecker-factored Approximate Curvature, 2015, https://arxiv.org/abs/1503.05671
>
> [2] Papyan, Traces of Class/Cross-Class Structure Pervade Deep Learning Spectra, 2020, https://arxiv.org/abs/2008.11865
>
> [3] Dong et al., Towards Quantifying the Hessian Structure of Neural Networks, 2025, https://arxiv.org/pdf/2505.02809

---

### Author Response · Authors · 2025-11-22
**General Response to All Reviewers**

We would like to thank all the reviewers for their constructive feedback and the reception of our results, with reviewers noting that they “provide valuable analytical tools for understanding loss landscapes” [iw5q], enhance "insights into optimization, generalization, and loss landscapes” [wW7M], and offer “new insights into Hessian behavior throughout training” [iw5q].

Regarding our theoretical contributions, we are glad to know that reviewers recognize our “novel proof technique” [wW7M] which yields “first successful derivation of complete, exact Hessian spectra for non-trivial neural networks without approximations” [iw5q] and which allows for direct “interpretability” [wW7M, UcKL]. Besides, it is also helpful to know that the reviewers also appreciate our empirical verifications. Specifically, they note that our empirical studies aid “comprehensive in-depth understanding” [Lt5D], “provide actionable insights” [iw5q], and “offer concrete guidance for [...] architecture design” [iw5q].

We nevertheless do understand that the reviewers still have some remaining concerns and we hope to address them thoroughly in the individual rebuttal responses. But, in our general response, we would like to emphasize the importance of utilizing relatively simple models for our theoretical analysis.


* *Closed-form Hessian Spectrum is a surprising feat:*  Firstly, if we reflect upon the complexity of the underlying object of study, namely the Hessian of the loss and its spectrum, it frankly comes as a surprise that we can actually even get a closed-form for a non-trivial family of neural network models! Especially, when there are no restrictions in our results upon the number of data points or the number of parameters in the network. Prior work had largely operated under the presumption of the closed-form of the spectrum being intractable, and our work marks a strong departure from this underlying tenet.

* *The Power of Simplified Models:* Our methodology follows a rich tradition in theoretical deep learning, where simplified models are the standard tool for isolating and explaining complex phenomena. For example, Saxe et al. (2014) utilized deep linear networks to solve non-linear learning dynamics; Cohen et al. (2021) relied on quadratic approximations to discover the "Edge of Stability"; and von Oswald et al. (2023) restricted analysis to single-head linear attention to explain In-Context Learning. In this spirit, through our simplified models, we are able to provide a mathematically tractable and rigorous analysis of the Hessian spectra.

* *Predictive Power and Generalizability:* Crucially, these simplified models remain strictly relevant in the face of general architectures because their predictions hold up in broader settings. For instance, our theoretical analysis predicts specific spectral signatures, such as paired outliers, which we empirically verify in more complex settings. Likewise we show that the Transformer-related Hessian insights also hold in the GPT-2 setup, despite the simplified model we used for our theoretical analysis.

---

### Author Response · Authors · 2025-12-02
**Summary of the Discussion phase**

We would like to thank the reviewers and the AC for their time and constructive efforts. We believe that incorporating the review feedback has significantly strengthened the paper—a sentiment reflected by *Reviewer UcKL raising their score to 6*.

Although the discussion period was unfortunately cut short before other reviewers could respond, *Reviewer iw5q previously expressed a willingness to reconsider their score*, and we have since addressed their concerns in detail. Furthermore, since our *revisions addressed several shared concerns, we are optimistic* that these improvements would resonate similarly with the other reviewers.



Lastly, we remain fully available to answer any further questions or clarify any details to assist the AC in their final decision.

---

### Meta-Review · Area_Chair_iUSd · 2025-12-22

**Summary:**

This paper presented closed-form expressions for the complete Hessian spectra of fundamental neural networks and uncovered a `paired’ structure of eigenvalues that sum to the matrix trace, which is unified both theoretically and empirically. The proposed Hessian interpretation are the first exact derivation in the field, and may serve as valuable analytical tools to understand the optimization landscape in deep learning.

However, the reviewers raise several core concerns including (1) Highly simplified assumptions, (2) Large theoretical-empirical gap, and (3) Lack direct practical implications.

The author’s rebuttal have addressed part of the concerns, while outstanding issues in presentation and lack of integrity are considered critical for accepted at the conference.

In summary, I believe this submission is slightly below the standard of ICLR. I recommend the submission for Reject. I strongly encourage the authors to further polish the contents accordingly.

**Reviewer Concerns:**

I think the authors have addressed part of the concerns, while some major concerns are still outstanding.

As the summary made by the authors suggest, the detailed responses addressed the main concerns, respectively.

Addressed Concerns:

- Spectral Norm – Eigenvalue Association: The authors have supplied a formal explanation in Appendix D.3 to resolve Reviewer iw5q’s concern.

- Proof Technique Scalability: The authors have revised the manuscript with advanced discussions on the assumptions and generalizability of the conclusions in section 3.3, and modified the explanation of the motivation and presentation in section 4.2, 4.3, regarding the different concerns of all reviewers.

- Discussions to Related Works: The authors have expanded the discussions of spin glass models and Random Matrix Theory in Appendix A (as response to Reviewer UcKL), theoretical studies on neural optimization in response to Reviewer wW7M and iw5q.

Outstanding Concerns:

- Simplified Assumption: Reviewer wW7M, UcKL, Lt5D have raised the concern that the theoretical studies in this paper are derived under the assumptions on D=1 inputs/outputs. While the authors have supplemented results of the D=2 case in Appendix B.3. and provided explanations on the verifiable empirical observations, they should further discuss the tractability of their theoretical results in the manuscript.

- Large Theoretical-Empirical Gap: The authors have proposed a guidance to verify the approximations of simplified systems in Appendix D.5 to resolve Reviewer iw5q’s concern, and supplied experiment results for the ReLU networks in Appendix D.3 to resolve Reviewer UcKL’s concern. However, the explanations still have limitations in practical implications.

- Presentation Clarity: Instead of presenting the expanded study beyond the simplified assumptions (D=1) in the Appendix, it would be nice to demonstrate the tractability of the proof technique further in the manuscript.

- Novelty of the work: The authors partially addressed the Reviewer’s concerns regarding the restrictive assumptions from two core dimensions: (1) the simplified settings are widely accepted in studies w.r.t. neural networks, and (2) the theoretical predictions hold up in broader settings. Nevertheless, it is concerned that the causal relationship between the proposed theoretical insights and the empirical observations are less clarified.

**Reviewer Scores:**

Reviewer wW7M: 6 -> 6
- Justification: I think the authors have partially addressed the raised main concerns W1 by relating to relevant literatures in the field. However, the response to W2 (the novelty of relying on structural assumptions) may be less than satisfying.

Reviewer UcKL: 4 -> 6
- Justification: I think the authors have addressed the raised main concern with additional experiments on the progressive sharpening and edge of stability phenomena.

Reviewer Lt5D: 4 -> 4
- Justification: I think the authors have partially addressed the mentioned Q1 and Q3 questions. However, the discussion around Q2 (empirical observations on eigenvalue outliers) and the general response to W1 is not fully explored.

Reviewer iw5q: 4 -> 4
- Justification: I think the authors have addressed the reviewer’s concerns on W3. However, the responses to W1 and W2 are not fully satisfying, partially limited by the existing scope of the paper.

---

### Decision · Program_Chairs · 2026-01-26

Reject